# Probing the subtropical lowermost stratosphere, tropical upper troposphere, and tropopause layer for inorganic bromine

B. Werner[1], J. Stutz[2], M. Spolaor[2], L. Scalone[1], R. Raecke[1], J. Festa[2], F. Colosimo[2], R. Cheung[2], C. Tsai[2], R. Hossaini[3], M. P. Chipperfield[4], G. S. Taverna[4], W. Feng[5], J. W. Elkins[6], D. W. Fahey[6], Ru-Shan Gao[6], E. J. Hintsa[6,7], T. D. Thornberry[6,7], F. L. Moore[6,7], M. A. Navarro[8], E. Atlas[8], B. Daube[9], J. Pittman[9], S. Wofsy[9], and K. Pfeilsticker[1]

[1]Institute of Environmental Physics, University of Heidelberg, Heidelberg, Germany
[2]Department of Atmospheric and Oceanic Science, University of California Los Angeles, Los Angeles, California, USA
[3]Lancaster Environment Centre, University of Lancaster, Lancaster, UK
[4]Institute for Climate and Atmospheric Science, School of Earth and Environment, University of Leeds, Leeds, UK
[5]National Centre for Atmospheric Science, School of Earth and Environment, University of Leeds, UK
[6]NOAA Earth System Research Laboratory, Boulder, Colorado, USA
[7]Cooperative Institute for Research in Environmental Sciences (CIRES), University of Colorado, Boulder, Colorado, USA
[8]The Rosenstiel School of Marine and Atmospheric Science, University of Miami, Miami, Florida, USA
[9]School of Engineering and Applied Sciences, Harvard University, Cambridge, Massachusetts, USA

*Correspondence to:* Klaus Pfeilsticker
(Klaus.Pfeilsticker@iup.uni-heidelberg.de)

**Abstract.** We report on measurements of $CH_4$ (in-situ measured by the Harvard HUPCRS and NOAA UCATS instruments), $O_3$ (in-situ measured by the NOAA dual-beam UV photometer) $O_3$, $NO_2$, $BrO$ (remotely detected by spectroscopic UV/vis limb observations, see the companion paper of Stutz et al. (2016)) and of some key brominated source gases in whole air samples of the Global Hawk Whole Air Sampler (GWAS) instrument within the subtropical lowermost stratosphere (LS), tropical upper troposphere (UT) and tropopause layer (TTL). The measurements were performed within the framework of the NASA-ATTREX (National Aeronautics and Space Administration - Airborne Tropical Tropopause Experiment) project from aboard the Global Hawk (GH) during 6 deployments over the Eastern Pacific in early 2013. These measurements are compared with TOMCAT/SLIMCAT 3-D model simulations, aiming at improvements of our understanding of the bromine budget and photochemistry in the LS, UT, and TTL.

Changes in local $O_3$ (and $NO_2$ and $BrO$) due to transport processes are separated from photochemical processes in intercomparisons of measured and modeled $CH_4$ and $O_3$. After excellent agreement is achieved among measured and simulated $CH_4$ and $O_3$, measured and modeled $[NO_2]$ are found to closely agree with $\leq 15$ ppt in the TTL (which is the detection limit) and a typical range of 70 to 170 ppt in the subtropical LS at daytime. Measured $[BrO]$ ranges between 3 - 9 ppt in the subtropical LS. In the TTL, $[BrO]$ reaches $0.5 \pm 0.5$ ppt at the bottom (150 hPa/355 K/14 km) and up to about 5 ppt at the top (70 hPa/425 K/18.5 km, for the used TTL definition see Fueglistaler et al. (2009)), in overall good agreement with the model simulations. Depending on the photochemical regime, the TOMCAT/SLIMCAT simulations tend to slightly under-predict measured $BrO$ for large $BrO$ concentrations, i.e. in the upper TTL and LS. The measured $BrO$ and modeled $BrO/Br_y^{inorg}$ ratio is further used to calculate inorganic bromine, $Br_y^{inorg}$. For the TTL (i.e. when $[CH_4] \geq 1790$ ppb), $[Br_y^{inorg}]$ is found to increase from a mean

of 2.63 $\pm$ 1.04 ppt for potential temperatures ($\theta$) in the range of 350 - 360 K to 5.11 $\pm$ 1.57 ppt for $\theta = 390 - 400$ K, whereas in the subtropical LS (i.e. when [CH$_4$] $\leq$ 1790 ppb), it reaches 7.66 $\pm$ 2.95 ppt for $\theta$ in the range of 390 - 400 K. Finally, for the Eastern Pacific (170°W - 90°W) the TOMCAT/SLIMCAT simulations indicate a net loss of ozone of - 0.3 ppbv/day at the base of the TTL ($\theta = 355$ K) and a net production of + 1.8 ppbv/day in the upper part ($\theta = 383$ K).

**1   Introduction**

At present bromine is estimated to be responsible for roughly 1/3 of the photochemical loss in global stratospheric ozone (WMO, 2014). Past research has revealed that total stratospheric bromine (Br$_y$) has (in 2013) 4 major sources, or contributions: (1) CH$_3$Br which is emitted by natural and anthropogenic sources with a present contribution of 6.9 ppt to Br$_y$, (2) 4 major Halons (CClBrF$_2$ or Halon-1211; CBrF$_3$ or Halon-1301; CBr$_2$F$_2$ or Halon-1202 and CBrF$_2$CBrF$_2$ or Halon-2402) all

emitted from anthropogenic activities with a present contribution of 8 ppt to Br$_y$, (3) so-called very short-lived species (VSLS), and (4) inorganic bromine transported into the upper troposphere, e.g. previously released from brominated VSLS and/or sea salt (e.g., Saiz-Lopez et al. (2004), Fernandez et al. (2014), Schmidt et al. (2016)). This inorganic bromine is also partly transported into the stratosphere. Together sources 3 and 4 are assessed to contribute 5 (2 - 8) ppt to stratospheric bromine (WMO, 2014). Previous assessments of total Br$_y$ and its trend revealed [Br$_y$] levels of $\approx$ 20 ppt (16 - 23 ppt) in 2011, which

has been decreasing at a rate of -0.6 %/yr since the peak levels observed in 2000. This decline is consistent with the decrease in total organic bromine in the troposphere based on measurements of CH$_3$Br, and the Halons (WMO, 2014).

Estimates of stratospheric Br$_y$ essentially rely on two methods: First, the so-called organic (Br$_y^{\text{org}}$) method, where all bromine from organic source gases (SG) found at stratospheric entry level is summed (Wamsley et al. (1998), Pfeilsticker et al. (2000), Sturges et al. (2000), Brinckmann et al. (2012), Navarro et al. (2015)). Second, total inorganic bromine (Br$_y^{\text{inorg}}$)

is inferred from atmospheric measurements (e.g., performed from the ground, aircraft, high-flying balloons, or satellites) of the most abundant Br$_y$ species, BrO, assisted by a suitable correction for the Br$_y^{\text{inorg}}$ partitioning inferred from photochemical modeling (e.g., Pfeilsticker et al. (2000), Richter et al. (2002), Van Roozendael et al. (2002), Sioris et al. (2006), Dorf et al. (2006a), Hendrick et al. (2007), Dorf et al. (2008), Theys et al. (2009), Theys et al. (2011), Rozanov et al. (2011), Parrella et al. (2013), Stachnik et al. (2013)). Further constraints on stratospheric Br$_y$ (range 20 - 25 ppt) were obtained by satellite-

borne measurements of BrONO$_2$ in the mid-IR spectral range at nighttime (Höpfner et al., 2009). While the organic method is rather precise for the measured species (accuracies are several tenths of a ppt), it suffers from the shortcoming of not accounting for any inorganic bromine (contribution 4) directly entering the stratosphere. Uncertainties in the inorganic method arise from uncertainties in measuring BrO as well as from modeling Br$_y^{\text{inorg}}$ partitioning, of which the combined error amounts to $\pm(2.5 - 4)$ ppt, depending on the type of observation and probed photochemical regime.

Past in-situ measurements of Br$_y^{\text{org}}$ were performed at different locations and seasons within the upper troposphere, TTL, and stratosphere. In the present context the most important were measurements performed within the TTL (for the TTL definition see Fueglistaler et al. (2009)) over the Pacific from where most of the stratospheric air is predicted to originate (e.g., Fueglistaler et al. (2009), Aschmann et al. (2009), Hossaini et al. (2012b), Ashfold et al. (2012), WMO (2014), Orbe et al. (2015)). These

include the measurements (a) by Schauffler et al. (1993, 1998, 1999), who found [VSLS] = 1.3 ppt (contribution 3) at the tropical tropopause over the central Pacific (Hawaii) in 1996, (b) by Laube et al. (2008), and Brinckmann et al. (2012) with [VSLS] = 2.25 $\pm$ 0.24 ppt (range 1.4 - 4.6 ppt) and [VSLS] = 1.35 ppt (range 0.7 - 3.4 ppt) found within the TTL over northeastern Brazil in June 2005 and June 2008, respectively, and (c) most recently those by Navarro et al. (2015), who found [VSLS] = 2.96 $\pm$ 0.42 ppt and 3.27 $\pm$ 0.49 ppt at 17 km over the tropical Eastern and Western Pacific in 2013, and 2014, respectively. Information on contribution 3 was further corroborated by measurements performed in the upper tropical troposphere by (b) Sala et al. (2014) who found [VSLS] = 3.72 $\pm$ 0.60 ppt in the upper tropical troposphere over Borneo in fall 2011 and (c) by Wisher et al. (2014), who inferred [VSLS] = 3.4 $\pm$ 1.5 ppt for the CARIBIC flights from Germany to Venezuela/Colombia during 2009 - 2011, Germany to South Africa during 2010 and 2011, and Germany to Thailand/Kuala Lumpur, Malaysia during 2012 and 2013.

Supporting information on brominated VSLS concentrations typical for the boundary layer of the Western Pacific came from measurements performed during the TransBrom ship cruise in October 2009 (median 2.23 ppt and range from 1.45 - 4.14 ppt, Brinckmann et al. (2012)) and the VSLS measurements made around Borneo during the SHIVA project (median 5.7 ppt and range from 3.9 - 10.7 ppt, Sala et al. (2014)). Corroborating model calculations to these field studies by (a) Tegtmeier et al. (2012) indicated that from the Western Pacific on average only 0.4 ppt and at maximum up to 2.3 ppt of the emitted VSLS bromine may reach the stratosphere, while (b) Liang et al. (2014) estimated that up to 8 ppt of VSLS bromine may enter the base of the TTL at 150 hPa, whereby the VSLS emissions from the tropical Indian Ocean, the tropical Western Pacific, and off the Pacific coast of Mexico are suspected to be most relevant, and finally (c) the CAM-Chem modeling performed within the study of Navarro et al. (2015) which indicates that over the Eastern and Western Pacific contributions 3 and 4 (called [VSLS + $Br_y^{inorg}$] in the study) amount to 6.20 ppt (range 3.79 - 8.61 ppt) and 5.81 ppt (range 5.14 - 6.48 ppt), respectively.

Using the inorganic method contributions 3 and 4 have been indirectly estimated from measured $BrO$ performed at the ground, high flying balloons, or satellites (e.g., Pfeilsticker et al. (2000), Richter et al. (2002), Van Roozendael et al. (2002), Sioris et al. (2006), Dorf et al. (2006b), Hendrick et al. (2007), Dorf et al. (2008), Theys et al. (2009), Theys et al. (2011), Rozanov et al. (2011), Parrella et al. (2013), Stachnik et al. (2013)). All together these studies pointed to a range between 3 - 8 ppt with a mean of 6 ppt for contributions 3 and 4. The most direct information on contributions 3 and 4 came by the studies of Dorf et al. (2008), WMO (2011), and Brinckmann et al. (2012). They inferred 1.25 $\pm$ 0.16 ppt (VSL-SGs, contribution 3) + 4.0 $\pm$ 2.5 ppt (PGs, contribution 4) = 5.25 $\pm$ 2.5 ppt (contributions 3 and 4) and 2.25 $\pm$ 0.24 ppt (VSL-SGs, contribution 3) + 1.68 $\pm$ 2.5 ppt (PGs, contribution 4) = 3.98 $\pm$ 2.5 ppt (contributions 3 and 4) from two balloon-borne soundings performed in the TTL and stratosphere over northeastern Brazil during the dry season in 2005, and 2008, respectively. The inferred bromine was thus often larger than [VSLS] inferred using the organic method (contribution 3), indicating that variable amounts of $Br_y^{inorg}$ (i.e. several ppt) are directly transported from the troposphere into the stratosphere (contribution 4).

Based on these findings, Saiz-Lopez et al. (2012) and Hossaini et al. (2015) provided evidence for the efficiency of short-lived halogens to influence climate through depletion of lower stratospheric ozone (for contribution 3), but without explicitly considering the effect of inorganic bromine readily transported across the tropical tropopause (i.e. contribution 4). They concluded that VSLS bromine alone exerts a 3.6 times larger ozone radiative effect than due to long-lived halocarbons, when

normalized to their halogen content. Moreover the benefit for ozone and UV radiation due to the declining stratospheric chlorine and bromine since the implementation of the Montreal protocol was quantified in a recent study by Chipperfield et al. (2015). Finally, in a recent study Fernandez et al. (2016) pointed out that bromine from contribution 3 and 4 contribute about 14% to the formation of the present antarctic ozone hole, in particular at its periphery. Further, they suggests a large influence of biogenic bromine on the future Antarctic ozone layer.

The present paper reports on measurements of $BrO$ (and $NO_2$, $O_3$, $CH_4$, and the brominated source gases) made during the ATTREX deployments of the NASA Global Hawk into the LS, UT, and TTL of the Eastern Pacific in early 2013. Corresponding data collected during the Western Pacific deployments in early 2014 will be reported in a forthcoming paper, primarily since most of the 2014 measurements were performed under TTL cirrus-affected conditions, for which the interpretation of UV/vis spectroscopic measurements is not straightforward (see below). The present paper further addresses the amount of inorganic bromine found in the TTL, and its transport into the lowermost tropical stratosphere (contribution 4), together with the implications for ozone.

Our study accompanies those of Navarro et al. (2015) and Stutz et al. (2016). While Stutz et al. (2016) discusses the instrumental details and the methods employed to remotely measure $BrO$, $NO_2$, and $O_3$, the study of Navarro et al. (2015) reports on the GWAS measurements of $CH_3Br$ (contribution 1), the Halons (contribution 2), and the brominated VSLS (contribution 3) analyzed in whole air samples, which were simultaneously taken from aboard the NASA Global Hawk over the Eastern and Western Pacific during the 2013 and 2014 deployments, respectively.

The paper is organized as follows. Section 2 briefly describes all key methods used in the present study. Section 3 discusses the measurements and along with some (necessary) data reduction. In section 4, the major observations are presented and they are compared with previous $BrO$ measurements and our modelling results, along with their implications for the amount of inorganic bromine present within the TTL. Further implications of our measurements for the photochemistry of bromine and ozone within the TTL and lowermost subtropical stratosphere are discussed. Section 5 concludes the study.

## 2 Methods

The instruments of the NASA-ATTREX package most important for the present study consist of a fast UV photometer for measurement of ozone (Gao et al., 2012), a gas chromatograph (UCATS, Wofsy et al. (2011) and Moore et al. (2003)) as well as a Picarro instrument (HUPCRS, Crosson (2008), Rella et al. (2013), and Chen et al. (2013)) to measure $CH_4$, $CO_2$, and $CO$, a whole air sampler (GWAS, Schauffler et al. (1998) and Schauffler et al. (1999)) to analyze a large suite of stable trace gases, and a 3-channel scanning limb mini-DOAS instrument for spectroscopic detection of $O_3$, $NO_2$, $BrO$, $OClO$, $IO$, $O_4$, $O_2$, $H_2O_{vapor}$, $H_2O_{liquid}$, and $H_2O_{solid}$ in the UV/vis/near-IR spectral ranges (e.g., Weidner et al. (2005), Platt and Stutz (2008), Kritten et al. (2010), Kreycy et al. (2013), Kritten et al. (2014), Stutz et al. (2016)).

All instruments, techniques, methods, and tools are briefly described in the following.

## 2.1 DOAS measurements of $O_3$, $NO_2$, and BrO

The mini-DOAS instrument is a UV/vis/near-IR 3-channel optical spectrometer by which scattered skylight received from limb direction and direct sunlight can be analyzed for $O_3$, $NO_2$, and BrO (beside for some other species, see above). Since the instrument and retrieval methods are described in detail in the accompanying paper by Stutz et al. (2016) (for further details see Table 2 therein), only some key elements of the data analysis are described here.

The post-flight analysis of the collected data for the detection of $O_3$, $O_4$, $NO_2$, and BrO and concentration retrieval include (a) the spectral retrieval of the targeted gases using the DOAS method (Platt and Stutz, 2008) (for the DOAS settings see Table 4 in Stutz et al. (2016)), (b) forward RT-modeling of each observation using the Monte Carlo model McArtim (Deutschmann et al., 2011) (for further details see section 2.6), and (c) for the concentration and profile retrieval either the non-linear optimal estimation (Rodgers, 2000), or the novel x-gas scaling technique (for details see sections 4.2. and 4.3 in Stutz et al. (2016)). Typical errors are $\pm 5$ ppb for $O_3$, $\pm 15$ ppt for $NO_2$, and $\pm 0.5$ ppt for BrO, to which possible systematic errors in the individual absorption cross section need to be added. These are for $O_3$-UV $\pm 1.3\%$, $O_3$-vis $\pm 2\%$, $NO_2$ $\pm 2\%$, and BrO $\pm 10\%$, respectively (for more details of error budget see Stutz et al. (2016)).

## 2.2 In-situ measurements of $O_3$

The NOAA-2 polarized $O_3$ photometer (Gao et al., 2012) is a derivative of the dual-beam, unpolarized, UV absorption technique described by Proffitt and McLaughlin (1983). Briefly, the ambient and $O_3$-free air flow is alternately directed into two identical 60 cm long absorption cells. The 253.7-nm UV light from a mercury lamp is split into two beams that are each directed into one of the absorption cells. Since $O_3$ strongly absorbs 253.7-nm photons, the UV beam passing through the cell containing ambient ozone is attenuated more than the beam passing through the cell containing $O_3$-free air. Knowing the $O_3$ absorption cross section ($\sigma(O_3)$) and the absorption path length (L), the $O_3$ partial pressure ($p(O_3)$) in the ambient air can be derived using Beer's law.

The instrument has a fast sampling rate (2 Hz at $< 200$ hPa, 1 Hz at 200 to 500 hPa, and 0.5 Hz at $\geq 500$ hPa), high accuracy (3% excluding operation in the 300 - 450 hPa range, where the accuracy may be degraded to about 5%), and excellent precision ($1.1 \times 10^{10}$ $O_3$ molecules/cm$^3$ at 2 Hz, which corresponds to 3.0 ppb at 200 K and 100 hPa, or 0.41 ppb at 273 K and 1013 hPa). The size (36 l), weight (18 kg), and power (50 - 200 W) make the instrument suitable for many unmanned aerial vehicle systems and other airborne platforms. In-flight and laboratory inter-comparisons with existing $O_3$ instruments have shown that measurement accuracy (3 %) is maintained in flight.

## 2.3 $CH_4$ measurements by UCATS

The Unmanned aircraft system Chromatograph for Atmospheric Trace Species (UCATS) measures atmospheric methane ($CH_4$) on one gas chromatographic channel along with hydrogen ($H_2$) and carbon monoxide (CO) once every 140 seconds. UCATS has two chromatographic channels with electron capture detectors (ECDs), two ozone ($O_3$) ultraviolet absorption spectrometers, and a water vapor ($H_2O$) tunable diode laser absorption spectrometer (TDLAS). The details of the $CH_4$ chro-

matography are similar to those on balloon and airborne instruments described in Moore et al. (2003) and Elkins et al. (1996). The addition of 100 ppm of nitrous oxide to the make-up line of the ECD enhances the sensitivity to $H_2$, CO, and $CH_4$ (Elkins et al. (1996) and Moore et al. (2003)). The separation of these gases in air is accomplished with a pre-column of Unibeads (2 m × 2 mm diameter), and a main column of molecular sieve 5A (0.7 m × 2.2 mm diameter) at $\sim 110$ °C (Moore et al., 2003). The precision of the $CH_4$ measurement during ATTREX was ±0.5% and is calibrated during flight with a secondary standard after every three ambient air measurements. Instrumental drift is corrected between the standard injections. UCATS measurements are traceable to the WMO Central Calibration Laboratory (CCL) and are on the $CH_4$ WMO X2004A scale (Dlugokencky et al. (2005), with updates given at http://www.esrl.noaa.gov/gmd/ccl/ch4_scale.html).

## 2.4  $CH_4$ measurements by HUPCRS

The Harvard University Picarro Cavity Ringdown Spectrometer (HUPCRS) consists of a G2401-m Picarro gas analyzer (Picarro Inc., Santa Clara, CA, USA) repackaged in a temperature-controlled pressure vessel, a separate calibration system with 2 multi-species gas standards, and an external pump and pressure control assembly designed to allow operation at a wide range of altitudes. The Picarro analyzer uses Wavelength-Scanned Cavity Ringdown Spectroscopy (WS-CRDS) technology to make high precision measurements of greenhouse gases (Crosson (2008), Rella et al. (2013), and Chen et al. (2013)). HUPCRS reports concentrations of $CO_2$, $CH_4$, and CO every $\sim 2.2$ seconds and the data are averaged to 10 seconds. In-flight precision for $CH_4$ is 0.2 ppb in 10 seconds.

Briefly, the analyzer uses three distributed feedback (DFB) diode lasers in the spectral region of 1.55 to 1.65 $\mu$m. Monochromatic light is injected into a high-finesse optical cavity with a volume of 35 $cm^3$ and a configuration of three highly reflective mirrors ($\geq$ 99.995%). Internal control loops keep the cavity at 140 ± 0.02 Torr and 45 ± 0.0005 °C in order to stabilize the spectra. The injected light is blocked periodically and when blocked, the exponential decay rate of the light intensity is measured by a photo-detector. The decay rate depends on loss mechanisms within the cavity such as mirror losses, light scattering, refraction, and absorption by a specific analyte. A sequence of specific wavelengths for each molecule is injected into the cavity in order to reconstruct the absorption spectra. A fit to the spectra is performed in real time and concentrations are derived based on peak height. High-altitude sampling (i.e. very low pressure and temperature) necessitated transferring the core components of the Picarro analyzer to a sealed tubular pressure vessel, which is maintained at 35 ° C and 760 Torr. The analyzer's components are isolated from the pressure vessel to provide vibration damping and decoupling from deformations in the pressure vessel caused by external pressure changes.

The sampling strategy for HUPCRS consists of bringing in air through a rear-facing inlet, filtered by a 2 $\mu$m Zefluor membrane, and dehydrating this air by flowing it through a multi-tube Nafion dryer followed by a dry-ice cooled trap prior to entering the Picarro analyzer. A choked upstream Teflon-lined diaphragm pump delivers ambient air to the analyzer at 400 Torr, regardless of aircraft altitude, via a flow bypass. A similar downstream pump, with an inlet pressure of 10 Torr, facilitates flow through the analyzer at high altitude and ensures adequate purging of the Nafion drier. Measurement accuracy and stability are monitored by replacing ambient air with air from two NOAA-traceable gas standards (low- and high-span) for a total of four

minutes every 30 minutes. These standards are contained in 8.4 L carbon fiber wrapped aluminum cylinders and housed in a temperature-controlled enclosure. The total weight of the package is 97 kg.

## 2.5 The GH Whole Air Sampler (GWAS)

The Global Hawk Whole Air Sampler (GWAS) is a modified version of the Whole Air Sampler used on previous airborne campaigns (Heidt et al. (1989), Schauffler et al. (1998, 1999), and Daniel et al. (1996)). Briefly, the instrument consists of 90 custom-made canisters, Silonite-coated (Entech Instruments, Simi Valley, CA) of 1.3 L, controlled with Parker Series 99 solenoid valves (Parker-Hannifin, Corp., Hollis, NH). Two metal bellows compressor pumps (Senior Aerospace, Sharon, MA) allow the flow of ambient air through a custom inlet at flow rates ranging from 2 to 8 standard L/minute, depending on altitude. The manifold and canister module temperatures are controlled to remain within the range of 0 - 30 °C. GWAS is a fully automated instrument controlled from the ground through an Ethernet interface. Parameters to fill the canisters, flush the manifold, and control the temperature, are pre-determined in the Data System Module (DSM) inside the aircraft, to fill the canisters automatically in case of failure of the aircraft networks. However, during the entire flight, the parameters are manually set with the ground laptop computer to improve the sampling collection at different altitudes. During the ATTREX campaign, the canisters were filled to $\sim$ 3 standard atmospheres (40 psi) in about 25 sec at 14 km and 90 sec at 18 km. The samples are analyzed using a high performance gas chromatograph (Agilent Technology 7890A) and mass spectrometer with mass selective, flame ionization and electron capture detector (Agilent Technology 5975C). Samples are concentrated on an adsorbent tube at -38 °C with a combination of cryogen-free automation and thermal desorber system (CIA Advantage plus UNITY 2, Markes International). The oven temperature profile is -20 °C for 3 min, then 10 °C/min to 200 °C and 200 °C for 4 min, for a total analysis time of 29 min. Under these sampling conditions the precision is compound/concentration dependent, and ranged from $\leq$2% to 20%. Calibration procedures as well as mixing ratios calculations are described elsewhere (Schauffler et al., 1999).

During ATTREX 2013 the whole air sampler measured a variety of organic trace gases, including non-methane hydrocarbons, CFCs, HCFCs, methyl halides, solvents, organic nitrates, and selected sulfur species. For this work a range of long and short lived organic bromine gases are measured including $CH_3Br$, $CH_2Br_2$, $CH_2BrCl$, $CHBrCl_2$, $CHBr_2Cl$, $CHBr_3$, and Halon 1211 ($CBrClF_2$) and Halon 2402 ($C_2Br_2F_4$). Halon 1301 ($CBrF_3$) is not measured and a constant value of 3.3 ppt is used to account for the bromine content from this compound.

## 2.6 Radiative transfer modeling

The measured limb radiances of the mini-DOAS instrument are modeled in spherical 1D, and in selected cases in 3D, using version 3.5 of the Monte Carlo radiative transfer (RT) model McArtim (Deutschmann et al., 2011). The model's input is chosen according to the on-board measured atmospheric temperatures and pressures, including climatological low latitude aerosol profiles from SAGE III (http://www.eosweb.larc.nasa.gov/PRODOCS/sage3/table-/sage3.htm), and lower atmospheric cloud covers as indicated by the cloud physics lidar measurements made from aboard the GH (see http://cpl.gsfc.nasa.gov/). In the standard run, the ground (oceanic) albedo is set to 0.07 in UV, and 0.2 in visible spectral range. The RT model is

further fed with the actual geolocation of the GH, solar zenith and azimuth angles as encountered during each measurement, the telescopes azimuth and elevation angles, as well as the field of view (FOV) of the mini-DOAS telescopes. Fig. 5 in Stutz et al. (2016) displays one example of an RT simulation for limb measurements at 18 km altitude. The simulation indicates that correctly accounting for the Earth's sphericity, the atmospheric refraction, cloud cover, ground albedo etc. is relevant

for the interpretation of UV/vis/near-IR limb measurements performed within the middle atmosphere (Deutschmann et al., 2011). Even though the 3 (UV/vis/near-IR) mini-DOAS spectrometers are not radiometrically calibrated on a absolute scale, past comparison exercises of measured and McArtim modeled limb radiance provide confidence on the quality of the RT simulations (e.g., see Fig. 5 and Fig. 6 in Deutschmann et al. (2011) and Fig. 2 in Kreycy et al. (2013)).

For the simulations of the trace gas absorptions measured in limb direction, the RT model is further fed with TOM-

CAT/SLIMCAT simulated curtains of the targeted gases simulated along the GH flight paths (see section 2.7). In the RT simulations [BrO] is set to 0.5 ppt near the ground, where TOMCAT/SLIMCAT predicts lower $BrO$ concentrations (see Figure 2 middle right panel), in agreement with the findings discussed in Stutz et al. (2016) and the recent study of Schmidt et al. (2016).

## 2.7    Photochemical modeling

For the interpretation of our measurements, we use simulations of the TOMCAT/SLIMCAT 3-D chemical transport model (CTM) (Chipperfield, 1999, 2006). More specifically, the simulations are used for inter-comparison with measured photochemical species, for assessment of the budget of $Br_y^{inorg}$, and for sensitivity studies on the impact of our measurements on the photochemistry of bromine and ozone in the subtropical UT/LS, tropical UT, and TTL.

For the present study the TOMCAT/SLIMCAT model is driven by meteorology from the ECMWF ERA-interim reanalyses

(Dee et al., 2011). The reanalyses are used for the large-scale winds and temperatures as well as convective mass fluxes (Feng et al., 2011). The model has a detailed stratospheric chemistry scheme with kinetic and photochemical data taken from JPL-2011 (Sander et al., 2011) with recent updates. The model chemical fields are constrained by specified time-dependent surface mixing ratios. For the brominated species, the following surface mixing ratios of stratospheric-relevant source gases are assumed: $[CH_3Br] = 6.9$ ppt, [halons]= 7.99 ppt, $[CHBr_3] = 1$ ppt, $[CH_2Br_2] = 1$ ppt, and $\Sigma$ $[CHClBr_2, CHCl_2Br, CH_2ClBr, ....]$

= 1 ppt of Br. Organic bromine is thus $[Br_y^{org}] = 20.89$ ppt at the surface, in agreement with recent reports (e.g., WMO (2014), Sala et al. (2014)). No other (cf., unknown organic or inorganic) sources of bromine for UT, LS, and TTL are assumed (e.g., Fitzenberger et al. (2000), Salawitch et al. (2010), Wang et al. (2015)). Omitting the release and heterogeneous processing of bromine from sea-salt aerosols (e.g., Saiz-Lopez et al. (2004)) in the model for the sake of saving computing time appears justified since (1) even though it is predicted to be relevant for bromine ($\sim$30% of the total $Br_y^{inorg}$) in the free troposphere

(Schmidt et al. (2016)), its contribution to $BrO$ in the TTL is at most of the order of the accuracy ($\sim$0.5 ppt) of our $BrO$ measurements, (2) its time and space dependent sources (as for the brominated VSLS) are not well constrained, (3) in the modelled troposphere inorganic bromine only serve as boundary condition for bromine in the TTL, and (4) the additional $BrO$ would not affect the $BrO$ measurements-based calculation of $Br_y^{inorg}$ for the TTL (see below). Further the surface concentration of $CH_4$

is specified based on observations of AGAGE (https://agage.mit.edu/) and NOAA, which reflect recent variations in its growth rate.

The standard model run (#583) is initialized in 1979 and spun-up for 34 years at low horizontal resolution ($5.6° \times 5.6°$) and with 36 unevenly spaced sigma-pressure vertical levels in the altitude range 0 - 63 km. Output from January 1, 2013 is interpolated to a high horizontal resolution ($1.2° \times 1.2°$) and the simulation continued over the ATTREX campaign period using this resolution. The model output is sampled on-line along the Global Hawk flight tracks for direct comparison with the observations. Two further high resolution sensitivity experiments are performed from January 1, 2013 onwards. In run #584, the ratio of the photolysis frequency of $BrONO_2$ and the three-body association rate reaction coefficient $k_{BrO+NO_2}$ is increased by a factor 1.75 (e.g., Kreycy et al. (2013)). In run #585 the second-order rate reaction coefficient $k_{Br+O_3}$ is set to the upper limit of its uncertainty range (Sander et al. (2011)).

For all model levels and for the time resolution ($\sim$30 s) of the mini-DOAS measurements, 'curtains' of the targeted gases along the flight track are stored (e.g., see Fig. 6 in Stutz et al. (2016) and Fig. 2). They are imported into the RT model McArtim for further forward simulations of the observations, and measurement versus model inter-comparison studies. The inclusion of simulated TOMCAT/SLIMCAT curtains in our study is particularly necessary for (a) the retrieval of absolute concentrations using the $O_3$-scaling technique (see Stutz et al. (2016), section 4.3), (b) estimate of errors and retrieval sensitivities to various parameters (see section 4.4 and the supplement to Stutz et al. (2016)), (c) the separation of dynamical and photochemical processes in the interpretation of our data, (d) sensitivity tests for the assumed kinetic data, and (e) the assessment of total $Br_y^{inorg}$ (see section 4).

Finally, details of how the loss in ozone is calculated is provided in appendix A.

## 3 Measurements and data reduction

Within the framework of the NASA-ATTREX project, the Global Hawk performed 6 flights into the subtropical LS, UT, and TTL over the Eastern Pacific in early 2013 (Fig. 1) and another 9 flights over the Western Pacific in early 2014. The present paper reports on the 2013 flights, since the 2014 flights were mostly performed into the cold TTL, where cirrus clouds mostly prevailed at flight level. Evidently due to the multiple scattering of light by the cirrus cloud particles, the interpretation of our UV/vis limb measurements is not straightforward. Accordingly the data collected in 2014 will be reported elsewhere. Details of the NASA-ATTREX 2013 instrument package, the flights, their details as well as some results on the collected data can be found in the articles of Jensen et al. (2013, 2015), as well as on the project's website https://espo.nasa.gov/missions/attrex/content/ATTREX .

In February and March 2013, the NASA-ATTREX flights of the Global Hawk were strongly biased with respect to the sampled air masses, mostly because the scientific interest was primarily put on probing the TTL over the Eastern Pacific for aerosols and cirrus cloud particles during the convective season, rather than for the photochemistry of bromine in the LS, UT, and TTL (see Fig. 1). Therefore, and due to operational reasons typical flight patterns extended from Dryden/California into southern or south-western direction during daytime until a turn-point was reached and the back leg to Dryden in northeastern

direction occurred during the night, when the mini-DOAS instrument could not take measurements. The dives were mostly performed within the TTL, and occasionally within the subtropical lowermost stratosphere during the return legs at night but not during the out-going daytime legs. Finally the landings at Dryden were scheduled for the early local morning, mostly due to operational constraints. Therefore, no profiles of the targeted species could be obtained in the subtropical lowermost

stratosphere at daytime, but a large number within the UT and TTL.

    Furthermore, the latitudinal definition of the notations 'subtropical' LS and 'tropical' TTL need some clarification. According to the definition of Fueglistaler et al. (2009), the latitudinal boundary between the subtropics and tropics should be where the subtropical jet is located. However, since we do not infer dynamical parameters (such as the potential vorticity) from our data, we conveniently define the boundary according to proxies for (a) different air mass ages, i.e. $[CH_4]$ concentrations

$\leq 1790$ ppb are labeled 'subtropical' and $[CH_4] \geq 1790$ ppb are labeled 'tropical', and (b) photochemical regimes, i.e. $[O_3]$ (subtropical when $[O_3] \geq 150$ ppb, and TTL when $[O_3] \leq 150$ ppb), which we find suitable from a visual inspection of our data (see below).

    As mentioned above and outlined in detail in the study of Stutz et al. (2016), the processing of the mini-DOAS data included (a) spectral retrieval of the targeted gases from the mini-DOAS measurements (section 2.1), (b) forward modeling of the RT

for each measured spectrum (section 2.6), and either applying optimal estimation or the novel x-gas scaling technique (see sections 4.1 and 4.2 in Stutz et al. (2016)). Comprehensive sensitivity simulations indicated that optical estimation based on constraints inferred from measured $O_4$ and/or relative radiance would not result in the desired error range (Stutz et al. (2016), section 4.2). Therefore we decided to apply the x-gas scaling technique (Stutz et al. (2016) and Raecke (2013)) with x being ozone measured in-situ by the NOAA-2 $O_3$ photometer (see section 2.2).

The $O_3$-scaling technique makes use of the in-situ $O_3$ measured by the NOAA instrument (Gao et al., 2012) and the limb measured $O_3$ total slant column amounts ($SCD_{O3}$) either monitored in the UV (for the retrieval of $BrO$ in the 343 - 355 nm wavelength band) or visible wavelength range (for the retrieval of $NO_2$ in the 424 - 460 nm wavelength band) (see equation 12 in Stutz et al. (2016)). Here the ratio of the measured slant column and in-situ measured $SCD_{O3}/[O_3]$ can be regarded as a proxy for the (horizontal) light path length over which the absorption is collected. In fact in the paper of Stutz et al. (2016), it

is argued that the so-called $\alpha$ factors account for the fraction of the absorption of the scaling gas x (e.g., x = $O_3$ in our study) picked-up on the horizontal light paths ahead of the aircraft relative to the total measured absorption. The sensitivity study on the $\alpha$ factors presented in Stutz et al. (2016) (e.g., in the supplement) indicates that for the targeted gases uncertainties in $\alpha$ factor ratios due to assumptions regarding the RT (for example due to Mie scattering by aerosols and clouds) mostly cancel out, while uncertainties in the individual profile shapes of the targeted and scaling gas are most relevant for the errors of the

inferred gas concentrations. Therefore in the present study, profile shapes of the targeted and scaling gas predicted by the TOMCAT/SLIMCAT CTM are used in the RT calculations, aiming at the calculation of the $\alpha$ factors. The uncertainties in the profile shapes (assumed to be of the order of the altitude adjustment of the $CH_4$ and $O_3$ curtains, which are typically much smaller than the altitude grid spacing in the SLIMCAT/TOMCAT simulations) are then carried over to calculate the overall errors, as discussed in section 4.4 of the Stutz et al. (2016) study.

It should be noted that, for the flight on Feb. 21, 2013 (SF4-2013), the DOAS retrieval is much less robust than for all the other flights, most likely because the Fraunhofer reference spectra (taken via a diffuser) are affected by temporally changing residual structures likely due to ice deposits or some other residues on the entrance diffuser. Therefore the data of this flight are not analyzed in detail, but they are only reported for completeness here.

Finally, in our analysis only those data which are taken at a solar zenith angle (SZA) $\leq 88\,°$ are considered, because for increasing SZAs the received skylight radiance requires increasingly longer signal integration times (longer than the standard integration time which is 30 s), and are thus averaged over longer distances ahead of the aircraft. Moreover, as the SZA increases the skylight is expected to traverse an increasingly inhomogeneous curtain of the probed radicals (e.g., inspect Figs. 5, and 6 in Stutz et al. (2016)). As consequence, the spatial grid of TOMCAT/SLIMCAT ($1.2 \times 1.2\,°$) on which the photochem-
istry is simulated appeared too coarse for a useful interpretation of our measurements at large SZAs. Therefore for a tighter interpretation of our data, a model with higher spatial resolution than provided by TOMCAT/SLIMCAT would be required. Such an approach is for example followed in the balloon-borne studies of Harder et al. (2000), Butz et al. (2009), and Kreycy et al. (2013), and others. However, since both processes are likely to increase the error of our analysis, and since large SZA ($\geq 88^o$) measurements only constitute a minor part of all measurements, we refrain from this much more complicated approach.

**4    Results and Discussion**

In this section we first discuss how our mini-DOAS measurements of $O_3$, $NO_2$, and $BrO$, as well as of $CH_4$ (from UCATS and HUPCRS), and of the organic brominated source gases (from GWAS) compare with the model predictions of the TOMCAT/SLIMCAT model (sections 4.1 and 4.2). Then measured $BrO$ is compared with previous measurements in the UT/TTL/LS (section 4.3), and with the model predictions (section 4.4). Uncertainties and errors in the inferred $Br_y^{inorg}$ are assessed (section
4.5), before implications of our measurements for total $Br_y$ (section 4.6), and impacts of our measurements for TTL ozone are discussed (section 4.7).

**4.1    Comparison with TOMCAT/SLIMCAT predictions**

Figures 3 to 8 provide overviews on the measured data together with the TOMCAT/SLIMCAT modeled $Br_y^{inorg}$ partitioning (panels f) and inferred total $Br_y^{inorg}$ (panels g) as a function of universal time for each flight. The modeled values are obtained
by linear interpolation of the curtain data (see Fig. 2) to the exact altitude of the GH.

    The panels (b), and (c) of Figs. 3 to 8 show comparisons of measured and modeled $CH_4$, and $O_3$ mixing ratios. Here the measured and modeled species closely agree within the given error bars, after the modeled curtains are altitude-shifted (i.e., interpolated) by the same amount until measured and modelled $O_3$ agree (for details see Stutz et al. (2016)). Noteworthy is that in most cases the altitude adjustment is less than the grid spacing of TOMCAT/SLIMCAT (about 1 km in the TTL), thus
mostly accounting for the altitude mismatches of the actual cruise altitude of the Global Hawk and the model output rather than to deficits of the model to properly predict the vertical transport. The astonishingly good agreement achieved between measured and modeled $CH_4$, and $O_3$ lends confidence that the altitude-adjusted TOMCAT/SLIMCAT model fields reproduce

well the essential dynamical and photochemical processes of the probed air masses. The quality of the dynamical simulations are further tested by comparing modeled and measured $O_3$ as a function of $CH_4$ (Fig. 9). For all flights the agreement of the observed and modeled $O_3$ vs $CH_4$ correlation is reasonably good, except for flights SF1-2013 and SF2-2013, where the UCATS measured $CH_4$ scatters around the simulated $CH_4$ concentrations. This scatter is most likely due to precision errors of UCATS, rather than reflecting the real behavior of the atmosphere. Evidence for this conclusion is provided from the $CH_4$ comparisons for SF3-2013 to SF6-2013, in which the HUPCRS $CH_4$ data are taken; these data do not show such a scatter and compare reasonably well with the model predictions.

Panels (d) of Figs. 3 to 8 compare measured and modeled $NO_2$. Overall the measured (and modeled) $NO_2$ concentrations meet the expectations with respect to its partitioning and total $NO_x$ (= $NO + NO_2 + NO_3$) abundances in the LS, UT, and TTL over the pristine Pacific. Elevated $NO_2$ concentrations (range 70 to 170 ppt) are measured within the subtropical lowermost stratosphere, where aged air masses are probed, as indicated by depleted $CH_4$ concentrations and elevated $O_3$ concentrations (and presumably decreased $N_2O$ concentrations). Note that $N_2O$ is the primary source for stratospheric $NO_x$, and in the stratosphere $CH_4$ and $N_2O$ destruction closely follow each other (e.g., Michelsen et al. (1998), Ravishankara et al. (2009)). Very low $NO_2$ concentrations ($\leq 30$ ppt) are detected within the UT and TTL, indicating that the analyzed air does not originate from recently polluted, or lightning-affected regions. Further, the modeled $NO_2$ concentrations (red line in panel d) are found to fall into the given range of errors in the measured $NO_2$ concentrations. This finding strongly indicates that, the $NO_x$ and $NO_y$ (= $NO_x$, $N_2O_5$, $HONO_3$, $HO_2NO_2$,...) budget and photochemistry of the LS, UT, and TTL are reproduced well in the TOMCAT/SLIMCAT simulations, and that overall the $O_3$-scaling technique works well for $NO_2$.

Panels (e) in Figures 3 to 8 compares measured and modelled BrO. Again measured and modelled BrO mixing ratios reasonably compare for most flight sections, but sizable discrepancies are also discernible for some flight sections. Possible reasons for latter are discussed in following, which may be due to deficits in the model's assumption regarding the sources of bromine (see section 4.2), and/or deficits in the adopted photochemistry (see section 4.4).

## 4.2 Comparison of measured and model organic bromine

Before measured and modeled BrO can be compared quantitatively, it is necessary to compare the measured amounts of different brominated source gases with the model predictions (Fig. 10). For the assumed (constant) surface mixing ratios (see subsection 2.7), measured and modeled $CH_3Br$ (upper left panel), $CHBr_3$ (upper right panel), and for all other Halons, for example H1211 (lower right panel), compare well, even if the data is scattered from flight to flight. For $CH_2Br_2$, however, TOMCAT/SLIMCAT run # 583 underpredicts the observed mixing ratio for high concentrations (by 0.1 ppt) and overpredicts it by up to 0.2 ppt for low concentrations (lower left panel). This is most likely due to an assumed too low surface concentration (1 ppt), variable mixing ratios at the surface not correctly considered in the model, and/or errors the atmospheric lifetime by reactions of $CH_2Br_2$ with OH radicals in the model (e.g. Mellouki et al. (1992), Ko et al. (2013), WMO (2014)).

The flight-to-flight and sample-to-sample scatter in $CH_3Br$, and $CHBr_3$ is mostly due to different source regions of the air masses probed during SF1-2013 to SF6-2013. This implies a spatially (and possibly time-dependent) varying source strength of the brominated natural source gases (e.g. Hossaini et al. (2013), Ziska et al. (2013)). In the present version of the TOM-

CAT/SLIMCAT simulations, this scatter introduces an estimated uncertainty of $\pm$ 0.8 ppt into $Br_y^{org}$, and potentially in the inferred $Br_y^{inorg}$ available in the TTL. The systematic under-prediction of 0.1 ppt at high $CH_2Br_2$ concentrations, and its too long lifetime in the TTL leading to too large $CH_2Br_2$ concentrations in the model for old air (by up to 0.2 ppt) may cause an additional and systematic under-prediction of $Br_y^{inorg}$ of up to $\leq$ 0.4 ppt in the model. Both contributions to the uncertainty in the $Br_y^{org}$ are considered when comparing measured and modeled BrO, and $Br_y^{inorg}$ (see below).

## 4.3 Comparisons of measured BrO with previous studies

Next, we compare our data with previous BrO measurements in the UT and TTL, i.e. the balloon measurements of Dorf et al. (2008), and the aircraft measurements of Wang et al. (2015) and Volkamer et al. (2015) during the TORERO campaign.

Overall the balloon-borne BrO profile measurements of Dorf et al. (2008) performed over tropical Brazil during the dry (i.e. the non-convective season) in June 2005 and June 2008 compare well with the BrO profiles inferred from our measurements for the UT and TTL (i.e. typically [BrO] = 0.5 - 1.0 ppt in the upper UT and base of the TTL, and up to 5 ppt at the cold point tropopause (e.g., compare Fig. 1 in Dorf et al. (2008) with Fig. 11).

The present study and the BrO profile measurement of Dorf et al. (2008) do, however, not confirm the recently reported presence of BrO amounting up to 3 ppt in the tropical and subtropical UT, and around the bottom of the TTL (14 km) (Wang et al., 2015) (compare Fig. 2, panel A in Wang et al. (2015) with panel (c)in Fig. 11). Sensitivity studies using the BrO profile of Wang et al. (2015) as the a priori of an optimal estimation concentration retrieval for the ATTREX measurements results in a kink of BrO around 12 km (Fig. 11). This behavior can be explained with the disagreement between the observed profiles above 13 km and the insensitivity of the ATTREX observation to BrO below this altitude.

While the geographical location of the observations by Wang et al. (2015) and those of the GH did not overlap, the ATTREX flights covered a wide geographic area over which we do not find indications of unexpected high or elevated BrO concentrations in the UT, and TTL, either from inspecting the UT from above (e.g., see Fig. 15 in Stutz et al. (2016)), nor when directly probing the TTL (see Figs. 11 to 8).

Several similarities and differences exist between the TORERO measurements reported by Wang et al. (2015) and Volkamer et al. (2015) and our study. Using NSF/NCAR G-V HAIPER, Wang et al. (2015) probed the UT and the bottom of the TTL (up to about 14 km) for BrO over an adjacent part of the Pacific, i.e. mostly off the western coasts of South and Central America notably during the same season, but in area more to the south than probed during the present study.

It is possible that the TORERO observations Wang et al. (2015) and Volkamer et al. (2015) off the western coasts of South and Central America, i.e. further south than the ATTREX region but during the same season, encountered an unusual meteorological situation that would have caused downward transport of bromine rich air from the lower stratosphere to the UT and the bottom of the TTL (up to about 14 km), or that sea salt released bromine played a role (e.g., Schmidt et al. (2016)).

However, our study has identified possible problems when using optimal estimation technique with constraints based for example on measured $O_2$-$O_2$ for high altitude aircraft limb observations. The RT below the aircraft and in particular in the lower troposphere plays a crucial role for the observations, due to the much higher $O_2$-$O_2$ concentrations. Also since individual limb measurements already cover an area of typical 200 x 20 km in front of the aircraft (see Figure 5 in Stutz et al. (2016)),

and even more crucial when applying optimal estimation for profile inversion a series of measurements taken during the ascent and descent of the GH are jointly inverted. Hence the radiative field and its time dependence needs to be known over a larger food-print (i.e., the RT is 2-D, or even 3-D plus its time dependence over the period of single profile measurement).

We did not encounter conditions without (marine stratus cumulus) clouds in this footprint during any of the ATTREX flights.
Therefore, any skylight analyzed for the $O_2$-$O_2$ absorption in the limb direction may carry additional, or even substantial, information on the radiative transfer of lower atmospheric layers (see Figure 7 in Stutz et al. (2016)), rather than of the targeted atmospheric layers. We acknowledge that Wang et al. (2015) and Volkamer et al. (2015) selected 'cloud-free' conditions at the location of their profile measurement, but the cloudiness in the large area ahead of their aircraft is less clear.

Another challenge we encountered was that of the overhead BrO column, which can substantially contribute to the limb
BrO signal. The large concentrations of BrO in the stratosphere at daytime, and its potential column changes mostly due to a changing tropopause height or intrusion of tropospheric air (e.g., at the subtropical or polar jet) may thus mimic the presence of BrO in limb the direction, or at flight altitude (e.g., Wang et al. (2015), and Volkamer et al. (2015) and Fig. 14 in Stutz et al. (2016)). We solved this problem by using a highly resolved stratospheric CTM to study the potential influence of changing overhead BrO concentrations on our results.

In conclusion, our sensitivity studies have shown potential problem with the $O_2$-$O_2$ constrained RT calculations used to retrieve vertical BrO profiles, as well as the need to accurately determine the stratospheric BrO column. With this in mind and the disagreement between our UTLS BrO profiles and TORERO flights 12 and 17 Wang et al. (2015) and Volkamer et al. (2015), it is clear that future work is needed in reconciling the observations as well as the different retrieval approaches.

### 4.4  Comparison of measured and modeled BrO

Measured and modeled BrO are displayed in Figs. 3 to 8 (panel e), together with the modeled $Br_y^{inorg}$ partitioning (panel f) and inferred $Br_y^{inorg}$ (panel g). Elevated BrO concentrations are measured within the LS (range 3 - 9 ppt), and lower BrO concentrations in the TTL (range 0.5 - 5 ppt), with the smallest BrO concentrations (0.5 - 1 ppt) occurring near the bottom of the TTL. Overall this behavior is expected from arguments based on the amount and composition of the brominated organic and inorganic source gases, their lifetimes, atmospheric transport, and photochemistry (e.g., Fueglistaler et al. (2009),
Aschmann et al. (2009), Hossaini et al. (2012b), Ashfold et al. (2012), WMO (2014), Fernandez et al. (2014), and Saiz-Lopez and Fernandez (2016)). In particular, for our daytime measurements it is observed that (a) BrO increases with $O_3$ and available $Br_y^{inorg}$ and thus altitude, (b) the predicted BrO/$Br_y^{inorg}$ ratio decreases towards the bottom of the TTL, where (c) HBr and/or Br atoms may become comparable to BrO, but HOBr does not play a major role in the $Br_y^{inorg}$ partitioning. While observation (a) is due to the increased destruction of primarily the short-lived $Br_y^{org}$ species and the efficient reaction of the released Br
atoms with increasing altitude and increasing ozone concentrations, observations (b) and (c) are due to reactions of the Br atoms with $CH_2O$, (and less $H_2O_2$) into HBr which is recycled back by reactions with OH and by variable amounts heterogeneously (depending on the available surface of aerosols and cloud particles) to Br atoms, as predicted by Fernandez et al. (2014), and Saiz-Lopez and Fernandez (2016). Noteworthy is also the predicted minor role of HOBr eventually formed by reactions of OH radicals with heterogeneously produced $Br_2$, or by the reaction $HO_2 + BrO$ and photolytic destruction of HOBr in the

TTL. While the rate of the former reaction is anyway small due to short photolytic life-time of $Br_2$, the rate of latter reaction is small due to the small OH concentration in the TTL as compared to photolysis of HOBr at daytime.

Fig. 12 compares measured and modeled BrO. For the majority of all flights (except flight SF4-2014, for which a DOAS retrieval problem exists which causes a constant bias of about 2 ppt in inferred BrO), measured and modeled BrO closely compare for low concentrations (i.e. close to bottom to the TTL), or comparable younger air based on measured $CH_4$. For larger BrO concentrations (and older air) good agreement between the measurement and model is found for SF1-2013, SF5-2013, and SF6-2013, when mostly air of low $NO_2$ concentrations (and predicted low $BrONO_2$ concentrations) is probed. For large BrO concentrations as encountered during flights SF2-2013, and SF3-2013, the measured BrO is up to 2 ppt, or 25% larger than what the model predicts. This gap could partly be closed by adjusting the $CH_2Br_2$ surface concentration and atmospheric lifetime, or by considering a detailed scheme for dehalogenation of sea salt, i.e. bromine activation (e.g. Saiz-Lopez et al. (2004), Fernandez et al. (2014), Schmidt et al. (2016)). Adjusting $CH_2Br_2$ would add 0.4 ppt of $Br_y^{inorg}$, or $\sim$ 0.3 ppt to BrO, thus removing the flight-to-flight scatter in source gas concentrations ($\pm$ 0.8 ppt) in $Br_y^{inorg}$. This could for example be done by a detailed back trajectory and source appointment analysis to which a forthcoming study will be devoted. Likewise, dehalogenation of sea salt could add another 0.5 ppt to BrO (or about 0.7 ppt of $Br_y^{inorg}$) in the upper TTL (e.g., Saiz-Lopez et al. (2004), Fernandez et al. (2014), Schmidt et al. (2016)).

### 4.5  Uncertainties in estimating the inorganic bromine partitioning

Another reason for the gap in measured and modeled BrO may come from uncertainties in the used kinetic constants and how they affect the $Br_y^{inorg}$ (= Br + 2 · $Br_2$ + BrO + $BrONO_2$ + HOBr + HBr + BrCl) partitioning. Our photochemical modeling, aimed at reproducing measured $O_3$, $NO_2$, and BrO (see the panels (f) in Fig. 3 to Fig. 8), indicates that at daytime HOBr and HBr contribute less than 10% to $Br_y^{inorg}$. Therefore, we concentrate on the photochemical model errors due to the partitioning primarily among BrO, Br, and $BrONO_2$. In this context, most important are the reactions BrO + $NO_2$ + M $\rightarrow$ $BrONO_2$ + M followed by the photolysis of $BrONO_2$, and the reaction Br + $O_3$ $\rightarrow$ BrO + $O_2$.

How uncertainties of the photolytic destruction (J) and three-body formation reaction (k) (together referred to as J/k) of $BrONO_2$ propagate into BrO is tested in model run #584. Here, according to the finding of Kreycy et al. (2013) J/k was increased by a factor 1.7 (+0.4/-0.2) as compared to the JPL recommendation (Sander et al., 2011) (see the blue crosses in Fig. 12). Evidently increasing J/k helps to close the remaining gap in measured versus modeled BrO, which becomes particularly relevant to reproduce BrO when $NO_2$ is large, i.e. in the subtropical LS.

Furthermore, Sander et al. (2011) estimate the uncertainty in the reaction rate coefficient $k_{Br+O3}$ at low temperature (T = 190 K) to be $\pm$ 40% (see comment G31). When only considering the two studies which actually measured rather than extrapolated the reaction rate coefficient into the relevant temperature range (T = 190 - 200 K), smaller uncertainty (28%) is indicated (Michael et al. (1978) and Nicovich et al. (1990)). Therefore, in the following an uncertainty of 28% for $k_{Br+O3}$ is assumed. Overall, increasing $k_{Br+O3}$ (model run #585) to the upper limit possible according to the JPL compilation (i.e. by factor of 1.28) changes the measured vs modeled correlation for BrO very little (see the red crosses in Fig. 12). It does change, however, the $Br_y^{inorg}$ partitioning so that [BrO] is always largely prevalent over [Br] even at the lowest altitudes of

the TTL (e.g., see panel (f) in Fig. 3 to 8). Our joint measurement of $O_3$, $NO_2$, and BrO and the supporting CTM simulations thus indicate $[Br]/[BrO] < 1$ for all probed regimes. Our finding is therefore in contrast to the simulations of Fernandez et al. (2014), and Saiz-Lopez and Fernandez (2016) who suggest that $[Br]/[BrO]$ may become larger than unity in the tropical UT and TTL at daytime. This conclusion is due mostly to the larger measured $O_3$ concentrations than those modeled in the study of Fernandez et al. (2014) and Saiz-Lopez and Fernandez (2016), and the conclusion is irrespective of what (within the given error bars) is assumed for $k_{Br+O3}$.

Gaussian addition of all uncertainties and errors (i.e. the errors of the retrieved BrO concentrations (in section 4.4 of Stutz et al. (2016)), the cross section error, and the uncertainty in the modeled $[Br]/[BrO]$, and $[BrONO_2]/[BrO]$ ratios), leads to the $Br_y^{inorg}$ error, as indicated in the panel (f) of Figs. 3 to 8.

## 4.6  Inferred total $Br_y^{inorg}$

Finally, we discuss the inferred $Br_y^{inorg}$ (contribution 4) as function of potential temperature in the LS, UT, and TTL over the Eastern Pacific during the 2013 convective season (Figure 13). Here we discriminate between young air $[CH_4] \geq 1790$ ppb, mostly found within the tropical UT and TTL (Fig. 13, left panel), and older air $[CH_4] \leq 1790$ ppb (Fig. 13, right panel) mostly found in the subtropical lowermost stratosphere. The different histograms in Fig. 13 clearly indicate that $Br_y^{inorg}$ increases with increasing potential temperature, i.e. from $2.63 \pm 1.04$ ppt at $\theta = 350$ - $360$ K (at the bottom of the TTL) to $4.22 \pm 1.37$ ppt for $\theta = 390$ - $400$ K (just above the cold point tropopause). The inferred $Br_y^{inorg}$ thus brackets well the modelled $[Br_y^{inorg}] = 3.02 \pm 1.90$ ppt predicted to exist at 17 km in the TTL (Navarro et al., 2015).

The increase in $Br_y^{inorg}$ with increasing potential temperature $\theta$ and decreasing $CH_4$ concentration thus reflects the decrease in concentrations of brominated VSLS (contribution 3). The correspondence of decreasing $Br_y^{org}$, and increasing $Br_y^{inorg}$ concentrations is also found on a sample-to-sample as well as on a flight-to-flight basis. This correspondence keeps $[Br_y]$ almost constant within the TTL during an individual flight, but $[Br_y]$ varies from flight to flight in a range of $[Br_y] = 20.3$ ppt to $22.3$ ppt (Figure 14).

Moreover, it appears that the increase in $Br_y^{inorg}$ with $\theta$ mostly corresponds to a decrease in concentrations of the brominated VSLS, if only the same (young) air masses of large $CH_4$ concentrations are probed (Figure 15). For example for SF1-2013, SF5-2013, and SF6-2013 when mostly air masses of the TTL are probed, all data points fall into a band of about $\pm 1$ ppt in width, next to a flight-dependent diagonal line (not shown), but not for SF3-2013 when air masses of the LS (and thus older air) and TTL are probed. When extrapolating the data points along lines of constant $[VSLS]+[Br_y^{inorg}]$ bromine (grey dashed lines in Fig. 15) for SF1-2013, SF5-2013, and SF6-2013 to $[Br_y^{inorg}] = 0$, and assuming no bromine is effectively lost in the troposphere, then the apparent concentrations of brominated VSLS at the surface should range between 4 - 8.5 ppt. However, frequently larger concentrations of brominated VSLS (some 10 ppt) are measured in the boundary layer of the Pacific (e.g., Yokouchi et al. (1997), Schauffler et al. (1998), Wamsley et al. (1998) Yokouchi et al. (2005), Tegtmeier et al. (2012), Ashfold et al. (2012), Ziska et al. (2013), Sala et al. (2014)). Further bromine released from sea-salt also contributes to $Br_y^{inorg}$ in the marine boundary layer and may reach in variable amounts the bottom of the TTL (e.g., Saiz-Lopez et al. (2004), Fernandez

et al. (2014), Schmidt et al. (2016)). Therefore effective loss processes for inorganic bromine, for example by heterogeneous uptake of inorganic bromine on aerosol and cloud particles, must act in the atmosphere (e.g. Schmidt et al. (2016)).

Next, when subtracting from the given range (20.3 ppt to 22.3 ppt) of total $Br_y$, the almost constant contribution of $CH_3Br$ and the Halons to total stratospheric bromine (14.6 ppt in 2013), a variable contribution from VSLS bromine (contribution 3), and $Br_y^{inorg}$ (contribution 4) to total TTL bromine in the range of 5.7 ppt to 7.7 ppt ($\pm$ 1.5 ppt) is calculated (Figure 15). We note that this range falls well into the range assessed in WMO (2014), or recently estimated by Navarro et al. (2015) (6 ppt; range = 4 - 9 ppt) for contribution of 3 and 4 to the total stratospheric $Br_y$. It is, however, somewhat (up to 2 ppt) larger than indicated in some earlier work, including our balloon-borne studies (for details see section 1).

Here one may wonder whether (a) this result is significant, or that (b) some $Br_y^{inorg}$ is actually removed by heterogeneous processes in the TTL (e.g., Aschmann et al. (2011), Aschmann and Sinnhuber (2013)), or (c) that, TTL $Br_y$ shows some seasonality analogous to the tape recorder for $H_2O$ (e.g., Levine et al. (2008), Krüger et al. (2008), Fueglistaler et al. (2009), Schofield et al. (2011), Ploeger et al. (2011)).

Also remarkable are the non-negligible amounts of $Br_y^{inorg}$ (2.63 $\pm$ 1.04) ppt, range from 0.5 ppt to 5.25 ppt, which is from close to zero to 25% of all TTL bromine) inferred for altitudes at the bottom of the TTL ($\theta$ = 350 - 360 K), of which 40 to 50% may consist of BrO. This finding clearly sets a range and an upper limit for the $Br_y^{inorg}$ influx into the TTL due to entrained air masses of recent tropospheric origin (contribution 4). Again, the latter can most likely be attributed to different source regions (and thus emission strengths) of the brominated VSLS and bromine released from sea salt, and a varying degree of photochemical and heterogeneous processing of the air masses transported from the surface to the TTL. The increase in variance found for $Br_y^{inorg}$, which increases in absolute terms, but decreases in relative terms (i.e. from 0.4 for $\theta$ in the range 350 K to 360 K to 0.3 for $\theta$ = 390 K to 400 K) with increasing $\theta$ is also noteworthy. This may indicate a subsequent flattening-out of the air-mass-to-air-mass variability of $Br_y^{inorg}$ in aging air due to the photochemical decay of the brominated organic source gases and atmospheric mixing processes.

## 4.7 Implications for ozone

The ozone budget in the TOMCAT/SLIMCAT simulation has been analysed based on the rate-limiting steps of the catalytic ozone destruction cycles, according to the concept of Johnston and Podolske (1978). The chemical rates are averaged over the Eastern Pacific region (20°S - 20 °N, 170°W - 90°W) for the duration of the campaign. Within this domain, the net rate of ozone change varied from a loss of - 0.3 $\mathrm{ppbv/day}$ at the base of the TTL ($\theta$ = 355 K, p = 150 hPa) to a production of + 1.8 $\mathrm{ppbv/day}$ at the top ($\theta$= 383 K, p = 90 hPa). This increase of $O_3$ with height is due to the strong vertical gradient in the production rate of odd oxygen by $O_2$ photolysis. Within the catalytic ozone loss cycles in the TTL, the model indicates that those containing bromine contribute between 12% (base of TTL) and 22% (top of TTL) of the total (see appendix A). The dominant contribution to this is through the cycle involving $BrO + HO_2$ to form HOBr. Overall, the modeled ozone loss cycles which account for the majority of the destruction in this region are those with the rate-limiting steps of the reaction $HO_2 + O_3$ to form $OH + 2O_2$ and the reaction of $HO_2 + HO_2$ to form $H_2O_2$, i.e. cycles involving $HO_x$ species. Therefore, increases in

the bromine loading of the TTL caused by possible/expected increases have the potential to deplete ozone, in a region where ozone changes have the largest impact on radiative forcing (Riese et al., 2012).

Quantifying the radiative impact of the $O_3$ changes described above is the beyond the scope of this study. However, we can note that (i) recent work has highlighted the efficiency of brominated VSLS at influencing climate (through changed $O_3$), owing to their efficient breakdown in the UTLS (Hossaini et al., 2015), and (ii) a significant increase in $Br_y$ in this region (from VSLS or other sources) could be important for future climate forcing. The latter could conceivably occur given suggested climate-induced changes to (1) tropospheric transport (e.g., Hossaini et al. (2012b)), (2) changes in OH, affecting VSLS lifetimes (3), and/or an elevated bromine loading in the UT/LS and TTL to the expected increase in VSLS emissions from the rapidly growing aquaculture industry (WMO, 2014).

## 5  Conclusions

The subtropical lowermost stratosphere, upper troposphere, and tropopause layer of the Eastern Pacific are probed for inorganic bromine during the convective season (February and March 2013). The measurements of $CH_4$, $O_3$, $NO_2$, BrO, and some important organic brominated source gases are inter-compared with TOMCAT/SLIMCAT simulations. After the simulated TOMCAT/SLIMCAT curtains of $O_3$ are projected on the measured $O_3$ concentrations, measured and modeled $CH_4$ agree well. This agreement is not surprising, since $O_3$, and $CH_4$ are strongly anti-correlated (see Fig. 9). It thus provides evidence that the relevant dynamical processes are represented well in the TOMCAT/SLIMCAT simulations. When the simulated curtains of $NO_2$ are adjusted with the same parameters as inferred above, excellent agreement is again found between measured and modeled $NO_2$, thus providing further confidence in our measurement technique, in the modeled $NO_y$ photochemistry, and in our overall approach.

The measured and modeled TTL concentrations of $CH_2Br_2$ and $CHBr_3$ are found to compare reasonably to the surface concentrations and atmospheric lifetimes of both species adopted in the model ([$CHBr_3$] = 1.4 ppt, [$CH_2Br_2$] = 1 ppt at the surface). Further, the contribution to bromine in the LS, UT, and TTL by some other VSLS chloro-bromo-hydrocabrons ($\Sigma$ [$CHClBr_2, CHCl_2Br, CH_2ClBr, ....$]) is accounted for by assuming a constant surface concentration of 1 ppt in the model. From flight-to-flight total organic bromine inferred from these VSLS species is found to vary by $\pm$ 1 ppt in the TTL over the Eastern Pacific in early 2013, which clearly indicates different origins and possibly atmospheric processing of the investigated air masses.

The measured BrO concentrations range between 3 - 9 ppt in the subtropical LS. In the TTL they range between $0.5 \pm$ 0.5 ppt at the bottom of the TTL, and about 5 ppt at $\theta$ = 400 K, in overall good agreement with the model simulations, and the expectation based on the decay of the brominated source gases, and atmospheric transport. In the TTL, the inferred $Br_y^{inorg}$ is found to increase from a mean of $2.63 \pm 1.04$ ppt for $\theta$ in the range of 350 - 360 K to $5.11 \pm 1.57$ ppt for $\theta$ = 390 - 400 K, respectively, whereas in the subtropical LS it reaches $7.66 \pm 2.95$ ppt for $\theta$s in the range of 390 - 400 K. Also remarkable is the non-negligible $Br_y^{inorg}$ found for the lowest altitudes of the TTL, i.e. $2.63 \pm 1.04$ ppt with a range from 0.5 ppt to 5.25 ppt

(or close to zero percent up to 25% of all TTL bromine). This may indicate a sizable, but rather variable influx of inorganic bromine into the TTL, largely depending on the air mass history, i.e. source region, and atmospheric transport and processing.

Our findings on LS and TTL $Br_y^{inorg}$ are in broad agreement with past experimental and theoretical studies on the processes and amount of bromine injected by source gas and product gases into the TTL, and eventually into the extra-tropical lowermost stratosphere (Ko et al. (1997), Schauffler et al. (1998), Wamsley et al. (1998), Dvortsov et al. (1999), Pfeilsticker et al. (2000), Montzka et al. (2003), Salawitch (2006), Sinnhuber and Folkins (2006), Hendrick et al. (2007), Laube et al. (2008), Dorf et al. (2006b), Dorf et al. (2008), Sinnhuber et al. (2009), Salawitch et al. (2010), Schofield et al. (2011), Aschmann et al. (2011), Hossaini et al. (2012b), Ashfold et al. (2012), Hossaini et al. (2012a), Aschmann and Sinnhuber (2013), Sala et al. (2014), Wang et al. (2015), Liang et al. (2014), WMO (2014), Navarro et al. (2015), and many others). Our study, however, sets tighter limits than those previously existing on the amount of $Br_y^{inorg}$ and $Br_y^{org}$, the influx of brominated source and product gases, and the photochemistry of bromine in the TTL and LS.

In particular, our study (re-)emphasizes that (a) variable amounts of VSLS bromine and (b) non-negligible amounts of $Br_y^{inorg}$ are also transported into the TTL. While process (a) may strongly depend on the source region and season (Hossaini et al., 2016), process (b) may depend on the efficiency of heterogeneous processing and removal of some $Br_y^{inorg}$ by atmospheric (ice) clouds and aerosols (e.g., Aschmann et al. (2011), and Aschmann and Sinnhuber (2013)). Therefore it is not surprising that TTL $Br_y$ is rather variable (i.e. 20.3 ppt to 22.3 ppt) in the studied season.

We also note that the amount of $Br_y$ over the Eastern Pacific during the convective season assessed here and in the study of Navarro et al. (2015) is somewhat (up to 2 ppt) larger than that presently found on average in the stratosphere (e.g., Dorf et al. (2006b), Hendrick et al. (2007), Dorf et al. (2008), and WMO (2014)). By assuming that this gap is significant, additional processes may come into the focus of stratospheric bromine research, i.e. the seasonality and possibly long-term trend of the bromine transported into the stratosphere (e.g., Levine et al. (2008), Krüger et al. (2008), Fueglistaler et al. (2009), Schofield et al. (2011), Ploeger et al. (2011)).

Conceivably adding some inorganic bromine (from contribution 4) to TTL bromine exerts an additional impact on ozone. For the Eastern Pacific (170°W - 90°W) our model-based assessment indicate a net loss of ozone of - 0.3 ppbv/day at the base of the TTL ($\theta$ = 355 K) and a net production of + 1.8 ppbv/day in the upper part ($\theta$ = 383 K). Within the catalytic ozone loss cycles in the TTL (see appendix A), the model indicates that those containing bromine contribute between 12% (at the base of the TTL) and 22% (at the top of TTL) of the total.

## Appendix A: Ozone loss calculation

The TOMCAT/SLIMCAT CTM contains a detailed description of stratospheric ozone chemistry, including radical and reservoir species in the odd oxygen, odd hydrogen, odd nitrogen, chlorine and bromine families. The model solves for the full 'odd oxygen' ($O_x = O_3 + O(^3P) + O(^1D)$) continuity equation based on the relevant reactions involving the $O_x$ species. However, in order to ascribe chemical ozone loss to particular catalytic cycles, it is useful to transform the odd oxygen continuity

equation, using steady-state assumptions, so that loss terms can be associated with the rate-determining steps of identifiable cycles. The methodology to achieve this was outlined by Johnston and Podolske (1978).

In this study the modelled chemical ozone (or $O_x$) changes are related to catalytic cycles using this methodology applied to the full chemical scheme and results are given in Section 4.7. In this appendix we describe this analysis for bromine (and some odd hydrogen) chemistry and show how the catalytic cycles are identified. An approach such as this is necessary in order to avoid overestimation ('double counting') or underestimation of ozone loss, which could occur if catalytic cycles are extracted in isolation from the rest of the model chemistry scheme.

**Table 1.** TOMCAT/SLIMCAT reactions involving $Br_y$ radical and reservoir species along with other reactions involving $HO_x$ species. The numbering scheme follows that used in the model. Here O means $O(^3P)$.

| | | | |
|---|---|---|---|
| 8 | $OH + O$ | $\rightarrow$ | $H + O_2$ |
| 9 | $O_2 + H + M$ | $\rightarrow$ | $HO_2 + M$ |
| 10 | $HO_2 + O$ | $\rightarrow$ | $OH + O_2$ |
| 11 | $OH + O_3$ | $\rightarrow$ | $HO_2 + O_2$ |
| 12 | $H + O_3$ | $\rightarrow$ | $OH + O_2$ |
| 14b | $OH + OH$ | $\rightarrow$ | $H_2O_2$ |
| 15a | $H + HO_2$ | $\rightarrow$ | $OH + OH$ |
| 15b | $H + HO_2$ | $\rightarrow$ | $H_2 + O_2$ |
| 24 | $HO_2 + HO_2$ | $\rightarrow$ | $H_2O_2 + O_2$ |
| 36 | $HO_2 + O_3$ | $\rightarrow$ | $OH + 2\ O_2$ |
| 120 | $Br + O_3$ | $\rightarrow$ | $BrO + O_2$ |
| 121 | $BrO + O$ | $\rightarrow$ | $Br + O_2$ |
| 123 | $BrO + NO$ | $\rightarrow$ | $Br + NO_2$ |
| 124 | $BrO + OH$ | $\rightarrow$ | $Br + HO_2$ |
| 125a | $BrO + ClO$ | $\rightarrow$ | $Br + OClO$ |
| 125b | $BrO + ClO$ | $\rightarrow$ | $Br + Cl + O_2$ |
| 125c | $BrO + ClO$ | $\rightarrow$ | $BrCl + O_2$ |
| 127 | $BrO + BrO$ | $\rightarrow$ | $Br + Br + O_2$ |
| 128 | $BrO + NO_2 + M$ | $\rightarrow$ | $BrONO_2 + M$ |
| 130 | $BrO + HO_2$ | $\rightarrow$ | $HOBr + O_2$ |
| 131 | $O + HOBr$ | $\rightarrow$ | $OH + BrO$ |
| 134 | $Br + CH_2O$ | $\rightarrow$ | $HBr + CHO$ |
| 135 | $Br + HO_2$ | $\rightarrow$ | $HBr + O_2$ |
| 136 | $HBr + OH$ | $\rightarrow$ | $Br + H_2O$ |
| 137 | $HBr + O(^1D)$ | $\rightarrow$ | $Br + OH$ |
| 139 | $HBr + O$ | $\rightarrow$ | $Br + OH$ |
| 145 | $BrONO_2 + O$ | $\rightarrow$ | $BrO + NO_3$ |
| 146 | $BrONO_2 + Br$ | $\rightarrow$ | $Br + Br + NO_3$ |
| 147 | $HBr + Cl$ | $\rightarrow$ | $Br + HCl$ |

| | | | |
|---|---|---|---|
| $J_{BrO}$ | $BrO + h\nu$ | $\rightarrow$ | $Br + O$ |
| $J_{BrONO2}$ | $BrONO_2 + h\nu$ | $\rightarrow$ | $Br + NO_3$ |
| $J_{BrCl}$ | $BrCl + h\nu$ | $\rightarrow$ | $Br + Cl$ |
| $J_{HOBr}$ | $HOBr + h\nu$ | $\rightarrow$ | $Br + OH$ |
| $J_{H2O2}$ | $H_2O_2 + h\nu$ | $\rightarrow$ | $2OH$ |

Based on the subset of the TOMCAT/SLIMCAT reactions listed in Table 1, the odd oxygen continuity (equation A1) would be written as:

$$\frac{d[O_x]}{dt} = -k_8 \cdot [OH] \cdot [O] - k_{10} \cdot [HO_2] \cdot [O] - k_{11} \cdot [OH] \cdot [O_3] - k_{12} \cdot [H] \cdot [O_3] - k_{36} \cdot [HO_2] \cdot [O_3] - k_{120} \cdot [Br] \cdot [O_3]$$
$$-k_{121} \cdot [BrO] \cdot [O] - k_{137} \cdot [HBr] \cdot [O(^1D)] - k_{139} \cdot [HBr] \cdot [O] - k_{145} \cdot [BrONO_2] \cdot [O] + J_{BrO} \cdot [BrO]$$

5 This equation can be modified by assuming that short-lived species are in steady state. Based on the above reaction scheme, we can put Br in steady state to get the following:

$$\frac{d[Br]}{dt} = -k_{120} \cdot [Br] \cdot [O_3] + k_{121} \cdot [BrO] \cdot [O] + k_{123} \cdot [BrO] \cdot [NO] + k_{124} \cdot [BrO] \cdot [OH] + k_{125a} \cdot [BrO] \cdot [ClO]$$
$$+ k_{125b} \cdot [BrO] \cdot [ClO] + k_{127} \cdot [BrO] \cdot [BrO] - k_{134} \cdot [Br] \cdot [CH_2O] - k_{135} \cdot [Br] \cdot [HO_2]$$
$$+ k_{136} \cdot [OH] \cdot [HBr] + k_{137} \cdot [O(^1D)] \cdot [HBr] + k_{139} \cdot [O] \cdot [HBr] + k_{146} \cdot [Br] \cdot [BrONO_2]$$

$$+ k_{147} \cdot [HBr] \cdot [Cl] + J_{BrO} \cdot [BrO] + J_{BrONO_2} \cdot [BrONO_2] + J_{BrCl} \cdot [BrCl] + J_{HOBr} \cdot [HOBr]$$
$$= 0$$

We can derive similar equations for many other short-lived species. Johnston and Podolske (1978) also discussed placing the rate of change of longer lived species to zero in order to help simplify the odd-oxygen continuity equation. In practice the magnitude of these terms for odd oxygen loss would be very small. Using 'steady state' expressions of Br, $BrONO_2$, HBr,

15 Cl, $NO_2$, OClO, OH, HCl, and $H_2O$ equation (A1) can be converted to:

$$\frac{d[Br]}{dt} = -k_{120} \cdot [Br] \cdot [O_3] + k_{121} \cdot [BrO] \cdot [O] + k_{123} \cdot [BrO] \cdot [NO] + k_{124} \cdot [BrO] \cdot [OH] + k_{125a} \cdot [BrO] \cdot [ClO]$$
$$+ k_{125b} \cdot [BrO] \cdot [ClO] + k_{127} \cdot [BrO] \cdot [BrO] - k_{134} \cdot [Br] \cdot [CH_2O] - k_{135} \cdot [Br] \cdot [HO_2]$$
$$+ k_{136} \cdot [OH] \cdot [HBr] + k_{137} \cdot [O^1D] \cdot [HBr] + k_{139} \cdot [O] \cdot [HBr] + k_{146} \cdot [Br] \cdot [BrONO_2]$$
$$+ k_{147} \cdot [HBr] \cdot [Cl] + J_{BrO} \cdot [BrO] + J_{BrONO_2} \cdot [BrONO_2] + J_{BrCl} \cdot [BrCl] + J_{HOBr} \cdot [HOBr]$$

$$= 0$$

This is part of the transformed odd-oxygen continuity equation as used in the diagnosis of the full TOMCAT/SLIMCAT chemistry scheme. For the chemical scheme given in Table 1, this equation can be further simplified by applying the steady state approximation to HOBr ($d[H_2O_2]/dt = 0$) and $H_2O_2$ ($d[HOBr]/dt = 0$) to get:

$$
\begin{aligned}
\frac{d[O_x]}{dt} =& -2 \cdot k_{10} \cdot [HO_2] \cdot [O] - 2 \cdot k_{12} \cdot [H] \cdot [O_3] - 2 \cdot k_{15a} \cdot [H] \cdot [HO_2] - 2 \cdot k_{24} \cdot [HO_2] \cdot [HO_2] \\
& -2 \cdot k_{36} \cdot [HO_2] \cdot [O_3] - 2 \cdot k_{121} \cdot [BrO] \cdot [O] - 2 \cdot k_{125b} \cdot [BrO] \cdot [ClO] - 2 \cdot k_{127} \cdot [BrO] \cdot [BrO] \\
& -2 \cdot k_{130} \cdot [BrO] \cdot [HO_2] - 2 \cdot k_{137} \cdot [HBr] \cdot [O(^1D)] - 2 \cdot k_{139} \cdot [HBr] \cdot [O] - 2 \cdot J_{BrCl} \cdot [BrCl]
\end{aligned}
$$

As discussed in Section 4.7, the dominant terms for ozone loss due to bromine chemistry are those involving the formation of HOBr ($2 \cdot k_{130} \cdot [BrO] \cdot [HO_2]$). Overall, the $HO_x$ cycle represented by $2 \cdot k_{36} \cdot [HO_2] \cdot [O_3]$ dominates the ozone loss with a minor contribution from reactions involving $H_2O_2$ production ($2 \cdot k_{24} \cdot [HO_2] \cdot [HO_2]$).

*Acknowledgements.* This study was funded through the NASA Upper Atmosphere Research Program (NASA ATTREX Grant numbers NNX10AO82A for HUPCRS, NNX10AO83A for GWAS, and NNX10AO80A for the mini-DOAS measurements). The NOAA ozone photometer and UCATS measurements were supported by the NASA ATTREX inter-agency agreement numbers NNA11AA54I and NNA11AA55I, respectively. Additional support for the mini-DOAS measurements came through the Deutsche ForschungsGemeinschaft, DFG (through grants PF-384 5-1/2, PF384 7-1/2 PF384 9-1/2, and PF384 12-1), and the EU project SHIVA (FP7-ENV-2007-1-226224). RuShan Gao, T. D. Thornberry, and D. W. Fahey were supported by the NOAA Atmospheric Composition and Climate Program, and the NASA Radiation Sciences Program. The TOMCAT/SLIMCAT modeling was supported by the NERC National Centre for Atmospheric Science (NCAS), UK and by the NERC TropHal project (NE/J02449X/1). MPC was supported by a Royal Society Wolfson Merit Award. We thank Dr. Eric Jensen (NASA Ames Research Center, Moffett Field, California) and his team for coordinating the NASA-ATTREX mission. We thank Joe McNorton for help with the NOAA and AGAGE $CH_4$ data. E. Atlas and M. Navarro gratefully acknowledge R. Lueb, R. Hendershot and S. Gabbard for technical support in the field, and X. Zhu and L. Pope for GWAS data analysis. Jim Elkins of NOAA would like to acknowledge the assistance of G. S. Dutton, J. D. Nance, and B. D. Hall during the ATTREX flights, calibration and integration. The authors are grateful for the comments given by two anonymous reviewers, and the comments of Barbara Dix and Rainer Volkamer (CU, Boulder, USA).

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

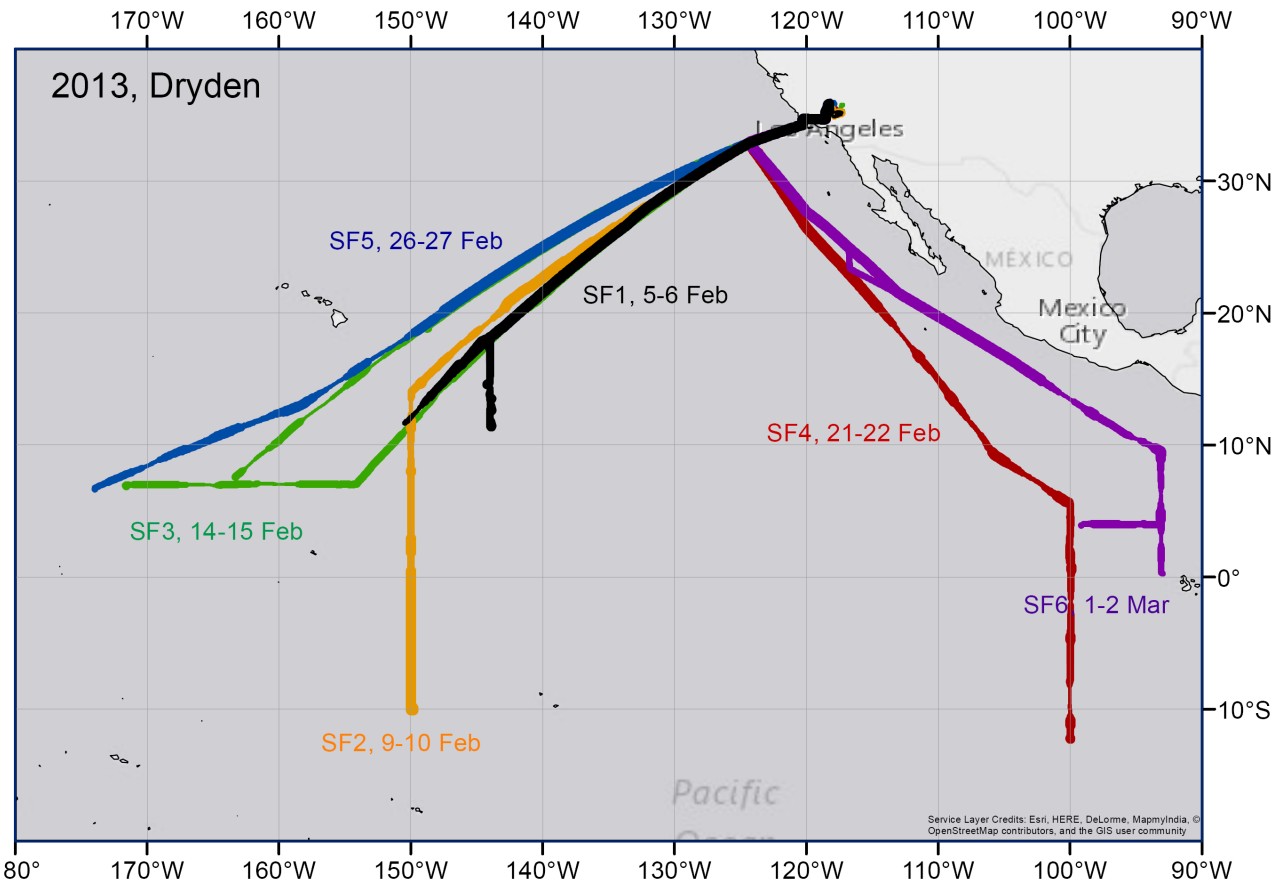

**Figure 1.** Overview of the NASA Global Hawk ATTREX flights conducted from Dryden in 2013. The thickness of the lines corresponds to flight altitudes, where the thinnest line is for an altitude of around 14 km and the thickest line for around 18 km.

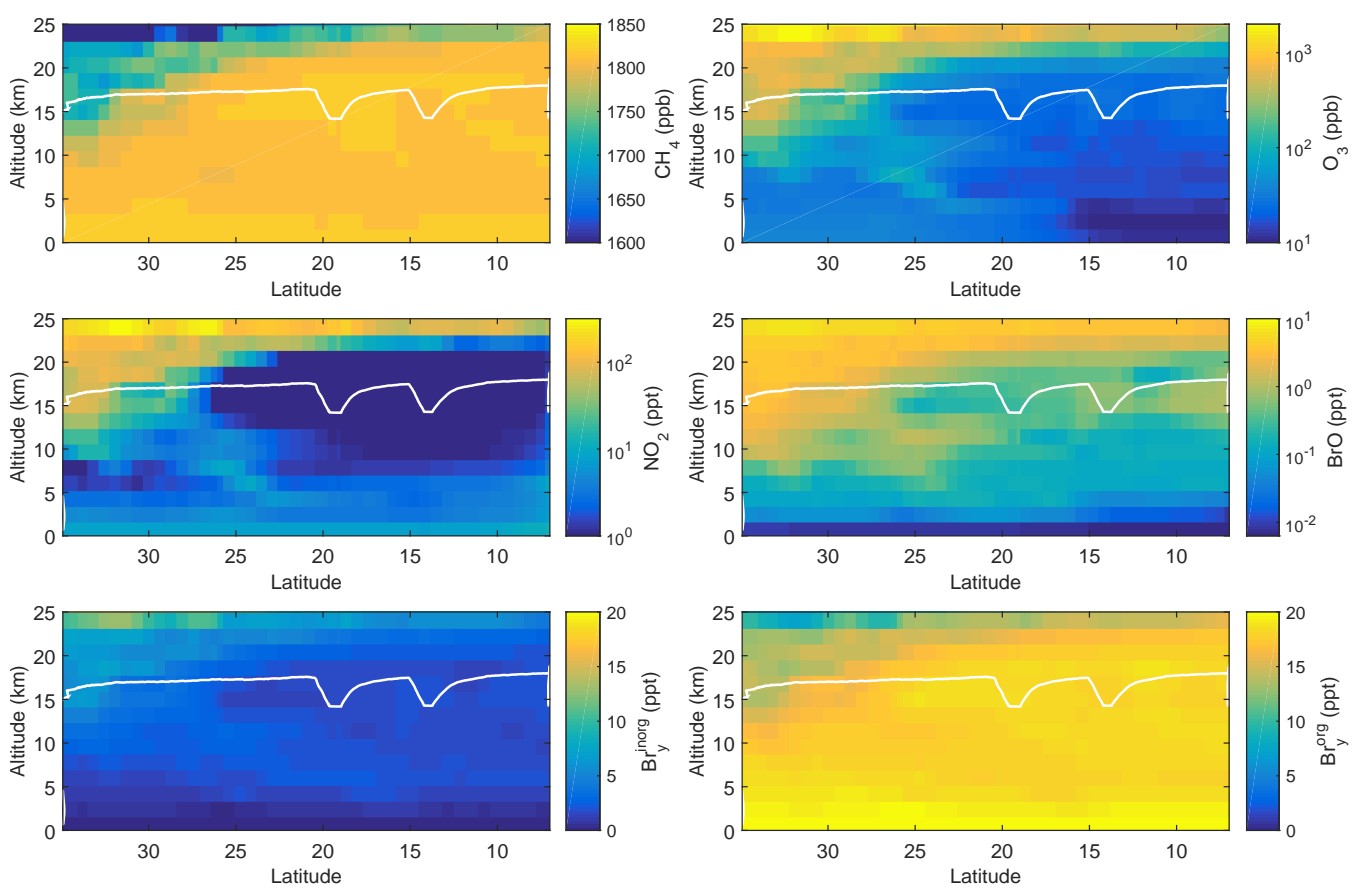

**Figure 2.** TOMCAT/SLIMCAT predictions of mixing ratio curtains of $CH_4$ (upper left), $O_3$ (upper right), $NO_2$ (middle left), BrO (middle right), $Br_y^{inorg}$ (bottom left), and $Br_y^{org}$ (bottom right) for the sunlit part of SF3-2013 (Feb. 14, 2013). Note the different color scale ranges. The white line is the flight trajectory of the Global Hawk. For better visibility, the simulated mixing ratios are shown for the altitude range 0 - 25 km, although the TOMCAT/SLIMCAT simulations cover the range of 0 - 63 km altitude

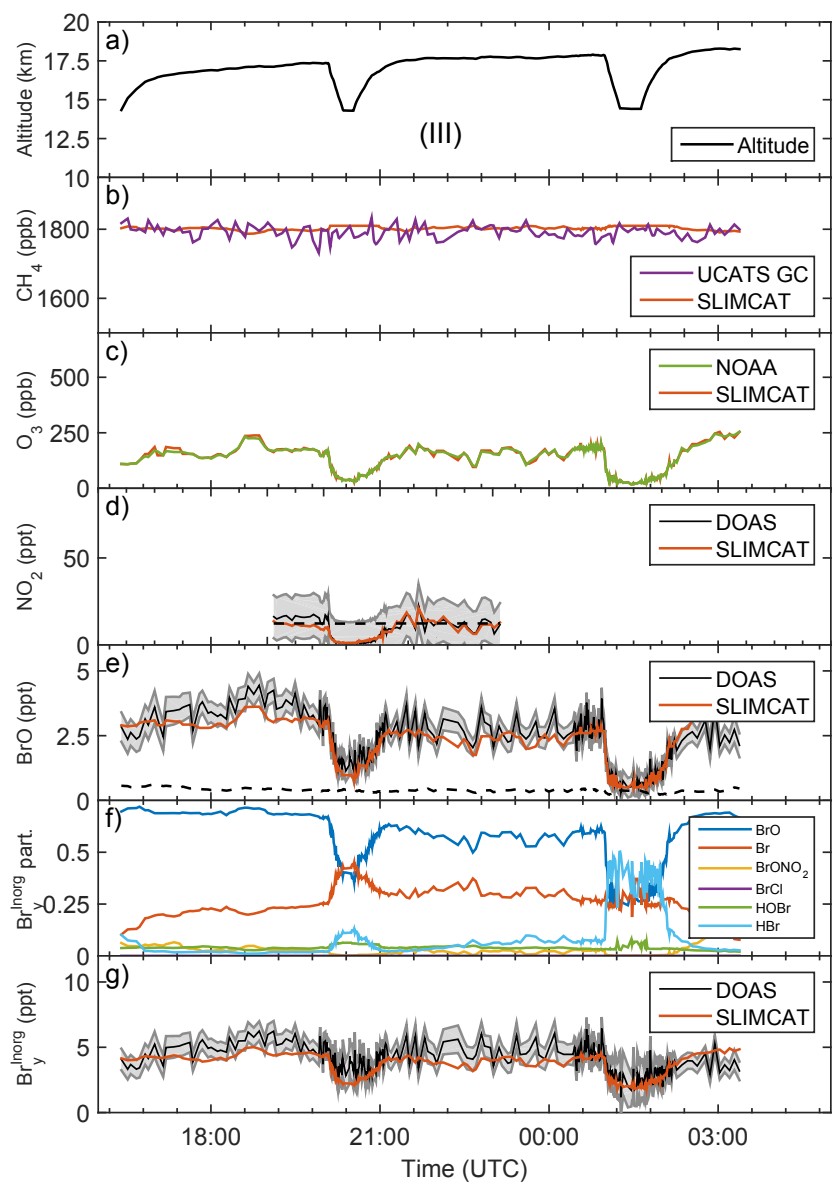

**Figure 3.** Panel (a) shows the time-altitude trajectory of the sunlit part of the GH flight track (SF1-2013) on Feb. 4/5, 2013 (SF1-2013). Panels (b)-(e) show inter-comparisons of TOMCAT/SLIMCAT-simulated fields with observations of (b) $CH_4$ (UCATS), (c) $O_3$ (NOAA), (d) $NO_2$ (mini-DOAS), and (e) BrO (mini-DOAS). The grey-shaded error bars of the mini-DOAS $NO_2$ and BrO measurements includes all significant errors, i.e. the spectral retrieval error, the error due to a contribution to the slant absorption from above the aircraft and from the troposphere, and the absorption cross section uncertainty. Panel (f) shows the SLIMCAT modeled $Br_y$ partitioning for the standard run #583. Panel (g) shows a comparison of inferred and modeled $Br_y^{inorg}$, including the uncertainty as a grey band.

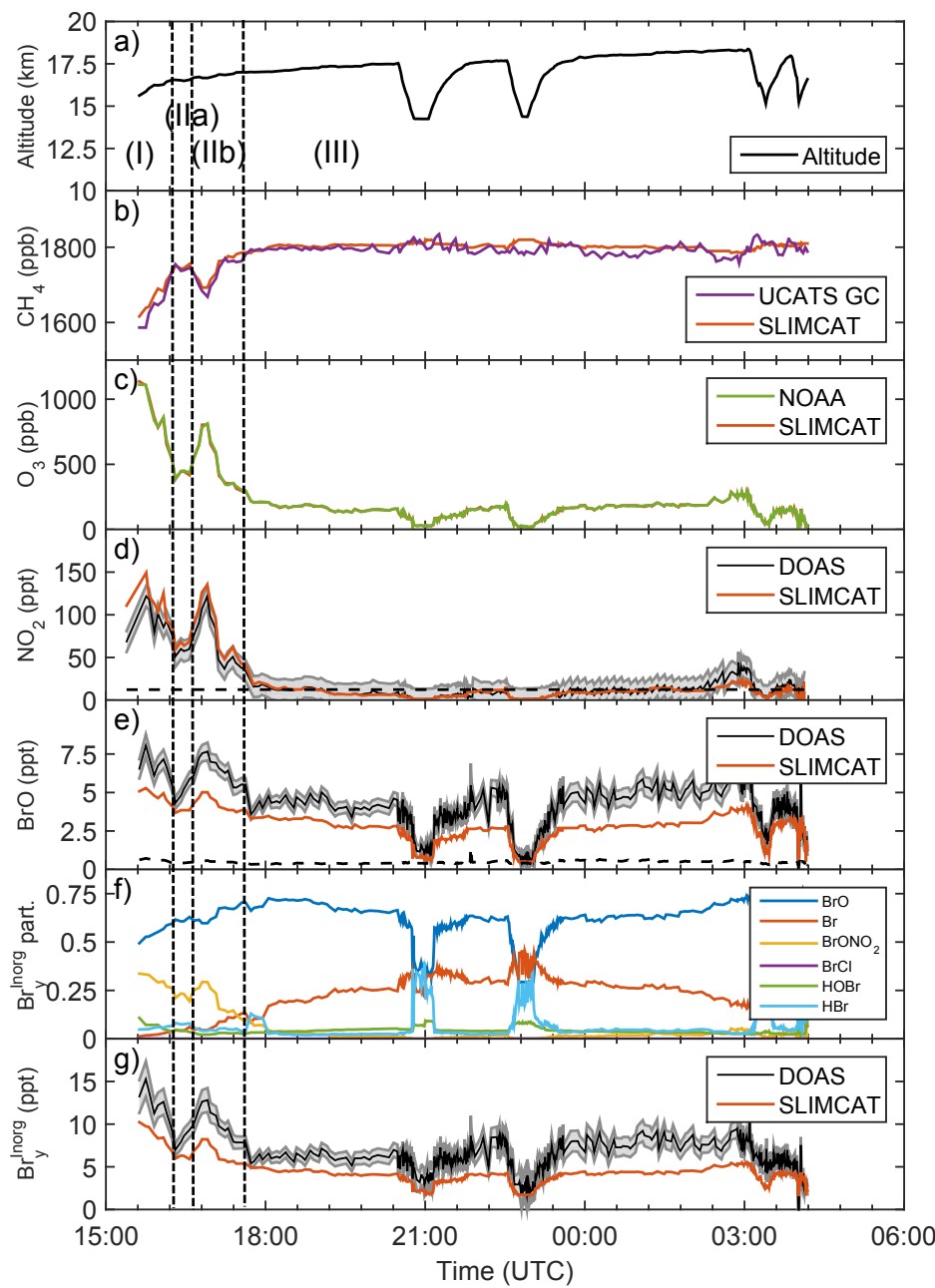

**Figure 4.** Same as figure 3 but for the research flight on Feb. 9/10, 2013 (SF2-2013). The dashed vertical lines in Figures 4-9 separate different atmospheric regimes: (I) is the extra-tropical lowermost stratosphere, (IIa, IIb, ...) different mixing regimes of air from the extra-tropical lowermost stratosphere, and (III) from the tropical tropopause layer.

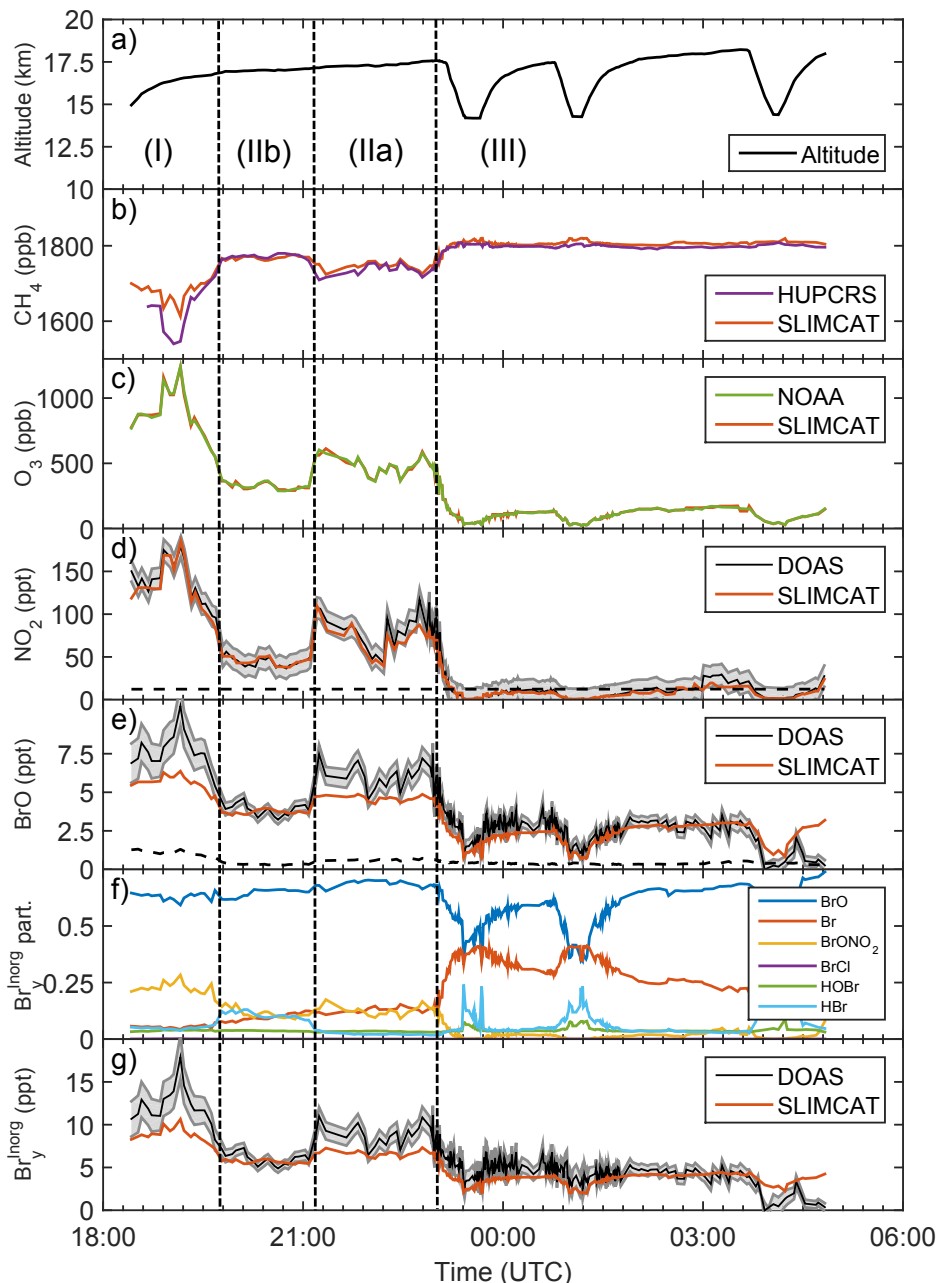

**Figure 5.** Same as figure 3 but for the research flight on Feb. 14/15, 2013 (SF3-2013).

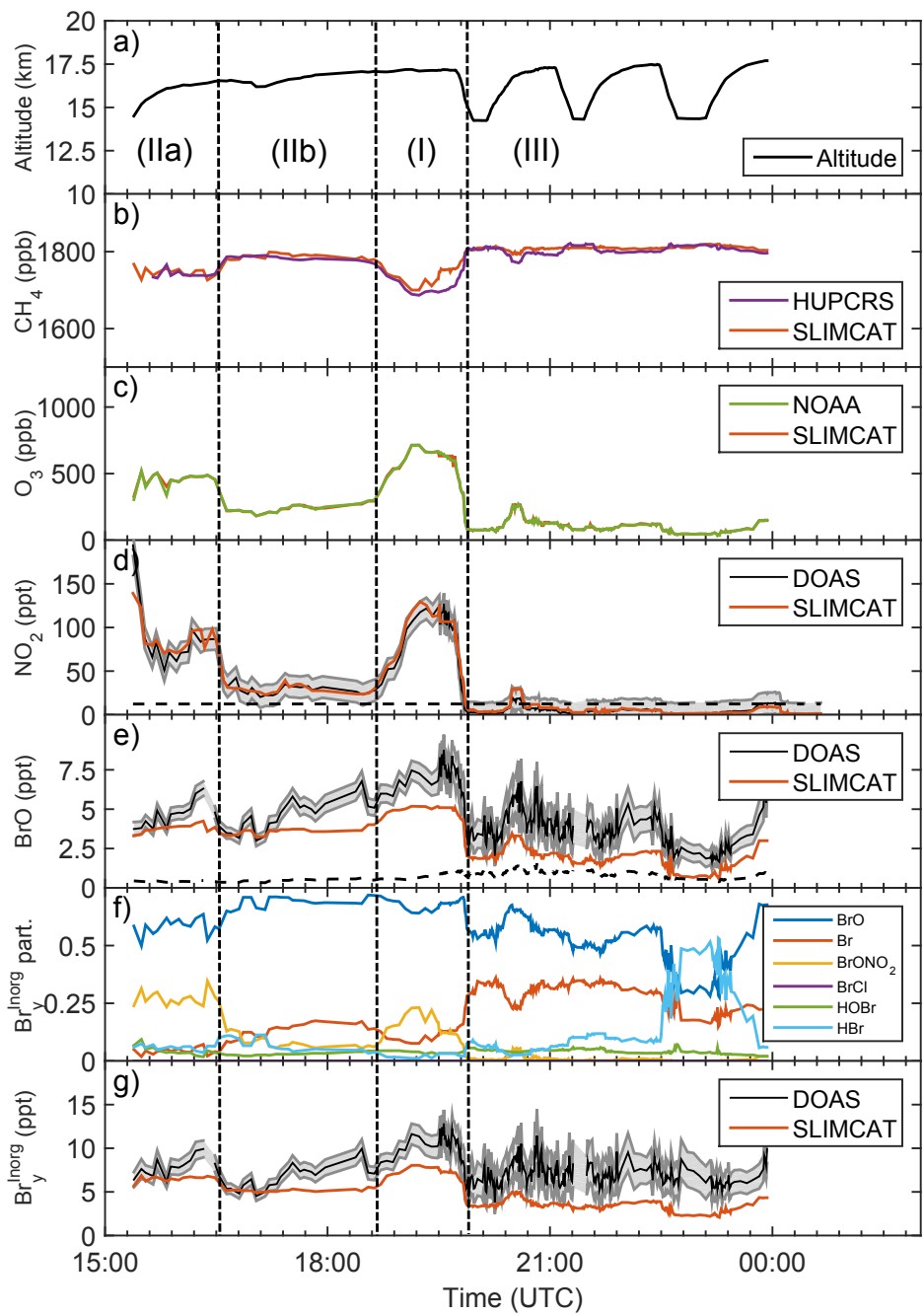

**Figure 6.** Same as figure 3 but for the research flight on Feb. 21/22, 2013 (SF4-2013). Note that DOAS analysis of BrO for SF4-2013 is somewhat uncertain because the Fraunhofer reference spectra (taken via a diffuser) are affected by temporally changing residual structures likely due to ice deposits or some other residues on the entrance diffuser (see text).

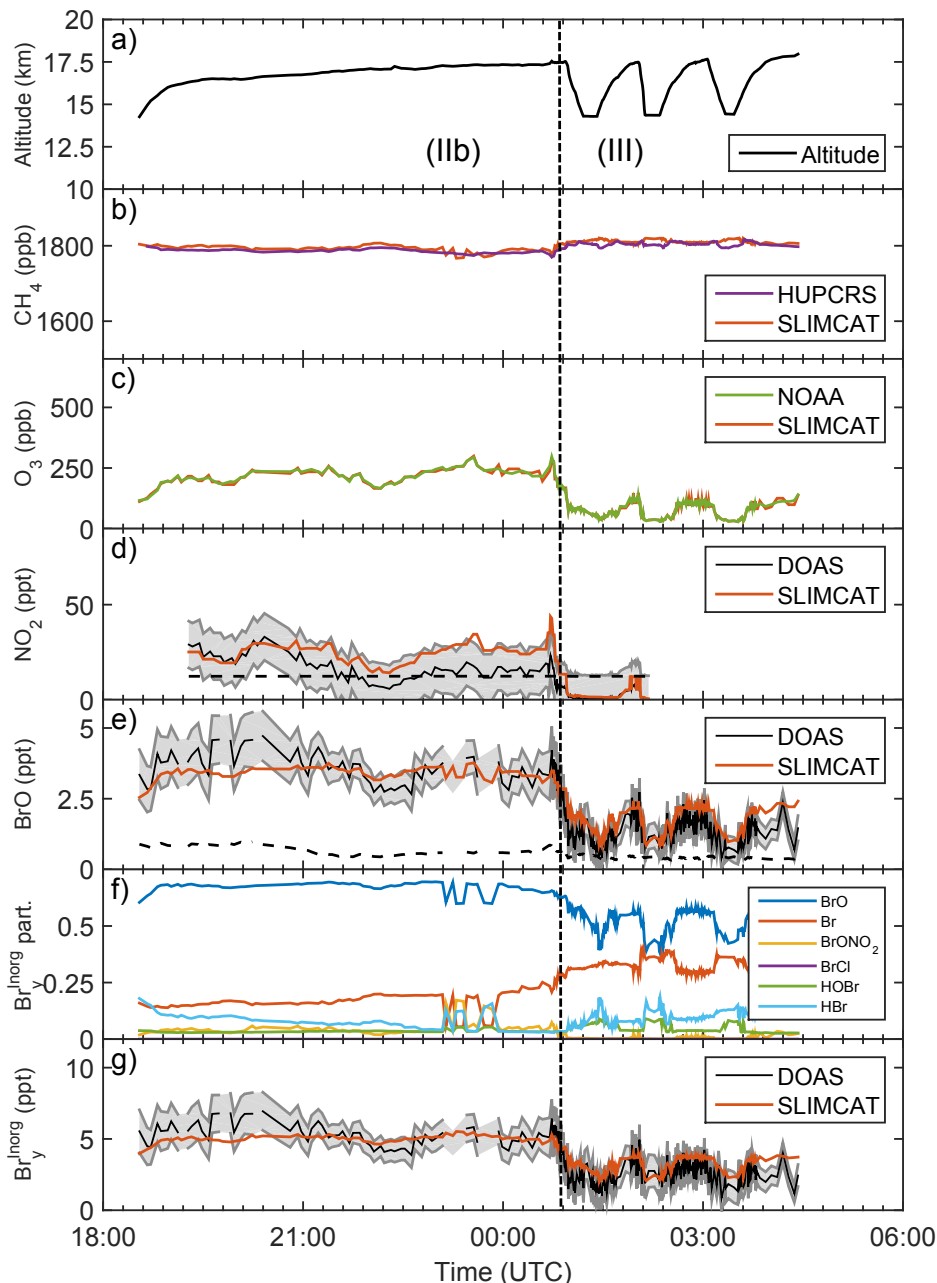

**Figure 7.** Same as figure 3 but for the research flight on Feb. 26/27, 2013 (SF5-2013).

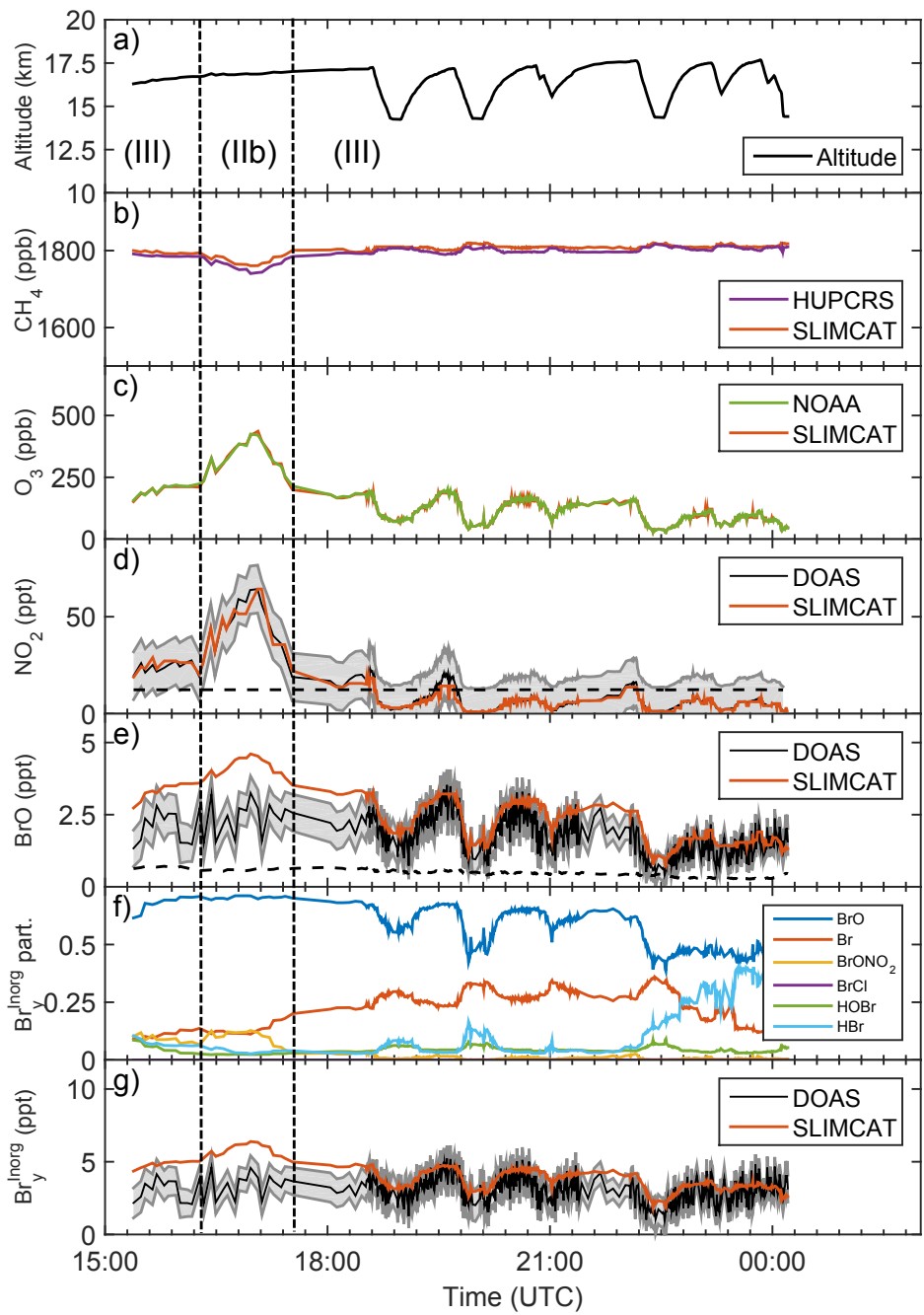

**Figure 8.** Same as figure 3 but for the research flight on Mar. 1/2, 2013 (SF6-2013).

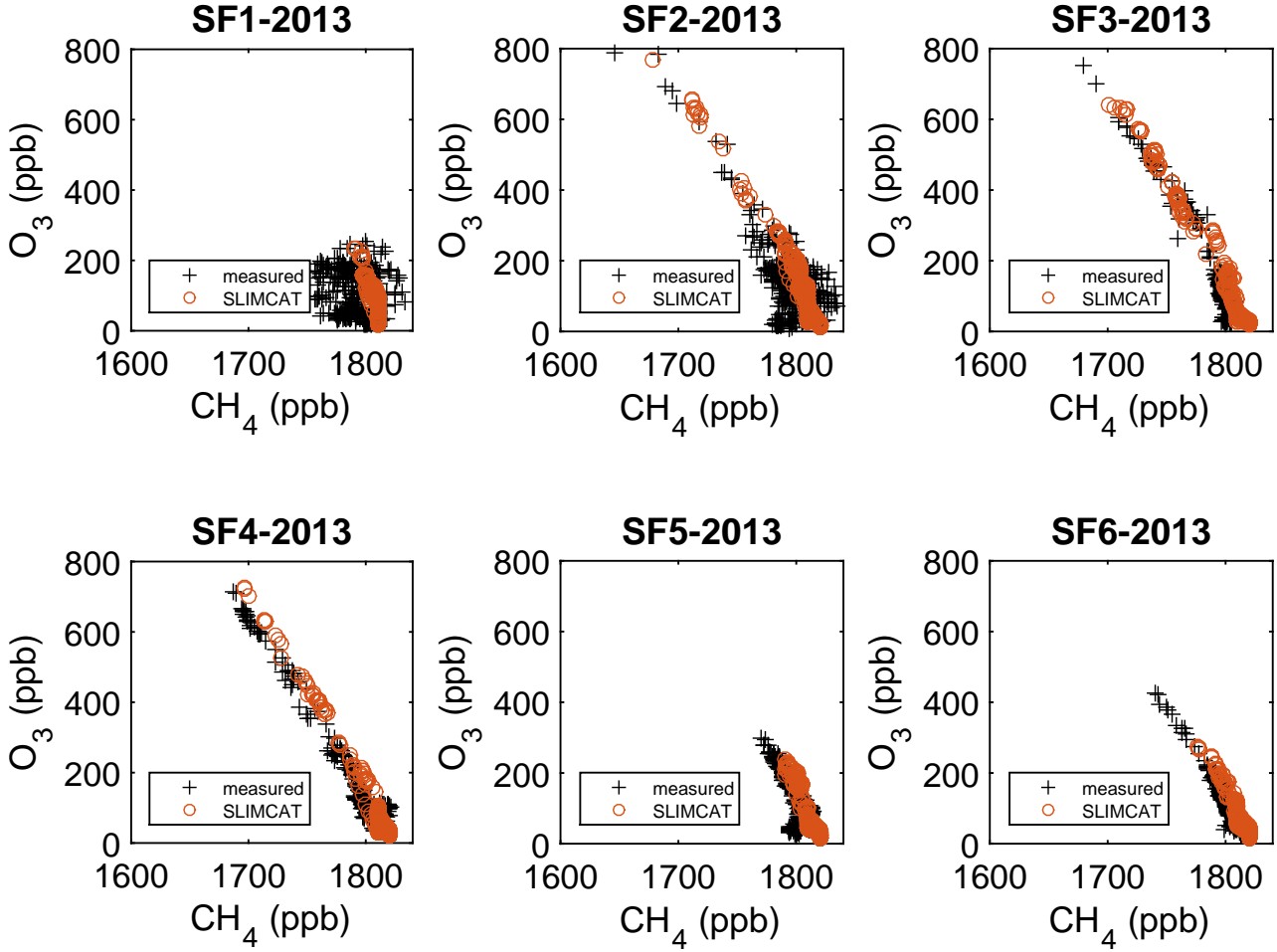

**Figure 9.** Correlation of observed $CH_4$ (UCATS SF1-2013 and SF2-2013; HUPCRS SF3-2013 to SF6-2013) and $O_3$ (NOAA) for the 6 NASA-ATTREX science flights in 2013. Also shown are the corresponding correlations from the TOMCAT/SLIMCAT simulation.

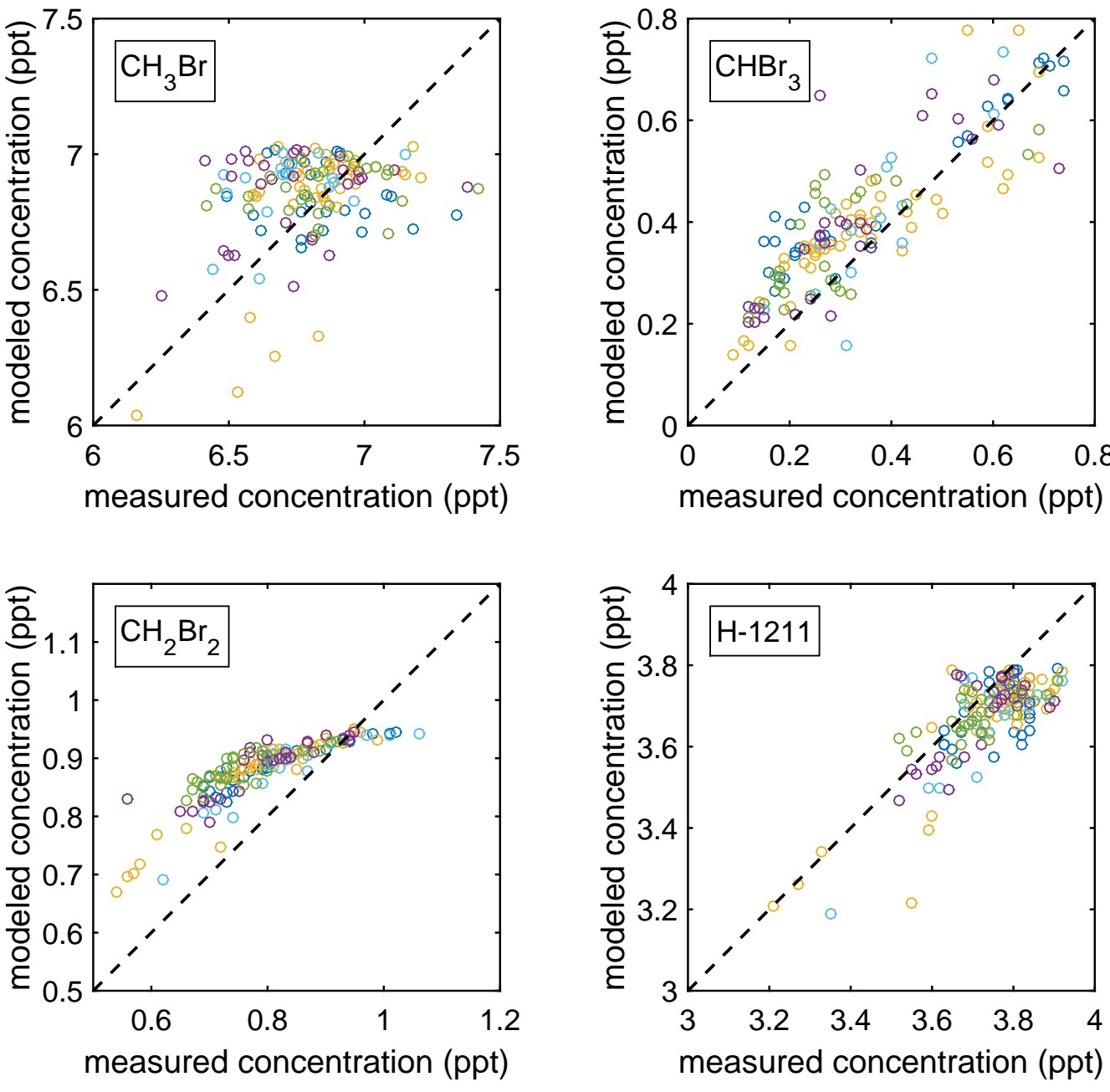

**Figure 10.** Correlation of GWAS measured and TOMCAT/SLIMCAT modeled major brominated source gases. Upper left panel for $CHBr_3$, upper right panel for $CHBr_3$, lower left panel for $CH_2Br_2$, and lower right panel for Halon 1211. The concentrations for different flights are color-coded; SF1-2013 in blue, SF3-2013 in yellow, SF4-2013 in light blue, SF5-2013 in purple, and SF6-2013 in green.

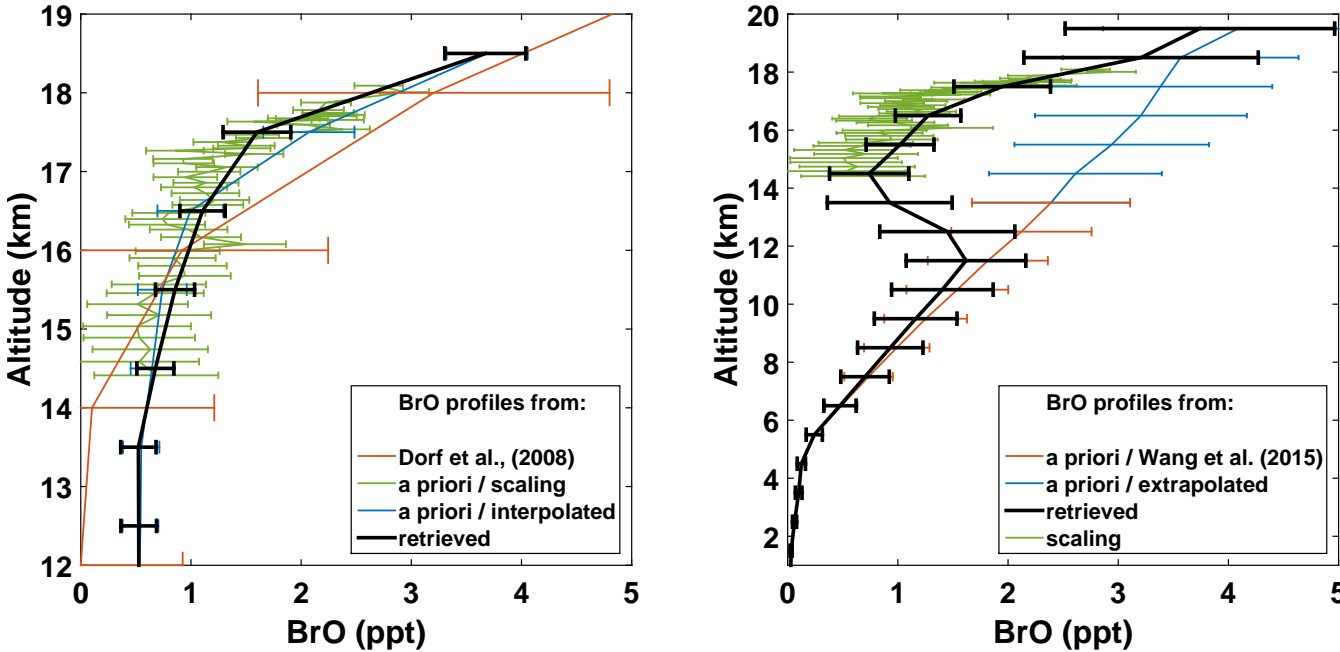

**Figure 11.** Comparison of the inferred BrO profile for the ascent after dive # 2 of the flight on Feb. 5/6, 2013 with previously published (modeled and measured) BrO profiles. Please note the different altitude range in the two panels. BrO profiles retrieved using the optimal estimation method are shown in black and those using the $O_3$-scaling technique are shown by green symbols, error bars and lines. In the two panels different a priori information is used to constrain the optimal estimation retrieval. Left panel: TOMCAT/SLIMCAT model predictions are used as a priori (blue). Also shown for comparison is the BrO profile published by Dorf et al. (2008), which was measured over northeastern Brazil in June 2005 (red). Right panel: The BrO profile of Wang et al. (2015) (red) and its extrapolation to 20 km (blue) are used as a priori in the optimal estimation. The kink in the retrieved BrO profile (black) at about 12 km strongly indicates that BrO profile of Wang et al. (2015) is neither compatible with the BrO profiles inferred using $O_3$-scaling technique (green) nor from optimal estimation (black) (for further details see section 4.3).

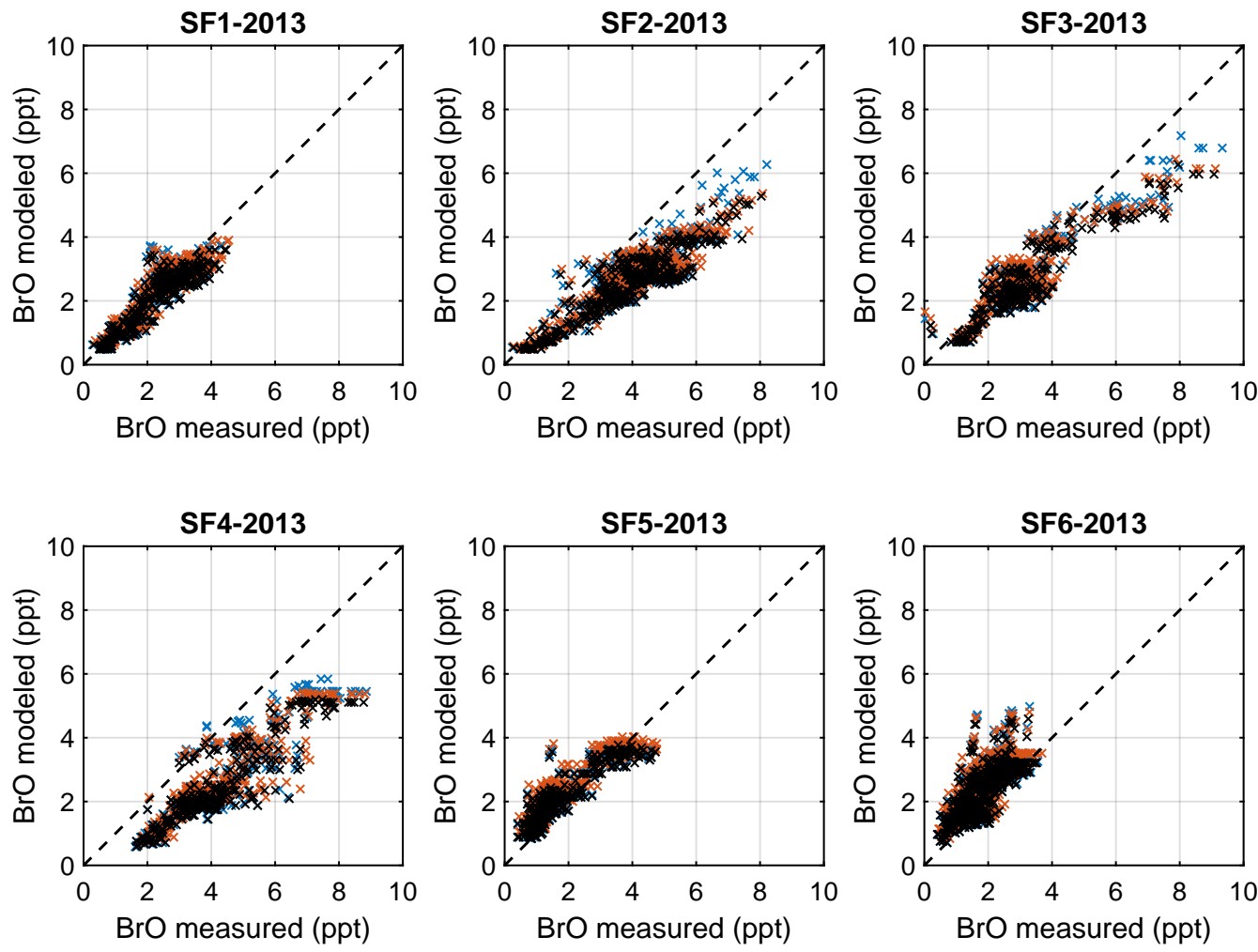

**Figure 12.** Comparison of measured and modeled BrO for the NASA-ATTREX science flights 1 to 6 in 2013. Black crosses are for model run # 583, blue crosses for # 584, and red crosses # 585.

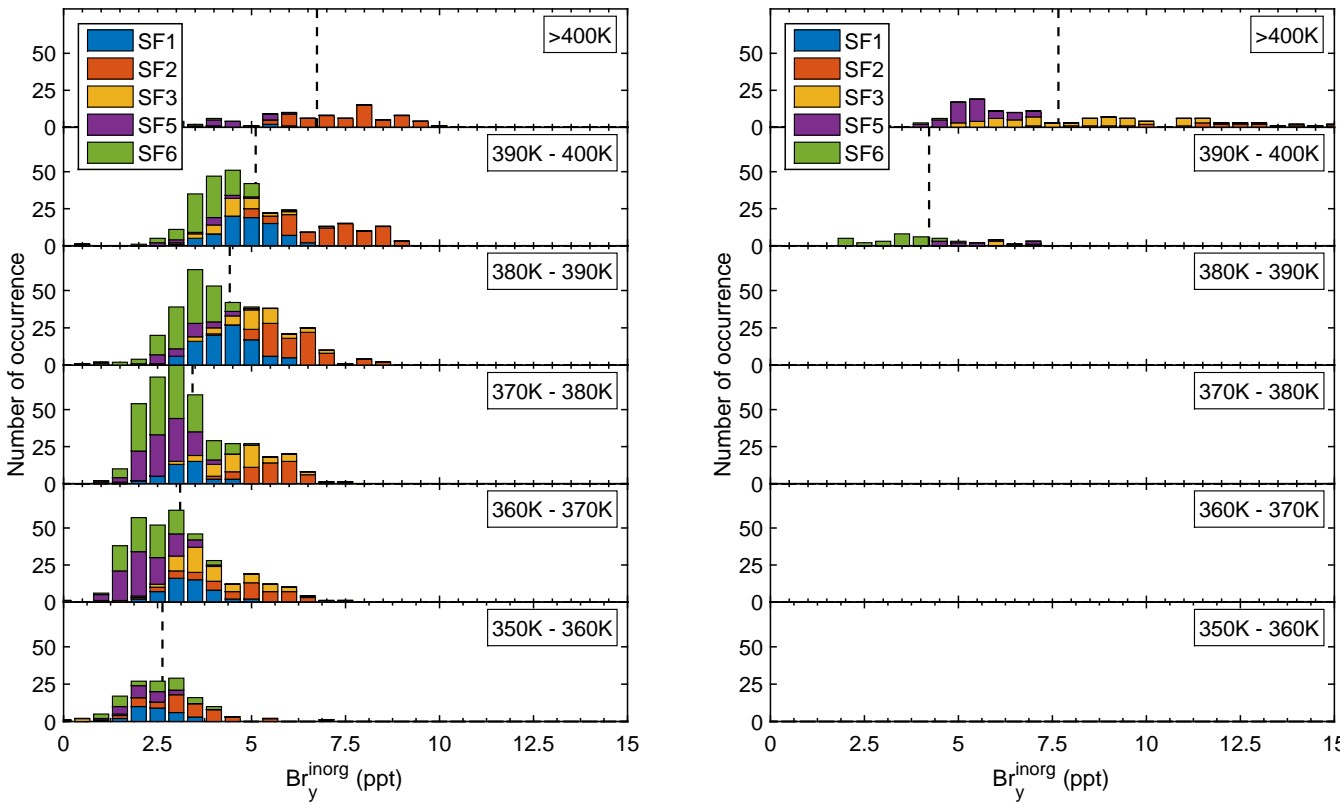

**Figure 13.** Histogram of $Br_y^{inorg}$ occurrence as a function of potential temperature for $[CH_4] \geq 1790$ ppb (left panel) and $[CH_4] \leq 1790$ ppb (right panel). High $[CH_4]$ can be considered as a marker for young air mostly found in the freshly ventilated TTL while low $[CH_4]$ can be considered as a marker for aged air mostly found in the subtropical lowermost stratosphere. The mean and the variance of $Br_y^{inorg}$ for young air (left panel) are: for $\theta = 350 - 360$ K, $2.63 \pm 1.04$ ppt; $\theta = 360 - 370$ K, $3.1 \pm 1.28$ ppt; $\theta = 370 - 380$ K, $3.43 \pm 1.25$ ppt; $\theta = 380 - 390$ K, $4.42 \pm 1.35$ ppt; $\theta = 390 - 400$ K, $5.1 \pm 1.57$ ppt, and $\theta \geq 400$ K, $6.74 \pm 1.79$ ppt. Aged air (right panel): for $\theta = 390 - 400$ K, $4.22 \pm 1.37$ ppt, and $\theta \geq 400$ K, $7.67 \pm 2.72$ ppt.

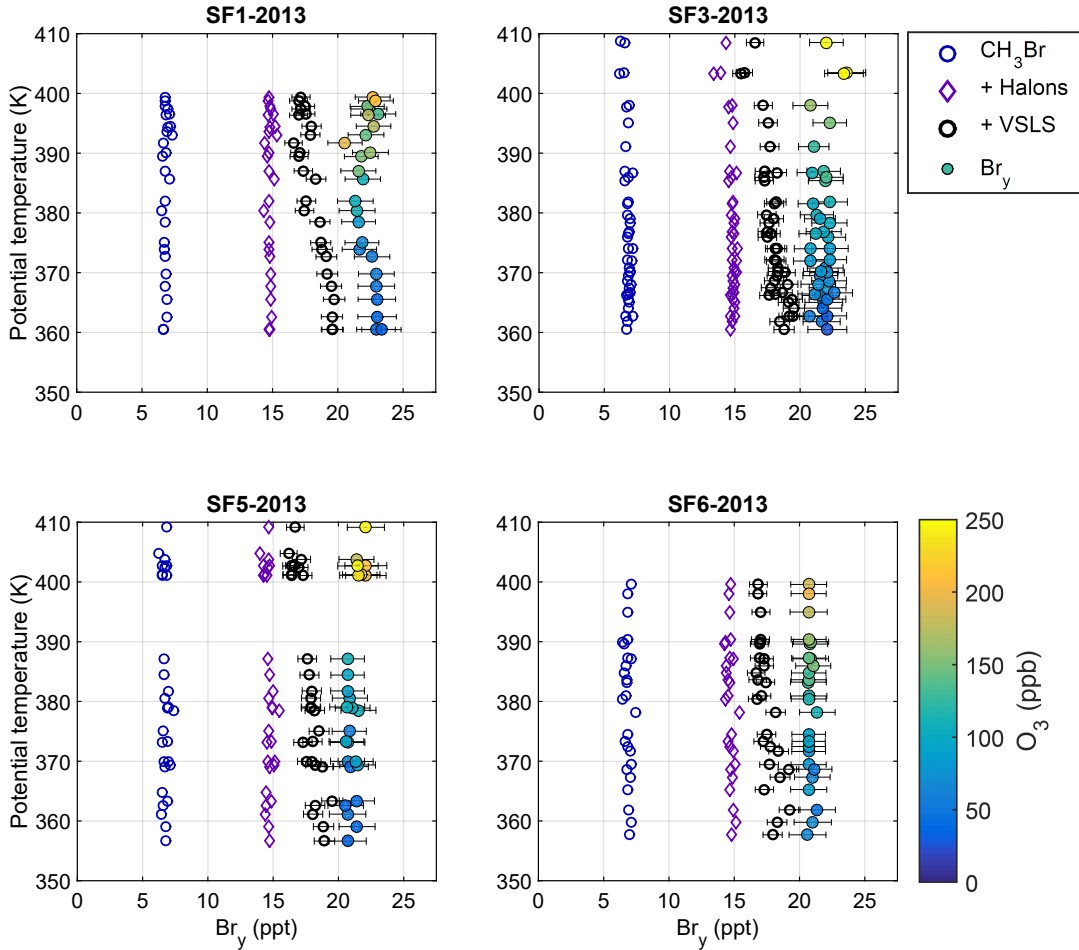

**Figure 14.** $Br_y$ as a function of potential temperature ($\theta$) for all dives during the 2013 NASA-ATTREX flights, when joint measurements of $Br_y^{org}$ and $Br_y^{inorg}$ are available.

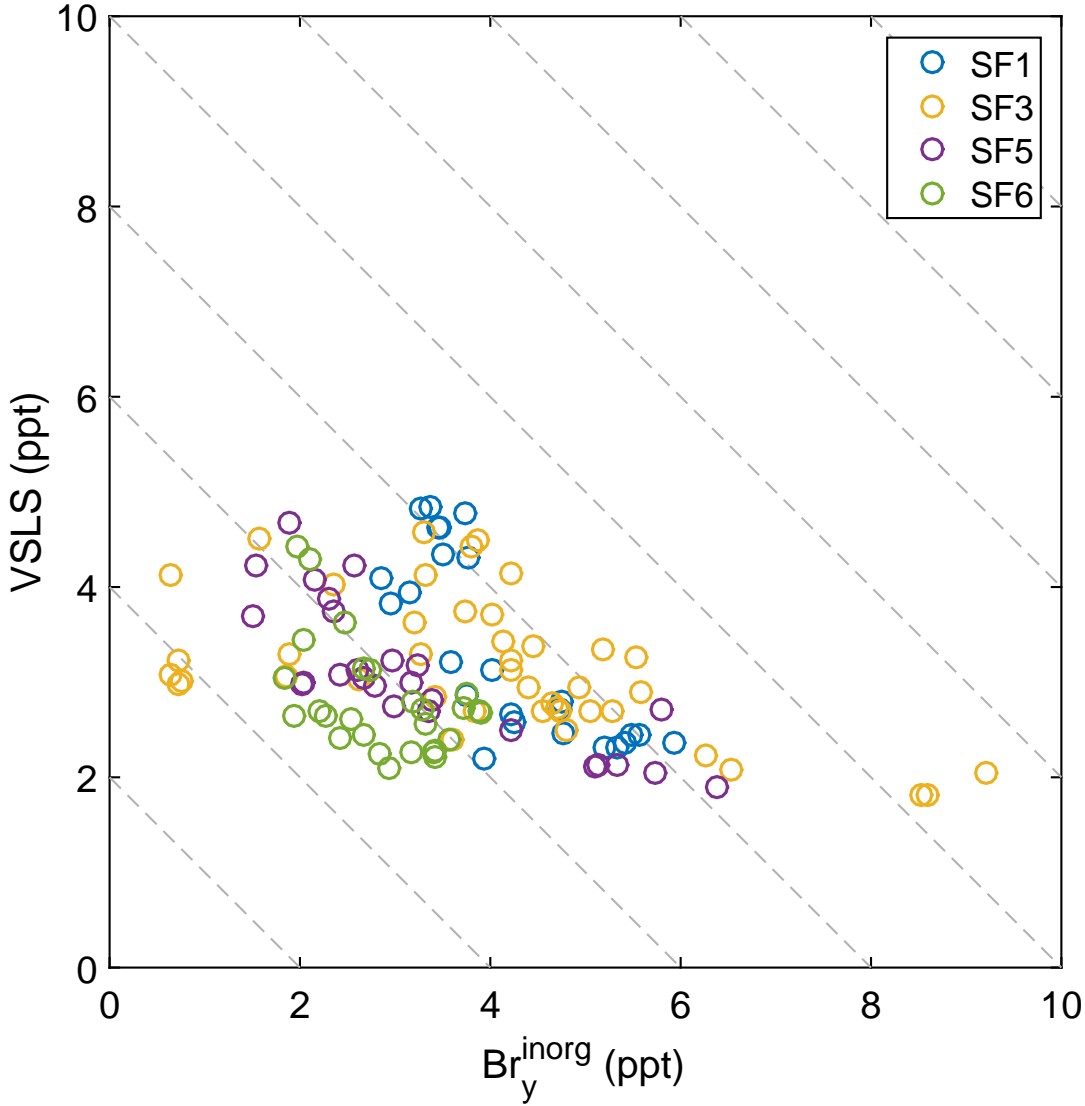

**Figure 15.** $Br_y^{inorg}$ as a function of the sum of all brominated VSLS using the same color code as in figure 10. If all $Br_y^{inorg}$ resulted from destroyed VSL bromine of the same air mass from near the surface, then all data points should follow individual diagonal lines.