# Peer review of "Probing the subtropical lowermost stratosphere, tropical upper troposphere, and tropopause layer for inorganic bromine"

_Atmospheric Chemistry and Physics, 2016_

## Referee Comment (RC1) · Anonymous Referee #1 · 14 Oct 2016

**General Comments:**

The paper reports results from the ATTREX campaign performed in the Eastern Pacific during 2013, which probed the UT, TTL and LMS. The experimental results are of high importance for determining the overall fraction of inorganic bromine injected to the stratosphere, i.e. capable of destroying ozone in the LMS. BrO measurements are compared with modelled output from the TOMCAT/SLIMCAT model, which runs at high resolution with the specific meteorology (i.e. ECMWF Era-Interim reanalysis) present during the ATTREX campaign.

Even when the presented results possess a notable scientific impact as they probe inorganic bromine species in a region of major importance for constraining stratospheric bromine injection, the presentation of results is in this reviewer´s opinion rather disordered, and the description of the model configuration/output used to interpret their measurements is rather confusing. Therefore several issues, both structural/technical and scientific/descriptive, must be dealt with before this MS is ready for publication. I recommend the following major comments be addressed before this paper can be accepted in ACP.

**Major Comments:**

1. The abstract is too long and contains unnecessary information such as i) the instruments used for probing the composition of the tropical atmosphere, ii) a description of the minor deficiencies in the modelled transport, iii) a reference to a paper with the definition of the TTL. Also, the most important information in the abstract is quite disordered and the main results (i.e., measured numbers which are the major findings of this work) are only given in the very last lines.

2. The authors used a quite simple proxy to separate tropospheric air from stratospheric air: the condition of $[CH_4]$ being larger or smaller than 1790 ppb. This selection is neither justified nor referenced within the MS, and its use should be clearly justified as many of the forthcoming results depend on the validity of this assumption. Indeed, the histograms shown in Fig. 13 seem to contradict the validity of the $CH_4$ proxy in splitting tropospheric from stratospheric air for theta > 390 K. Also, within the abstract and text, the $CH_4$ condition is defined both respect to 1390 and 1790 ppb, introducing an additional inconsistency to the definition.

3. In the abstract and methodology, the authors declare that the TOMCAT/SLIMCAT model was "constrained to the measured O3 and NO2 and adjusted to match the observed concentrations of some brominated source gases" (P2,L6-7). But no detail is given in the methodology about how this special configuration was applied in the model. Later, many of the results are interpreted and discussed based on the output obtained from the global model. In light of the importance of the observations and conclusions drawn here, mostly the inferred $Bry^{inorg}$, the authors should describe specifically how the model was constrained or adjusted. For example:
   a. My major concern is about how removal/washout is considered in the model, mostly within the UT and TTL, as these processes will control the overall $Bry^{inorg}$ burden in that region of the atmosphere. The only reference to removal rates in the MS that I could found was on Page 15: L26 "…, and assuming no bromine is effectively lost in the troposphere, …" and L30-31 "Therefore effective loss processes for inorganic bromine, for example by heterogeneous uptake of inorganic bromine on aerosol and cloud particles, must act in the atmosphere". Detailed information on the removal

processes considered here for brominated species should be given and also how they affect the BrO/Br ratios.

   b. Also, In the model description (P8,L21-23), the surface concentration of VSL is 1.00 pptv for CHBr3, CH2Br2 and other VSLS. Why at (P12,L12) a value of 1.05 pptv is informed?. Further on, in the conclusions (P17,L15) the 1.0 ppt value is mentioned again. It may simply be a typo? Even when a 0.05 ppt value will not make a difference, this point should be made clear and consistent. How were the surface emissions adjusted?

4. In relation with my previous comment about an improved description of the specific model configuration used in this study, the authors state that "No other (c.f., unknown organic or inorganic) sources of bromine for UT, LS, and TTL are assumed (e.g., Fitzenberger et al. (2000), Salawitch et al. (2010), Wang et al. (2015), and others), except that we add 0.5 ppt to the modeled tropospheric BrO in agreement with the finding discussed below (section 4.6)". Later in Section 4.6, no specific mention is given about this additional source of BrO.

   a. As BrO is used to constrain inorganic bromine using TOMCAT/SLIMCAT, a clear description of this additional source of BrO must be given in the text.

   b. Yang et al., 2005 and Ordoñez et al. 2012 show that an additional source of inorganic bromine from sea-salt in the MBL is required to reproduce observations. Fernandez et al 2014 highlighted the importance of the sea-salt contribution for Bry in regions of strong convection such as the tropical western-pacific. Is this additional source of BrO in this work related to sea-salt recycling?, if so, is it constrained as a boundary condition or explicitly calculated? Please expand the discussion about this important omission. In Section 4.6 (P16,L10-13) the impact of sea-salt (or any other additional source) on Inferred Bry$^{total}$ should also be discussed.

5. In Section 2.1 (DOAS measurements of O3, NO2 and BrO), a companion paper (Stutz et al., 2016), describing the DOAS measuring technic used during the ATTREX campaign, is introduced. However, the current manuscript makes too many references to Tables, Figures and Sections within the Stutz et al. paper, which do not introduce additional clarification and in most cases difficult a direct reading of the main results. Please, revise the whole manuscript on this respect and keep the references to the Stutz et al., 2015 only when they are relevant for the results presented here.

6. The results presented here somehow contrast with previously published measurements/modelling approaches. While the discussion respect to the findings from Wang et al., 2015 and Volkamer et al., 2015 are extensively discussed in section 4.3 (P13,L3-26), just a brief comment on the discrepancies respect to Fernandez et al., 2014 and Saiz-Lopez and Fernandez, 2016 is presented (Section 4.5, P15,L0-4). With regards to the [Br]/[BrO] ratio, the common pattern shown in Figs. 4-9 is that whenever BrO mixing ratios decrease, both HBr and atomic Br increase. When this inversion is observed, the flight altitude and O3 levels also decrease, indicating that the lower TTL is being probed. In most cases HBr surpass Br, while at some points (e.g., Fig 5f, 23:00) [Br] dominates. Thus, the Br and HBr prevalence seems to depend on the height (and possibly temperature, not shown) at which the TTL is being probed. I would suggest further discussion and interpretation of those results. Particularly, if the absence of heterogeneous recycling is affecting the inferred BrO/Br ratio?.

7. In section 4.1 the authors directly start showing results for Figs. 4-9. Not a single description is given to Fig. 3. Thus, Fig. 3 should be moved further down in the text. In P11,L17-20 you state that "The excellent

agreement achieved between measured and modeled CH4, and O3 lends confidence that the altitude-adjusted TOMCAT/SLIMCAT model fields reproduce well the essential dynamical and photochemical processes of the probed air masses.". The authors can only assure a sentence like this by means of a full vertical profile validation of the species, not only comparing data at a specific level.

8. The works possess several informal phrases which may not fit well within a scientific work. I list below some of them (but certainly not all), that maybe should be rephrased to a more appropriate structure, justified by numbers/references or removed from the MS.

    Abstract, P2,L4: "… and the expectation based on the destruction of brominated gases."
    P4,L1: "… indicating that some Bry_inorg (i.e. several ppt) is directly transported from the tropopause"
    P9,L17: "…, some exciting observations and details"
    P11,L4: "… we refrain from this much more complicated approach"
    P11,L27: "… NO2 meet the expectations for NOx"
    P16,L24: "By far …"

9. There are several other sentences that may benefit from revision. I list some of them below:

    P8,L22: [CHClBr2,CHCl2Br,CH2ClBr,…]. What the three dots means? Please specify.
    P12,L3: NOy=(NOx,N2O5,HNO3,HO2NO2,…. What the three dots means? Please specify.
    P17,L14: Once again [CHClBr2,CHCl2Br,CH2ClBr,…]
    P18,L3: (Here: cite Hossaini 2016, acp when published)?

**Figures Comments:**

Fig. 1: All results shown in this paper are for the ATTREX Flights performed during 2013 in the Eastern Pacific. Then why do you show panel A with the ATTREX 2014 flights that are not used here?

Fig. 3: This Figure should be moved down in the text to make it consistent with the presentation of results. Also, it may be unified into a 2 panel figure, with only 1 caption that distinguishes between the a) and b) panels. The same applies for Fig. 13.

Fig. 4: A portion of the caption is completely missing. This should be carefully controlled before submission. Also, following the equivalent figures for the rest of the flights (Figs. 5-9), no mention is given to what the vertical dashed lines represent. In addition, it would be very helpful to show a constant (dashed) line for O3 = 150 ppb and CH4 = 1790 pptv to distinguish the periods where the subtropical and TTL air was being probed.

Wouldn't it be a good idea to show the "mean temporal profile" measured by all of the ATTREX flights (i.e., for equivalent SZA)??

Fig. 13: The large values of BrO at theta = 390-400 K within the freshly ventilated TTL (BrO > 7.5 pptv) shown during flight SF2 makes me doubt about the ability of the the CH4 < 1790 pptv as a good proxy for distinguishing stratospheric air. Also, What is the meaning of showing Fig. 13b., for which most of the panels show no data at all??

Fig. 14: Flights SF2 and SF4 are not shown. Why not?. It would be a good idea to show the mean results for the campaign, and then highlight results for each of the flights.

Fig. 11: I was surprised about the dispersion on the modeled/measured scatter plot for CH3Br. Being the bromocarbon with largest lifetime, I would expect it to have an equivalent dispersion to the halons. No mention is given about this issue in the text.

**Additional Scientific Comments:**

P2,L15: Is there a direct assignation of WMO, 2014 values specifically to year 2011?

P3,L1: From which reference/s do you obtain the (2.5-4) ppt error on Bry inorg uncertainties within the inorganic method?

P3,L4: UT, TTL and LMS should be defined independently in the abstract and introduction. There is no sense to introduce exactly which definition of the TTL you used in the abstract/introduction.

P3,L32: Dorf et al., 2008 reported a contribution from VSL of (4.0 ± 2.5) for PG and (5.2 ± 2.5) for total (SG+PG). I do not find where you get the value of (2.5 ± 2.6) ppt. Also, there seems to be a confusion in the interpretation of contributions 3) and 4) from Dorf et al.

P4,L4: Saiz-Lopez et al., 2012 also estimated the climatic impact of VSL sources within a CCM.

P7,L10: Please describe which species were measured by the GWAS sampler, and which ones were used within this work.

P7,L27: "The received limb radiances …". Do you mean the limb radiances measured with the mini-DOAS instrument? Please specify.

P9,L3: Other works used this type of experimental data measurements on top of a model "curtains" (i.e Nicely et al., 2016). Some reference to any of these articles could be given here.

P9:L20: What do you mean by "the NASA-ATTREX flights of the Global Hawk were strongly biased with respect to the sampled air masses". Please clarify.

P11,L27-30: Please specify for which Flights the values for NO2 range between (70-170) ppt and were interpreted as belonging to the LMS.

P12,L9: What do you mean by "even if the data is scattered from flight to flight"?

P12,L17: Ordoñez et al. 2012 also describes the geographical and temporal variability of oceanic VSL sources.

P14,L6-7: Please explain better what you mean by "very young air" and "older air". Are you considering the [CH4] proxy given before?

P15,L5-7: A printed value of the overall Bryinorg error (and range) derived from the analysis shown on panel f of Fig. 4-9 would be helpful in the text.

P15,L22-26: All the analysis about the Bry$_{inorg}$ increment due to the decrease in VSLS is performed based on the "theoretical" dashed diagonal lines shown in Fig. 15. But the "modelled" diagonal lines are not shown for neither of the flights. Why? Including the modelled lines would strengthen the analysis, and justify the conclusions obtained here. Also, what do you mean by "…extrapolating the data points along lines of constant [VSLS]+[Bry$_{inorg}$] (…) to [Bry$_{inorg}$] = 0"? Do you mean getting the intercept of each of the (not shown) lines for each flight?

P16,L12: It is not clear how the range (0.5 to 5.25) pptv or uncertainty (± 1.04 pptv) of inferred Bry$^{inorg}$ values is computed. Are these the maximum-minimum values modelled with SLIMCAT for all flights? Is this a model average within the Eastern Pacific region where measurements took place?

P16,L19: The chemical loss rates were computed within the tropics (i.e, considering the 0º - 360º longitudes) or only within the Eastern Pacific region? It is important to make this clear, and in case of considering the whole tropics, a comparison to the values obtained within the EP is needed.

P16,L22-27: The chemical loss analysis gets into details of which independent families and specific channels are the major contributor to the total (or bromine) ozone loss rate. I suggest including a table/sentence defining all the families considered, and which reactions are considered at least for the bromine channel.

P17,L9: I do not agree that your results provide confidence in the modelled NOy photochemistry. You have only shown results for NO2 in this work. Much deeper analysis of nitrogen cycles should be given in order to perform this statement within the conclusions. Please see also text in P12,L3-4.

**Technical/Linguistic Comments:**

P2,L11: It is not necessary to include a minus sign whenever you state that the value represent a net destruction. There are many other places in the MS when this also occurs.

P2,L15: Within the text and figures, the terms Bry and Bry$^{Total}$ are both used to represent the same quantity. Please unify the criterion.

P2,L17. All halons must be written either with capital H or lower case h.

P2,L32: … several tenths of ppt.

P3,L15: Colombia, not Columbia

P5,L29: UAS acronym is not used at all in the paper, there is no sense to define it. Also, if the size is in Liters, then capital L should be used.

P13, L3-7: Consider rephrasing.

P14,L17: What about BrCl? It is shown in Figs. 4-9 but not included in the definition of Bry$_{inorg}$?

P14,L19-20: Have you thought on including in a table the most important reactions that were changed between the three sensitivity runs?

P15,L35: Is the call to Fig. 15. correct?. If so, please explain.

P17,L6: do you mean anti-correlated?

Fig. 2: The altitude range should be given in between brackets.

Fig. 15: Consider rephrasing the sentence starting with: "If all Bry …" for one in the form "The dashed diagonal lines indicate …"

---

## Short Comment (SC1) · 12 Nov 2016

Comment on:

Werner, B., Stutz, J., Spolaor, M., Scalone, L., Raecke, R., Festa, J., Colosimo, F., Cheung, R., Tsai, C., Hossaini, R., Chipperfield, M. P., Taverna, G. S., Feng, W., Elkins, J. W., Fahey, D. W., Gao, R.-S., Hintsa, E. J., Thornberry, T. D., Moore, F. L., Navarro, M. A., Atlas, E., Daube, B., Pittman, J., Wofsy, S., and Pfeilsticker, K.: Probing the subtropical lowermost stratosphere, tropical upper troposphere, and tropopause layer for inorganic bromine, Atmos. Chem. Phys. Discuss., doi:10.5194/acp-2016-656, in review, 2016.

Barbara Dix[1], and Rainer Volkamer[1, 2]

[1]Institute for Chemistry and Biochemistry, University of Colorado, Boulder, CO, USA

[2]CIRES, University of Colorado, Boulder, CO, USA

Citations of the manuscript are marked blue.

p.13, line 15ff: Our study on the sensitivity of the $O_2$-$O_2$ absorption measured in limb direction as function of the cloud cover underneath (see sections 4.2 and Fig. 7 in Stutz et al. (2016)) as well as the results presented by Volkamer et al. (2015) (in their Fig. 3) clearly demonstrates the limitation of the $O_2$-$O_2$ method to constrain the radiative transfer for UV/vis studies above an altitude 10 km, mostly because the bulk of the $O_2$-$O_2$ collisional complex is located near the surface. Therefore, any skylight analyzed for the $O_2$-$O_2$ absorption in limb direction may carry additional, or even predominantly information on the radiative transfer of lower atmospheric layers (see Figure 7 in Stutz et al. (2016)), rather than of the targeted atmospheric layers.

We generally agree that "skylight analyzed for the $O_2$-$O_2$ absorption in limb direction may carry additional, or even predominantly information on the radiative transfer of lower atmospheric layers", but want to point out that BrO profiles published in Volkamer et al. (2015) and Wang et al (2015) were neither affected by underneath cloud cover, nor by cirrus above. Sections 2.10 and 3.1 in Volkamer et al. (2015) discuss explicitly the effect of aerosol and clouds, and make fully transparent that the presented RF12 and RF17 case studies are not affected by clouds. Furthermore, we show below HSRL data from these two flights (Fig. 1) that make transparent that no aerosols or thin cirrus layers were present above the aircraft. Moreover, the authors are referred to Fig. 2 and Section 2.1 in Dix et al. (2016a), where it is shown that in cloud-free conditions measurements of $O_2$-$O_2$ are suitable to constrain RTM up to 15 km. The statement

by the authors is too broad, and certainly does not apply to the Wang et al. and Volkamer et al. case studies. This should be corrected.

[Figure]

**Figure 1.** *Comparison of $O_4$ ratios at 360 nm and 477 nm with HSRL particulate backscatter cross section data for TORERO RF04, RF12 and RF17 (Dix et al. (2016a); Volkamer et al. (2015); Wang et al. (2015)). Altitude resolved HSRL backscatter data is plotted and color coded along the flight track. Larger signals denote the presence of aerosol/clouds. HSRL is either measuring above or below the aircraft. The shading directly around the flight track seen in part of RF12 and RF17 is a near field effect that leads to erroneous large back scatter signals by HSRL. Green boxes in RF12 and RF17 mark data periods that were used for BrO, IO and $NO_2$ optimal estimation profile retrievals as published in Volkamer et al. (2015). Regular HSRL upward scans show that for these time periods no aerosol or cloud layers were present above the aircraft. For more information see Dix et al. (2016a and b).*

p.13, line 22ff: Furthermore, Wang et al. (2015), and Volkamer et al. (2015) did not use a stratospheric CTM to study the potential influence of changing overhead BrO concentrations on their results. As result, the predominant occurrence of atmospheric BrO in the stratosphere at daytime, and its potential column changes mostly due to a changing tropopause height (e.g., at the subtropical or polar jet) may mimic the presence of BrO in limb the direction, or at flight altitude.

This is incorrect, and a misleading reflection of the literature. First, Volkamer and Wang et al. (2015) used a stratospheric model (RAQMS) to study the influence of changing BrO concentrations above, and show that potential changes in the stratospheric BrO VCD, or apparent changes in the measured limb dSCDs due to a changing tropopause altitude do not affect the results. Second, the authors are referred to section 2.10 in Volkamer et al. (2015), and Fig. S4 in the SI text of Wang et al. (2015) for the excellent agreement with the aircraft microwave temperature profiler measurements and the location of the thermal tropopause in the model. Third, the supplement of Volkamer et al. (2015) shows that the stratospheric profile above the aircraft is accurately corrected. Finally, Dix et al. (2016a) used RAQMS BrO profiles for the correction of stratospheric BrO contributions to the limb dSCD measurement, and confirms excellent agreement with the optimal estimation case study profiles from Wang et al. (2015) and Volkamer et al. (2015) using a parameterization method within low error bars.

p.13, line 27ff: In conclusion, even though the reported TORERO flights 12 and 17 were performed under clear-skies (Volkamer et al., 2015), it is unclear the extent to which unaccounted scattering due to aerosols and (probably) optically thin upper tropospheric clouds, lower level clouds, or changing overhead stratospheric BrO contributed to the inferred (or by error attributed) elevated BrO in the UT, and around the bottom of the TTL.

This conclusion is incorrect in all aspects listed. See our above responses. TORERO flight RF12 and RF17 are neither affected by aerosol/cloud extinction above, nor lower level clouds below, nor changing stratospheric BrO.

Caption of Figure 3b: …, the unexpected kink around 12 to 13 km in the inferred BrO profile when the inversion is constrained to the Wang et al. (2015) BrO profile indicates that our and the Wang et al. (2015) BrO profiles are not compatible … .

We respectfully disagree, and show below that the results presented in Fig. 3b of Werner et al. (2016) and Wang et al., (2015) are in fact quite compatible.

Werner et al. show that optimal estimation (OE) profile retrievals in Figs. 3a and b yield within error bars the same results for the altitude range between 14.5 and 18.5 km, regardless of a priori profile choice. This shows that the OE inversion is well constrained by measurements for these altitudes. However, below 14.5 km, the measurements by Werner et al. are not well constrained, and essentially follow the a priori in both cases shown. The "unexpected kink around 12 to 13 km" is therefore not unexpected at all, but to the contrary, it is the expected result of the OE solution that transitions from 'constrained by measurements' (above 14 km) to reproducing the a priori profile at lower altitudes (below 12 km). This behavior is likely reflected in the averaging kernel that are not shown, and should be included in the manuscript. Also, is the OE based on limb spectra only or are downward scans included? This information is missing in the paper.

Furthermore, section 4.4 in Stutz et al. (2016) states that GH measurements during SF3 are compatible with up to 1.5 pptv of BrO directly below flight altitude. This is quite compatible with the TORERO campaign average BrO vertical profile, which shows a significant decrease of BrO above 14 km, with a mean of 1.86 ± 0.16 pptv at 13.5 km, and 1.38 ± 0.16 pptv at 14.5 km (Dix et al., 2016b). The TORERO average profile is compared with model predictions in Fig. 5 in Schmidt et al., (2016), and is shown with a better resolution in Fig. 10 of Dix et al. (2016b) (included as Fig.2 here). Notably, the case studies in Volkamer et al. (2015) and Wang et al. (2015) are 100% consistent within low error bars (5%) with the parameterization retrieval (Dix et al., 2016a), if the same data subsets are compared. These case studies had probed primarily air masses influenced by convection over oceans. The lower mean BrO for the complete TORERO data set is mainly reflecting different air mass histories, consistent with the variability in $Br_y$ noted in Wang et al. (2015), and our hypothesis that sea-salt derived $Br_y$ is a source for BrO in the upper free troposphere downwind of marine convection.

[Figure]

***Figure 2.*** *TORERO AMAX-DOAS BrO (left) and IO (right) volume mixing ratio data derived by parameterization method (VMR$_{para}$) for the complete campaign. Plotted are 1 km altitude means with their 95% confidence interval as uncertainty in darker colors, while the lighter colored whiskers denote 5, 25, 75 and 95 percentiles.*

**References**

Dix, B., Koenig, T. K. and Volkamer, R.: Parameterization Retrieval of Trace Gas Volume Mixing Ratios from Airborne MAX-DOAS., Atmos. Meas. Tech. Discuss., 2016a.

Dix, B., Koenig, T. K. and Volkamer, R.: Parameterization Retrieval of Trace Gas Volume Mixing Ratios from Airborne MAX-DOAS., Atmos. Meas. Tech., in press, 2016b.

Schmidt, J. A., Jacob, D. J., Horowitz, H. M., Hu, L., Sherwen, T., Evans, M. J., Liang, Q., Suleiman, R. M., Oram, D. E., Le Breton, M., Percival, C. J., Wang, S., Dix, B. and Volkamer, R.: Modeling the observed tropospheric BrO background: Importance of multiphase chemistry and implications for ozone, OH, and mercury, J. Geophys. Res. - Atmos., 121(19), 11,819–11,835, 2016.

Stutz, J., Werner, B., Spolaor, M., Scalone, L., Festa, J., Tsai, C., Cheung, R., Colosimo, S. F., Tricoli, U., Raecke, R., Gao, R.-S., Hintsa, E. J., Elkins, J.W., Moore, F. L., Hossaini, R., Feng,W., Chipperfield, M. P., Daube, B., Pittman, J.,Wofsy, S., and Pfeilsticker, K.: A New Differential Optical Absorption Spectroscopy Instrument to Study Atmospheric Chemistry from an High-Altitude Unmanned Aircraft, Atmos. Meas. Tech. Discuss., 2016

Volkamer, R., Baidar, S., Campos, T. L., Coburn, S., DiGangi, J. P., Dix, B., Eloranta, E. W., Koenig, T. K., Morley, B., Ortega, I., Pierce, B. R., Reeves, M., Sinreich, R., Wang, S., Zondlo, M. A. and Romashkin, P. A.: Aircraft measurements of BrO, IO, glyoxal, NO2, H2O, O2–O2 and aerosol

extinction profiles in the tropics: comparison with aircraft-/ship-based in situ and lidar measurements, Atmos. Meas. Tech., 8(5), 2121–2148, 2015.

Wang, S., Schmidt, J. A., Baidar, S., Coburn, S., Dix, B., Koenig, T. K., Apel, E., Bowdalo, D., Campos, T. L., Eloranta, E., Evans, M. J., DiGangi, J. P., Zondlo, M. A., Gao, R.-S., Haggerty, J. A., Hall, S. R., Hornbrook, R. S., Jacob, D., Morley, B., Pierce, B., Reeves, M., Romashkin, P., Ter Schure, A. and Volkamer, R.: Active and widespread halogen chemistry in the tropical and subtropical free troposphere., Proc. Natl. Acad. Sci. U. S. A., 112(30), 9281–6, 2015.

---

## Referee Comment (RC2) · Anonymous Referee #2 · 13 Nov 2016

This study by Werner et al. presents DOAS measurements of bromine monoxide in the tropical and subtropical upper troposphere, tropopause region and lower stratosphere from the Global Hawk. These are important observations in a key region of the atmosphere. While the interpretaion of the DOAS observations, and in particular the applied O3 scaling technique, has to rely on a number of assumptions, this may be the best technique available to measure bromine monoxide in this important atmospheric region. However, I suggest that more details on this method and the uncertainty due to the assumptions made are given here, rather than refering to the companion paper by Stutz et al.

The paper would clearly benefit from some rearrangement of the presented material,

as further explained below. At parts also more detail is needed, as given in my specific comments below. With these modifications and after consideration of the other comments I recommend publication in Atmos. Chem. Phys.

Specific comments:

Abstract: The abstract is too long and should focus on the main findings.

p1, l1: why does this list here starts with CH4, O3 and NO2 ? I suggest using a similar statement as on page 4, where the list starts with BrO, as this is really the focus of this study.

p2, l2: split: "...LS. In the TTL ..."

p2, l9: how do these numbers relate to the reported range of BrO in the TTL?

p2, l12: top of TTL defined to be 425K in line 3

p2, l14: "chemical depletion": not clear what this means. 1/3 of observed global ozone trends (and if yes: over which period, which altitude region,...) or 1/3 of the chemical loss?

p2, l16: what does "mostly by natural and anthropogenic" mean? Are there other sources than natural and antropogenic? Or do you mean mostly by natural, but also some antropogenig sources?

p2, l31: Maybe useful to include a sentence or two on observations of BrONO2 in the stratosphere (e.g., Höpfner et al.)

p5, l1: a few lines below, the phrase "a large number of species, including O3, NO2 and BrO" is used, which may be appropriate here as well. I suggest to give this list of possible species only once (and consistent) and refer to it.

p5, l1: maybe better move long list of references into section 2.1

p7, l29: "The received limb radiances ...": need a few more words that this refers to the

mini-DOAS measurements. However, I suggest merging section 2.6 with 2.1, as this is an essential part of the mini-DOAS data analysis.

p8, l4: Strange sentence: "Demonstrates that Earth sphericity, ...are relevant". I suggest to rather say, "...are relevant and taken here into account."

p8, l23: "which together contain 1 ppt of bromine atoms" just repeats the first part of the sentence, or I don't understand what is meant here.

p8, l25: include explicitely that no sea salt aerosol source is assumed, in contrast to some other recent studies (e.g., Saiz-Lopez et al.)

p8, l28: "growth rate": why is the growth rate relevant? Because the CH4 content varies with age-of-air?

p9, l1: How? By changing the BrONO2 photolysis, by changing the rate of BrO + NO2, or both?

p9, l18: include "as well as" after the reference to Jensen et al.

p10, l1: closing bracked has to be after "tropical"

p10, l6+l7: "optical" -> "optimal"

p10, l32: I coulnd't easily find information on the integration time. Please organise the description in a way that all relevant information for the DOAS measurements can be easily found at one place of the manuscript. Currently this is distributed over Secs. 2.1, 2.6 and 3

p11, l1: Is really a higher spatial resolution required, or a finer temporal (or SZA) resolution?

Fig 7: Maybe it would make sense to indicate in the caption of Fig. 7 that DOAS data quality for SF4 is reduced?

Fig. 10: I assume for Fig. 10, model data are used as they are, i.e. not altitude

adjusted?

p12, l4: The agreement between measured and modeled NO2 is indeed strong evidence for the validity of the approach, but be careful with the reasoning: It is not possible to validate both measurements and model at the same time from this comparison.

p12, l7: better say "surface air mixing ratios"

p12, l9: "data is " -> "data are"

p12, l13: Not sure if you can draw this conclusion from this comparison. Does this not simply show that there is some spatial variability in CH2Br2 while the model assumes a constant mixing ratio at the surface? See next sentence.

p12, l16: The sentence should finish before "... to be implemented in the model". Whether or not this should be implemented in a model is a totally different issue.

p12, l27: I assume Wang et al and Volkamer et al use the same measurements, so better say "the TORERO measurements reported by Wang et al. and Volkamer et al."

p13, l3: I suggest to discuss similarities and differences to the results of Wang et al and Volkamer et al., but limit the speculations about possible discrepancies. My impression is that too much weight is given on explaining possible differences to the TORERO results, while other studies are not mentioned.

p13, l3: spell out the name of the aircraft

p13, l6: rephrase sentence, avoid the double use of "but"

p13, l6/7: This sentence does not contain any solid information and could be removed: It is trivial that any two measurements that are not performed at the same place at the same time could differ just by chance.

p13, l8-12: I find this statement problematic: What do you want to imply?

p13, l33: remove "Again"

p14, l3: remove "and others" - already contained in "e.g."

p14, l4: What does this mean: "with these features in mind"?

p14, l5: Probably misleading formulation: if you really know there is a bias of 2ppt you could correct for it.

p14, l6: "bottom to" -> "bottom of"

p14, l10: Not sure what you mean here. Increasing CH2Br2 in the model would be easy, and does not require a detailed back trajectory study. But would this "remove flight to flight scatter"?

p14, l14: "well be" -> "will be"

p14, l15: "gap in" or "gap between" ?

p16, l26: "climate is most sensitive": maybe better say more carefully "where ozone changes have the largest impact on radiative forcing"

p16, l33: "oxidizing capacity due to expected increase in VSLS emissions" is probably not what you mean. There are actually three possible processes at work: (1) changes in atmospheric transport, (2) changes in OH, affecting VSLS lifetimes and (3) changes in VSLS emissions due to aquaculture.

p17, l4: "some" -> "important" (?)

p17, l12: what kind of "adjustments" are performed here? Please give more details and justify!

p18, l3: Hossaini et al., 2016 is now published

p18, l9: remove extra "to"
* * *

---

## Author Comment (AC1) · 20 Dec 2016

**Our response/reactions are in black and comments of the reviewer in blue**

**General Comments**

The paper reports results from the ATTREX campaign performed in the Eastern Pacific during 2013, which probed the UT, TTL and LMS. The experimental results are of high importance for determining the overall fraction of inorganic bromine injected to the stratosphere, i.e. capable of destroying ozone in the LMS. BrO measurements are compared with modelled output from the TOMCAT/SLIMCAT model, which runs at high resolution with the specific meteorology (i.e. ECMWF Era-Interim reanalysis) present during the ATTREX campaign.

Even when the presented results possess a notable scientific impact as they probe inorganic bromine species in a region of major importance for constraining stratospheric bromine injection, the presentation of results is in this reviewer´s opinion rather disordered, and the description of the model configuration/output used to interpret their measurements is rather confusing. Therefore, several issues, both structural/technical and scientific/descriptive, must be dealt with before this MS is ready for publication. I recommend the following major comments be addressed before this paper can be accepted in ACP.

We are very grateful for the time the reviewer spent on reviewing the manuscript. The comments will help us to improve the manuscript. Please find below our point-to-point responses to the comments and queries.

**Major Comments:**
1. The abstract is too long and contains unnecessary information such as i) the instruments used for probing the composition of the tropical atmosphere, ii) a description of the minor deficiencies in the modelled transport, iii) a reference to a paper with the definition of the TTL. Also, the most important information in the abstract is quite disordered and the main results (i.e., measured numbers which are the major findings of this work) are only given in the very last lines.

We agree that the abstract was too long and shortened it by 8 lines. However, we disagree that it contains unnecessary information since the description of i) is relevant for the interpretation of our measurements, ii.) is honest in that the modelled curtains need a (joint) vertical adjustment in order to match the observations, iii.) since conflicting definitions of the TTL (in particular its bottom) appear in the literature.

2. The authors used a quite simple proxy to separate tropospheric air from stratospheric air: the condition of [CH4] being larger or smaller than 1790 ppb. This selection is neither justified nor referenced within the MS, and its use should be clearly justified as many of the forthcoming results depend on the validity of this assumption. Indeed, the histograms shown in Fig. 13 seem to contradict the validity of the CH4 proxy in splitting tropospheric from stratospheric air for theta > 390 K. Also, within the abstract and text, the CH4 condition is defined both respect to 1390 and 1790 ppb, introducing an additional inconsistency to the definition.

The [CH4] = 1390 ppb criteria were typos. They should have read [CH4] = 1790 ppb. We corrected these mistakes in the text. Our intent was not to separate tropospheric from stratospheric air since only air from subtropical LMS and TTL was sampled. The [CH4] >< 1790 ppb criterion is used to (roughly) separate younger air sampled within the TTL from relatively older air from the subtropical LMS. No other more robust method was available from the aircraft observations.

3. In the abstract and methodology, the authors declare that the TOMCAT/SLIMCAT model was "constrained to the measured O3 and NO2 and adjusted to match the observed concentrations of some brominated source gases" (P2,L6-7). But no detail is given in the methodology about how this

special configuration was applied in the model. Later, many of the results are interpreted and discussed based on the output obtained from the global model. In light of the importance of the observations and conclusions drawn here, mostly the inferred Bryinorg, the authors should describe specifically how the model was constrained or adjusted.

The method how to match the observed O3 (and CH4) is described I detail in Stutz et al., (2016). In short, the following: TOMCAT/ SLIMCAT has a height resolution of about 1 km in the TTL. The GH on the other hand, mostly flies at altitudes located between the model grid points. Therefore, an interpolation is necessary to account for sub-grid trace gas concentrations and to allow for an accurate comparison between the model results and the measurements. Moreover, in order to compensate for the models small (and varying) inaccuracies in describing the vertical air motion in the atmosphere (c.f., due to transient gravity waves), the modelled concentration curtains of all chemical tracers are jointly shifted in the vertical (by typically less than modelled vertical grid) until measured and modelled O3 agree. Once this vertical adjustment is performed (e.g., see Stutz et al., 2016, compare Figure 12 with 13), the measured and modelled CH4 (a transport tracer measured by UCATS and HUPCRS) and NO2 (a photochemical active species measured by the DOAS instrument) largely agree (see Figures 4 to 9). As a consequence, it can be concluded that (a) the vertical adjustment is reasonable and justified (from the agreement seen in CH4), (b) that the scaling method works well for NO2 and hence for BrO, and (c) that the adopted NOx/NOy photochemistry is able to reproduce measured NO2 (see Stutz et al., 2016).

We changed the following parts of the manuscript in order to clarify :

P1 line 16 cont:  Depending on the photochemical regime, the TOMCAT/SLIMCAT simulations tend to slightly under-predict measured BrO for large BrO concentrations, i.e. in the upper TTL and LS.

P11, line 16: The text now reads……  The panels (b), and (c) of Figs. 3 to 8 show comparisons of measured and modeled CH4, and O3 mixing ratios. Here the measured and modeled species reasonably agree within the given error bars, after the modeled curtains are altitude-adjusted by the same amount until measured and modelled O3 agree (for details see Stutz et al. (2016)). Noteworthy is, that in most cases the altitude adjustment is less than the grid spacing of TOMCAT/SLIMCAT (about 1 km in the TTL), thus mostly accounting for the altitude mismatches of the actual cruise altitude of the Global Hawk and the model output rather than deficits of the model to properly predict the vertical transport.

For example:
My major concern is about how removal/washout is considered in the model, mostly within the UT and TTL, as these processes will control the overall Bryinorg burden in that region of the atmosphere. The only reference to removal rates in the MS that I could found was on Page 15: L26 "…, and assuming no bromine is effectively lost in the troposphere, …" and L30-31 "Therefore effective loss processes for inorganic bromine, for example by heterogeneous uptake of inorganic bromine on aerosol and cloud particles, must act in the atmosphere". Detailed information on the removal processes considered here for brominated species should be given and also how they affect the BrO/Br ratios.

First, since the measurements were solely performed in the LMS and TTL, details in the budget and photochemistry of bromine in the troposphere only need to be considered in the model in as much as they are relevant to the budget of bromine in the TTL and LS. A description how the model is constrained to the budget of bromine in the troposphere and what processes are considered is given in section 2.7. In order to account for more complicated bromine sources (e.g., derived from sea salt and/or yet unknown minor organic brominated species), an additional 0.5 ppt of BrO is added to the near surface bromine, but only where TOMCAT/SLIMCAT predicts [BrO] < 0.5 ppt). Further, this modified BrO curtain is only used for the RT simulations, in order to better represent the BrO

absorption introduced by the small fraction of light traveling through the lower troposphere before being analysed by the limb oriented telescope on the GH. This approach is justified based on (a) the findings reported in Stutz et al., Figure 8b, and 15, (b) the findings of Volkamer et al., 2015 on BrO in the tropical tropics, (c) Dorf et al., 2008, and our earlier work (Harder et al., (1998), Fitzenberger et al., (2000), (d) Chen et al. (2016), and (e) our measurements around Borneo during SHIVA. This assumption also appears to be justified based on recent modelling studies which argues that sea salt derived bromine might be delivered to the TTL (e.g., Schmidt et al., JGR, 2016 see also below).

However, we re-call here again that in the measurement/model inter-comparison (sections 4.1, 4.2 and 4.4) measured/inferred quantities are directly compared with the TOMCAT/SLIMCAT model output, except!!! that all modelled curtains are vertically shifted by the same amount until measured and modelled $O_3$ agree. The latter is necessary since (a) to avoid interpolation errors between the actual measurements altitude, the measurements themselves, and the model predictions provided in discrete step which do necessarily correspond to the altitude of the measurements, and (b) to compensate for small mismatches in the actual and modelled vertical transport. Mostly (> 98%), the necessary vertical shifted than the vertical spacing of the model (~1 km).

Accordingly, and to account for the omitted sea-salt source (and in order to avoid any confusion) we (1) changed our text (page 8 line 25 cont.) to …

No other (cf., unknown organic or inorganic) sources of bromine for UT, LS, and TTL are assumed (e.g., Fitzenberger et al. (2000), Salawitch et al. (2010), Wang et al. (2015)). Omitting the release and heterogeneous processing of bromine from sea-salt aerosols (e.g., Saiz-Lopez et al. (2004)) in the model for the sake of saving computing time appears justified since (1) even though it is predicted to be relevant for bromine (~30% of the total $Br_y^{inorg}$) in the free troposphere (Schmidt et al. (2016)), its contribution to BrO in the TTL is at most of the order of the accuracy (~0.5 ppt) of our BrO measurements, (2) its time and space dependent sources (as for the brominated VSLS) are not well constrained, (3) in the modelled troposphere inorganic bromine only serve as boundary condition for bromine in the TTL, and (4) the additional BrO would not affect the BrO measurements-based calculation of $Br_y^{inorg}$ for the TTL (see below).

(2) added the following in the discussion (section 4.4, page 14, line 35 cont.)

This gap could partly be closed by adjusting the CH2Br2 surface concentration and atmospheric lifetime, or by considering a detailed scheme for dehalogenation of sea salt, i.e. bromine activation (e.g. Saiz-Lopez et al. (2004), Fernandez et al. (2014), Schmidt et al. (2016)). Adjusting CH2Br2 would add 0.4 ppt of $Br_y^{inorg}$, or ~ 0.3 ppt to BrO, thus removing the flight-to-flight scatter in source gas concentrations (~ 0.8 ppt) in $Br_y^{inorg}$. This could for example be done by a detailed back trajectory and source appointment analysis to which a forthcoming study will be devoted. Likewise, release of sea salt halogens to the gas-phase could add another 0.5 ppt to BrO (or about 0.7 ppt of $Br_y^{inorg}$) in the upper TTL (Schmidt et al., 2016).

and
(3) changed text in the introduction (page 2, line 11 cont).

…(3) so-called very short-lived species (VSLS), and (4) inorganic bromine transported into the upper troposphere, e.g. previously released from brominated VSLS and/or sea salt (e.g., Saiz-Lopez et al. (2004), Fernandez et al. (2014), Schmidt et al. (2016)). This inorganic bromine is also transported into the stratosphere.

(4) On page 14, line 23 cont.: Therefore, effective loss processes for inorganic bromine, for example

by heterogeneous uptake of inorganic bromine on aerosol and cloud particles, must act in the atmosphere (e.g., Schmidt et al. (2016)).

b. Also, In the model description (P8,L21-23), the surface concentration of VSL is 1.00 pptv for CHBr3, CH2Br2 and other VSLS. Why at (P12,L12) a value of 1.05 pptv is informed?. Further on, in the conclusions (P17,L15) the 1.0 ppt value is mentioned again. It may simply be a typo? Even when a 0.05 ppt value will not make a difference, this point should be made clear and consistent. How were the surface emissions adjusted?

Thanks for finding this error, we corrected the typos on P12 and P17.

4. In relation with my previous comment about an improved description of the specific model configuration used in this study, the authors state that "No other (c.f., unknown organic or inorganic) sources of bromine for UT, LS, and TTL are assumed (e.g., Fitzenberger et al. (2000), Salawitch et al. (2010), Wang et al. (2015), and others), except that we add 0.5 ppt to the modeled tropospheric BrO in agreement with the finding discussed below (section 4.6)". Later in Section 4.6, no specific mention is given about this additional source of BrO.

The reviewer is correct this was not clearly stated. As explained already above, only in the simulated curtains used for the RT calculations BrO was set to 0.5 ppt in the lower troposphere when the modelled predicted lower BrO. This was done in order to better represent the BrO absorption introduced by the small fraction of light traveling through the lower troposphere before being analysed by the limb oriented telescope on the GH (at 14 to 18 km) (see Figures 14 and 15 in Stutz et al., 2016).

We accordingly changed the text in section 2.6 (page 8, line10 cont.) to: "Only in the RT simulations [BrO] is set to 0.5 ppt near the ground, where TOMCAT/SLIMCAT predicted lower BrO concentrations (see Figure 2 middle right panel), in agreement with the findings discussed in Stutz et al. (2016) and the recent study of Schmidt et al. (2016). "

As BrO is used to constrain inorganic bromine using TOMCAT/SLIMCAT, a clear description of this additional source of BrO must be given in the text.
No, TOMCAT/SLIMCAT is freely run with the sources of bromine as given in section 2.7.
b. Yang et al., 2005 and Ordoñez et al. 2012 show that an additional source of inorganic bromine from sea-salt in the MBL is required to reproduce observations. Fernandez et al 2014 highlighted the importance of the sea-salt contribution for Bry in regions of strong convection such as the tropical western-pacific. Is this additional source of BrO in this work related to sea-salt recycling?, if so, is it constrained as a boundary condition or explicitly calculated?

Please see our response to previous questions

Please expand the discussion about this important omission. In Section 4.6 (P16,L10-13) the impact of sea-salt (or any other additional source) on Inferred Brytotal should also be discussed.

The recent modelling study by Schmidt et al., JGR (2016) indicates that 30% of inorganic bromine in the free troposphere might be due sea to salt. Their model also simulates [BrO] concentration of about 0.5 ppt in the free tropical troposphere. We find [BrO] < 0.5 ppt from the down-looking observations (Figure 15 in Stutz et al., 2016), and that between 0.5 ppt to 5.25 ppt (mean 2.63 +/- 1.04 ppt) of inorganic bromine is transported through the bottom of the TTL. Consequently, our conclusion on contribution 4 (which includes sea-salt derived bromine) in the tropical troposphere appears to be fairly reasonable. Since SLIMCAT/TOMCAT is a more stratospheric centric CTM, omitting details of heterogeneous and microphysical processes of sea-salt derived bromine in troposphere, i.e. treating it as a model boundary condition, is justifiable to save computer time. Finally, since in our assessment

on TTL bromine is based on measured BrO (CH4, O3, and NO2) and detailed photochemical modelling, no sizeable amount of bromine is omitted in the model, except that eventually some bromine (sub ppt, see below) might be tight to the TTL aerosol (which however is not really founded based on the findings of Murphy et al., 2016, (Halogen ions and NO+ in the mass spectra of aerosols in the upper troposphere and lower stratosphere, Geophys. Res. Lett., 27, 3217–3220, 2000 and our personal communication in 2000).

For the changes to the text see above.

5. In Section 2.1 (DOAS measurements of O3, NO2 and BrO), a companion paper (Stutz et al., 2016), describing the DOAS measuring technic used during the ATTREX campaign, is introduced. However, the current manuscript makes too many references to Tables, Figures and Sections within the Stutz et al. paper, which do not introduce additional clarification and in most cases difficult a direct reading of the main results. Please, revise the whole manuscript on this respect and keep the references to the Stutz et al., 2015 only when they are relevant for the results presented here.

In fact, we were trying to fit the entire material into a single manuscript, i.e. (a) a description of the instrument, (b) the spectral retrieval (DOAS), (c) the novel technique (scaling method) for the concentration retrieval, (d) the RT and CTM calculations and (e) the data interpretation. However combining (a) to (e) would have made a single manuscript too long and thus rather challenging to read. Consequently, we decided to split the study into two papers. We recommend to read the Stutz et al. (2016) paper in order to understand what information can be inferred from the measurements, and what information can not be obtained (c.f., averaged profiles of measured radicals, or a direct comparison to the Fernandez et al., 2014 and Saiz-Lopez and Fernandez, 2016 studies because they address a different atmospheric compartment or altitude range, see your comment below). Because the present manuscript often refers to the measurements, the underlying retrieval, and data interpretation, we feel it is rather necessary to frequently refer to the Stutz et al., (2016) manuscript. We would also like to add that the two manuscripts are linked to each other in the AMT/ACP environment and should be considered part 1 and 2 of the same study.

6. The results presented here somehow contrast with previously published measurements/modelling approaches. While the discussion respect to the findings from Wang et al., 2015 and Volkamer et al., 2015 are extensively discussed in section 4.3 (P13,L3-26), just a brief comment on the discrepancies respect to Fernandez et al., 2014 and Saiz-Lopez and Fernandez, 2016 is presented (Section 4.5, P15,L0-4).

The definition of the lower end of the TTL remains controversial, as already discussed above. We had several discussions with colleagues on this topic but no consensus could be reached. In fact, there is only little overlap (with respect to altitude, Θ, p ) between the BrO measurements of Wang et al., (2015) Volkamer et al., (2015) and modelling presented by Fernandez et al., (2014) and Saiz-Lopez and Fernandez (2016) (which according to our preferred TTL definition based on Fueglistaler et al. (2009) addressed more the bromine photochemistry in the upper tropical troposphere) and our study. Since two different regimes are described it is not necessary to address them in greater detail (except to compare Wang et al., (2015) and Volkamer et al., (2015) with our BrO profile (see Figure 3a).

With regards to the [Br]/[BrO] ratio, the common pattern shown in Figs. 4-9 is that whenever BrO mixing ratios decrease, both HBr and atomic Br increase.
Yes, this is a consequence of the adopted JPL bromine photochemistry in TTL and LS and decreasing O3 concentrations towards the (our) bottom of the TTL.

When this inversion is observed, the flight altitude and O3 levels also decrease, indicating that the lower TTL is being probed. In most cases HBr surpasses Br, while at some points (e.g., Fig 5f, 23:00) [Br] dominates. Thus, the Br and HBr prevalence seems to depend on the height (and possibly temperature,

not shown) at which the TTL is being probed. I would suggest further discussion and interpretation of those results. Particularly, if the absence of heterogeneous recycling is affecting the inferred BrO/Br ratio?.

We added two small paragraphs to clarify this issue.

P12, line 5 cont: Panels (e) in Figures 3 to 8 compare measured and modelled BrO. Again measured and modelled BrO mixing ratios reasonably compare for most of the flight, but sizable discrepancies are also discernible for some flight sections. Possible reasons for the latter, which are discussed in the following section, may be due to deficits in the models assumption of the sources of bromine (see section 4.2), and/or deficits in the adopted photochemistry (see section 4.4).

P14, line 5 cont: Overall this behavior is expected from arguments based on the amount and composition of the brominated organic and inorganic source gases, their lifetimes, atmospheric transport, and photochemistry (e.g., Fueglistaler et al. (2009), Aschmann et al. (2009), Hossaini et al. (2012b), Ashfold et al. (2012), WMO (2014), Fernandez et al. (2014), and Saiz-Lopez and Fernandez (2016), and others). In particular, for our measurements at daytime, it is observed that (a) BrO increases with $O3$ and available $Bry^{inorg}$ and thus altitude, (b) the predicted $BrO/Bry^{inorg}$ ratio decreases towards the bottom of the TTL, where (c) HBr and/or Br atoms may become as abundant as BrO, but HOBr does not play a major role in the $Bry^{inorg}$ partitioning. While observation (a) is primarily due to the increased destruction of the short-lived $Bry^{org}$ species and the efficient reaction of the released Br atoms with altitude dependent ozone concentrations, observations (b) and (c) are due to reactions of the Br atoms with CH2O, (and to a lesser extent H2O2) into HBr which is recycled to Br atoms through reactions with OH and on the available surface of aerosols and cloud particles, as predicted by Fernandez et al. (2014), and Saiz-Lopez and Fernandez (2016). Noteworthy is also the predicted minor role of HOBr formed by reactions of OH radicals with heterogeneously produced $Br_2$, or by the reaction HO2 + BrO. While the rate of the former reaction is small due to short photolytic life-time of Br2, the rate of latter reaction is small due to the small HOx concentrations in the TTL compared to photolysis of HOBr at daytime.

.

7. In section 4.1 the authors directly start showing results for Figs. 4-9. Not a single description is given to Fig. 3. Thus, Fig. 3 should be moved further down in the text.
We agree. Figure 3 is now moved to become Figure 12.

In P11,L17-20 you state that "The excellent agreement achieved between measured and modelled CH4, and O3 lends confidence that the altitude-adjusted TOMCAT/SLIMCAT model fields reproduce well the essential dynamical and photochemical processes of the probed air masses.". The authors can only assure a sentence like this by means of a full vertical profile validation of the species, not only comparing data at a specific level.

In the core we agree, but as matter of fact our observations are covering largely different altitudes (for example during the dives), photochemical regimes (the LMS, and TTL), SZA's ranging from 0° to 80° degrees et cetera that at least we were really surprised how well the measurement and the model agree. Accordingly, we replaced excellent with …..astonishingly good …. in the text.

8. The works possess several informal phrases which may not fit well within a scientific work. I list below some of them (but certainly not all), that maybe should be rephrased to a more appropriate structure, justified by numbers/references or removed from the MS.

Abstract, P2,L4: "… and the expectation based on the destruction of brominated gases."

We accordingly changed the phrase to …. and the expectation based on the photochemical destruction of the brominated source gases.

P4,L1: "… indicating that some Bry_inorg (i.e. several ppt) is directly transported from the tropopause"

We accordingly changed the phrase to … "… indicating that variable amounts of Bry$_{inorg}$ (i.e. several ppt) from the troposphere into the stratosphere"

P9,L17: "…, some exciting observations and details"

We accordingly changed the phrase to "…, some observations, their details as well as some results"

P11,L4: "… we refrain from this much more complicated approach"

We keep it since as conclusion (using a personal pronoun) the phrase appears to be justified.

P11,L27: "… NO2 concentrations meet the expectations for NOx"

We accordingly changed the phrase to …."… NO2 concentrations meet the expectations with respect to it's a partitioning and total NOx"

P16,L24: "By far …"

We accordingly erased …By far …"

9. There are several other sentences that may benefit from revision. I list some of them below:

P8,L22: [CHClBr2,CHCl2Br,CH2ClBr,…]. What the three dots means? Please specify.

In English grammar it is called an astrophic continuation, see https://en.wikipedia.org/wiki/Glossary_of_literary_terms i.e. it stands for other not considered minor brominated organic source gases. In fact the native speakers in list of authors had no objection to use it, but it could be necessary to contact an Anglist on this.

P12,L3: NOy=(NOx,N2O5,HNO3,HO2NO2,…. What the three dots means? Please specify.

See above. Other, not considered minor NOy species for example ClONOy and BrONOy.

P17,L14: Once again [CHClBr2,CHCl2Br,CH2ClBr,…]

See above. Other, other not considered minor brominated organic source gases.

P18,L3: (Here: cite Hossaini 2016, acp when published)?

Thanks. We updated the references to Hossaini et al. (2016).

**Figures Comments:**

Fig. 1: All results shown in this paper are for the ATTREX Flights performed during 2013 in the Eastern Pacific. Then why do you show panel A with the ATTREX 2014 flights that are not used here?

We accordingly changed the Figure to only display the 2013 flights.

Fig. 3: This Figure should be moved down in the text to make it consistent with the presentation of results. Also, it may be unified into a 2 panel figure, with only 1 caption that distinguishes between the a) and b) panels. The same applies for Fig. 13.

We moved both Figures to become the new Figures 12a and b, and we merged them into a two panel plot. We also merged Figure 13a and b.

Fig. 4: A portion of the caption is completely missing.

Figure 3. Panel (a) shows the time-altitude trajectory of the sunlit part of the GH flight track (SF1-2013) on Feb. 4/5, 2013 (SF1-2013). Panels (b)-(e) show inter-comparisons of TOMCAT/SLIMCAT-simulated fields with observations of (b) CH4 (UCATS), (c) O3 (NOAA), (d) NO2 (mini-DOAS), and (e) BrO (mini-DOAS). The grey-shaded error bars of the mini-DOAS NO2 and BrO measurements includes all dominating errors, i.e. the spectral retrieval error, the overhead and the error due to a tropospheric contribution to the slant absorption, and the absorption cross section uncertainty. Panel (f) shows the SLIMCAT modeled Bry partitioning for the standard run #583. Panel (g) shows a comparison of inferred and modeled Bry$^{inorg}$, including the errors as a grey band. The dashed vertical lines in Figures 4-9 separate different atmospheric regimes: (I) is the extra-tropical lowermost stratosphere, (IIa, IIb, ..) different mixing regimes of air from the extra-tropical lowermost stratosphere, and (III) from the tropical tropopause layer.

This should be carefully controlled before submission. Also, following the equivalent figures for the rest of the flights (Figs. 5-9), no mention is given to what the vertical dashed lines represent. In addition, it would be very helpful to show a constant (dashed) line for O3 = 150 ppb and CH4 = 1790 pptv to distinguish the periods where the subtropical and TTL air was being probed.

Wouldn't it be a good idea to show the "mean temporal profile" measured by all of the ATTREX flights (i.e., for equivalent SZA)??

We thought about it and came to the conclusion that this is not really a good idea because the amount of BrO would also depend on total $Br_y^{inorg}$, O3, NO2, ….

Fig. 13: The large values of BrO at theta = 390-400 K within the freshly ventilated TTL (BrO > 7.5 pptv) shown during flight SF2 makes me doubt about the ability of the the CH4 < 1790 pptv as a good proxy for distinguishing stratospheric air.

Again it tentatively separates LMS from TTL air, for which unfortunately no strict horizontal barrier or criteria exists to distinguish both compartments.

Also, What is the meaning of showing Fig. 13b., for which most of the panels show no data at all??

Inferred total $Br_y^{inorg}$ in the LMS. Unfortunately, due to operational constraints no measurements are available for Θ < 380 K.

Fig. 14: Flights SF2 and SF4 are not shown. Why not?.

For SF2, GWAS did not measure VSL bromine and for SF4 a DOAS retrieval problem exists (see text).

(P14, L5) It would be a good idea to show the mean results for the campaign, and then highlight results for each of the flights.

Based on our experience we think averaging is often not a good idea. In fact, we provided comparable results once (for IO), which was then incorrectly used (by not properly considering the actual photolysis frequency during the measurements) in follow-on studies.

Fig. 11: I was surprised about the dispersion on the modelled/measured scatter plot for CH3Br. Being the bromo-carbon with largest lifetime, I would expect it to have an equivalent dispersion to the halons. No mention is given about this issue in the text.

Probably your surprise calms considered that the data range from 6 to 7.5 ppt for CH3Br and from 3 to 4 for Halon 1211, i.e. due to the scaling where the point (0,0) is truncated.

**Additional Scientific Comments:**

P2,L15: Is there a direct assignation of WMO, 2014 values specifically to year 2011?

It is assigned to year 2013, since only publications until 2013 are included in the report.

P3,L1: From which reference/s do you obtain the (2.5-4) ppt error on Bry inorg uncertainties within the inorganic method?

From our own experience (+/-2.5 ppt) and published literature (+/-4 ppt) (c.f., Parella et al., 2013)

P3,L4: UT, TTL and LMS should be defined independently in the abstract and introduction. There is no sense to introduce exactly which definition of the TTL you used in the abstract/introduction.

For the definition of the UT and TTL see the off-line discussion in which you were included. Even more complicated is the separation of the TTL and LMS since there exists no fixed barrier, even not with respect to PV. Since no dynamical variable were measured from aboard the GH, we decided to use the methane >< 1790 pptv criterion to separate both air masses as act of necessity.

P3,L32: Dorf et al., 2008 reported a contribution from VSL of (4.0 ± 2.5) for PG and (5.2 ± 2.5) for total (SG+PG). I do not find where you get the value of (2.5 ± 2.6) ppt. Also, there seems to be a confusion in the interpretation of contributions 3 and 4) from Dorf et al.

We checked our files, and accordingly changed to sentence to "The most direct information on contributions 3 and 4 came by the studies of Dorf et al. (2008), WMO (2011), and Brinckmann et al. (2012). They inferred 1.25± 0.16 ppt (VSLS-SGs, contribution 3) + 4.0±2.5 ppt (PGs, contribution 4) = 5.25±2.5 ppt (contributions 3 and 4) and 2.25±0.24 ppt (VSLS-SGs, contribution 3) + 1.68±2.5 ppt (PGs, contribution 4) = (3.98±2.5) ppt (contributions 3 and 4) from two balloon- borne soundings performed in the TTL and stratosphere over north-eastern Brazil during the dry season in 2005, and 2008, respectively."

P4,L4: Saiz-Lopez et al., 2012 also estimated the climatic impact of VSL sources within a CCM.

We added the reference.

P7,L10: Please describe which species were measured by the GWAS sampler, and which ones were used within this work.

We added a paragraph on P7, L7cont.

P7,L27: "The received limb radiances …". Do you mean the limb radiances measured with the mini-DOAS instrument? Please specify.

We accordingly changed the text to… The measured limb radiances of the mini-DOAS instrument…

P9,L3: Other works used this type of experimental data measurements on top of a model "curtains" (i.e Nicely et al., 2016). Some reference to any of these articles could be given here.

Using predicted curtains in the interpretation retrieval of UV/vis remote sensing data is not new at all. For example, we used it already in 2000 in the Harder et al., study (Figure 1), and later in studies, Weidner et al., 2005, Dorf et al., 2006….. and of course the IR community uses modelled curtains to forward guess their observations since the early 1990s.

P9:L20: What do you mean by "the NASA-ATTREX flights of the Global Hawk were strongly biased with respect to the sampled air masses". Please clarify.

The primary focus of NASA ATTREX mission was to study aerosol and clouds within the TTL (Jensen et al. 2015), and not to probe air of the subtropical LMS. So the flights went from EAFB base straight to the predicted coldest spot in the TTL on the highest possible cruise altitude (also because of air safety reasons, without losing fuel in performing dives in between at daytime). In addition, the GH has only two gears, either full thrust or no thrust on the engine (the latter used to perform the dive)

P11,L27-30: Please specify for which Flights the values for NO2 range between (70-170) ppt and were interpreted as belonging to the LMS.

For the separation of the LMS and TTL see above. The reminder is inferred from the data.

P12,L9: What do you mean by "even if the data is scattered from flight to flight"?

During the flight and from sortie to sortie the measured and modelled source gas data agree fairly well, but not perfectly.

P12,L17: Ordoñez et al. 2012 also describes the geographical and temporal variability of oceanic VSL sources.

….which was not implemented in the present simulation, primarily due to reasons given above (replies to queries 3 and 4). Also the Ordoñez et al. 2012 emission scenario is not undisputed for all VSLS species (see Hossaini et al., 2016).

P14,L6-7: Please explain better what you mean by "very young air" and "older air". Are you considering the [CH4] proxy given before?

We accordingly changed the text to ….. or comparable younger air based on measured [CH4].

P15,L5-7: A printed value of the overall Bry$_{inorg}$ error (and range) derived from the analysis shown on panel f of Fig. 4-9 would be helpful in the text.

Unfortunately, this poses a problem since the error changes from observation to observation due to the factors mentioned in the text. So addressing them in text would require an extended table. Individual errors bars are however given in Figure 14, i.e. for flight section when the WAS instrument sampled air for the brominated organic species.

P15,L22-26: All the analysis about the Bry$_{inorg}$ increment due to the decrease in VSLS is performed based on the "theoretical" dashed diagonal lines shown in Fig. 15. But the "modelled" diagonal lines are not shown for neither of the flights. Why? Including the modelled lines would strengthen the analysis, and justify the conclusions obtained here. Also, what do you mean by "…extrapolating the data points along lines of constant [VSLS]+[Bry$_{inorg}$] (…) to [Bry$_{inorg}$] = 0"? Do you mean getting the intercept of each of the (not shown) lines for each flight?

Naively speaking yes. However, the plot expresses something like a 'thought experiment' in that (by incorrectly) assuming that if there were no loss of bromine in the troposphere (which we know is not true), then near surface air would at least need to contain the amount of VSL bromine as indicated by the interpolation to [Bry$_{inorg}$] = 0. Such a thought experiment is justified based on that (a) near surface VSL bromine measurements are much more frequent than VSLS plus inorganic bromine measurements (the latter virtually do not existing).

P16,L12: It is not clear how the range (0.5 to 5.25) pptv or uncertainty (± 1.04 pptv) of inferred Bry$_{inorg}$ values is computed. Are these the maximum-minimum values modelled with SLIMCAT for all flights? Is this a model average within the Eastern Pacific region where measurements took place?

Yes, the range (0.5 to 5.25) ppt spans the minimum and maximum Bry$_{inorg}$, and the (2.63± 1.04 pptv) is the average over all inferred Bry$_{inorg}$ at the bottom of TTL (according to our preferred definition). By

the way, the mean amount of inferred $Bry_{inorg}$ compares very well with the model results (at 17 km) mentioned in Navarro et al, 2015 for the Eastern Pacific (i.e. compare Figure 13a the panel for 370 to 380 K results with concentration given in Navarro et al., 2015, Table 1 for the Eastern Pacific, i.e. 5.98 (VSL + $Bry_{inorg}$) minus 2.96 ppt (VSLS) versus 3.1+/-1.28 ppt (see Figure 13a) of the presented study.

P16,L19: The chemical loss rates were computed within the tropics (i.e, considering the 0º - 360º longitudes) or only within the Eastern Pacific region? It is important to make this clear, and in case of considering the whole tropics, a comparison to the values obtained within the EP is needed.

We re-run the model and accordingly changed first the sentence in the introduction to ….. Finally, for the Eastern Pacific (170_W - 90_W) the TOMCAT/SLIMCAT simulations indicate a net loss of ozone of - 0.3 ppbv/day at the base of the TTL (Θ = 355 K) and a net production of + 1.8 ppbv/day in the upper part (Θ = 383 K)……and second in section 4.7 to….. The chemical rates are averaged over the Eastern Pacific region (20º S - 20ºN, 170º W - 90º W) for the duration of the campaign.

P16,L22-27: The chemical loss analysis gets into details of which independent families and specific channels are the major contributor to the total (or bromine) ozone loss rate. I suggest including a table/sentence defining all the families considered, and which reactions are considered at least for the bromine channel.

We accordingly included an annex explaining which reactions where considered in the ozone loss calculations.

P17,L9: I do not agree that your results provide confidence in the modelled NOy photochemistry. You have only shown results for NO2 in this work. Much deeper analysis of nitrogen cycles should be given in order to perform this statement within the conclusions. Please see also text in P12,L3-4.

We partly concur based on that other NOy species were not measured! However, the NOy species would need extra-long atmospheric life-times since the shorter lived NOy species (N2O5, HO2NO2, HONO, ClONO2, BrONO2) are readily photolysed in the TTL at daytime, and would add to NOx, and ultimately to NO2 (being in steady state with NO). Even a longer lived NOy species, such as HNO3 need to be correctly modelled, because its life-time is certainly shorter than that of CH4. If not true, it would it again add to NOx.

**Technical/Linguistic Comments:**

P2,L11: It is not necessary to include a minus sign whenever you state that the value represent a net destruction. There are many other places in the MS when this also occurs.

Yes but nevertheless correct.

P2,L15: Within the text and figures, the terms Bry and $Bry_{Total}$ are both used to represent the same quantity. Please unify the criterion.

We changed $Bry^{Total}$ to Bry where necessary (cf. Figure 14).

P2,L17. All halons must be written either with capital H or lower case h.

We changed all halon to Halon.

P2,L32: … several tenths of ppt.

We accordingly changed the text.

P3,L15: Colombia, not Columbia

We accordingly changed the text.

P5,L29: UAS acronym is not used at all in the paper, there is no sense to define it. Also, if the size is in Liters, then capital L should be used.

We accordingly changed the text.

P13, L3-7: Consider rephrasing.

We changed to text to…. Second, even though Wang et al. (2015) used a technique similar to ours and in particular they use the same radiative transfer code (e.g., McArtim see Deutschmann et al. (2011)), they inferred the BrO profiles using the optimal estimation technique (Volkamer et al., 2015).

P14,L17: What about BrCl? It is shown in Figs. 4-9 but not included in the definition of Bryinorg?

We accordingly changed the text.

P14,L19-20: Have you thought on including in a table the most important reactions that were changed between the three sensitivity runs?

No since the reaction are well known and are already tabulated in the JPL compilations.

P15,L35: Is the call to Fig. 15. correct?. If so, please explain.

Yes it is correct.

P17,L6: do you mean anti-correlated?

We accordingly changed the text.

Fig. 2: The altitude range should be given in between brackets.

We accordingly changed the text.

Fig. 15: Consider rephrasing the sentence starting with: "If all Bry …" for one in the form "The dashed diagonal lines indicate …"

We changed the text to "If all Bryinorg resulted from destroyed VSL bromine of the same air mass from near the surface, then all data points should follow individual diagonal lines."

In the acknowledgments, we added the following sentence: The authors are grateful for the comments given by two anonymous reviewers, and the comments of Barbara Dix and Rainer Volkamer (CU, Boulder, USA).

---

## Author Comment (AC2) · 20 Dec 2016

**Comments of the reviewer are given in black and our responses in blue**

We are grateful for the very detailed comments of the reviewer which certainly helped to improve the manuscript. Please see our detailed response below.

This study by Werner et al. presents DOAS measurements of bromine monoxide in the tropical and subtropical upper troposphere, tropopause region and lower stratosphere from the Global Hawk. These are important observations in a key region of the atmosphere. While the interpretation of the DOAS observations, and in particular the applied O3 scaling technique, has to rely on a number of assumptions, this may be the best technique available to measure bromine monoxide in this important atmospheric region. However, I suggest that more details on this method and the uncertainty due to the assumptions made are given here, rather than referring to the companion paper by Stutz et al.

In fact, in the first draft to report on the study, we tried to merge the contents of the Stutz et al., (2016) (http://www.atmos-meas-tech-discuss.net/amt-2016-251/ ) and the present manuscript. It turned out however, that reporting on (1) the new instrument, (2) the spectral retrieval, (3) a novel method to invert slant column amounts into absolute concentrations including sensitivity runs (see the supplement in Stutz et al., (2016), and (4) reporting on the bromine results would result in a manuscript in excess of 100 pages. So we decided to split it into two manuscripts, which unfortunately now requires some cross referencing. For the assumptions made in the scaling method and sensitivity runs aiming to test these assumptions see the manuscript of Stutz et al. (2016). If you are interesting in getting more insight into the novel scaling method and its sensitivity towards assumptions (e.g., mostly due to the relevant RT), we may additionally provide digital copies of the master theses of Raecke (2013), and Knecht (2015), the recent PhD theses of Werner (2015) and Hüneke (2016), as well as 4 recent bachelor theses, in which more facets and details of the novel scaling method and its sensitivity towards inevitable assumptions are investigated and discussed. For the reference see below.

The paper would clearly benefit from some rearrangement of the presented material, as further explained below.
Following the recommendations of reviewer 1, we rearrange the Figures, and added some text where necessary (see also below).

At parts also more detail is needed, as given in my specific comments below. With these modifications and after consideration of the other comments I recommend publication in Atmos. Chem. Phys.

Specific comments:
Abstract: The abstract is too long and should focus on the main findings.
We condensed the abstract by 8 lines.

p1, l1: why does this list here starts with CH4, O3 and NO2? I suggest using a similar statement as on page 4, where the list starts with BrO, as this is really the focus of this study.
The list somehow has to start with a given order of gases, since it is necessary to first understand changes in measured gases due to dynamics (CH4, O3), remaining uncertainties in method (comparison of measured and modelled NO2), before the photochemistry and budget of bromine in the LS and TTL can be discussed.

p2, l2: split: "...LS. In the TTL ..."
We accordingly changed the text.
p2, l9: how do these numbers relate to the reported range of BrO in the TTL?
Excellently. In fact as the text state, $[Br_y^{inorg}]$ is calculated from measured BrO and the modelled $[Br_y^{inorg}]/[BrO]$ (>1) ratio.
p2, l12: top of TTL defined to be 425K in line 3
We accordingly changed the text (…in the upper part…)
p2, l14: "chemical depletion": not clear what this means. 1/3 of observed global ozone
trends (and if yes: over which period, which altitude region,...) or 1/3 of the chemical loss?
We accordingly changed the text (…global photochemical loss…)
p2, l16: what does "mostly by natural and anthropogenic" mean? Are there other
sources than natural and anthropogenic? Or do you mean mostly by natural, but also
some anthropogenic sources?
We accordingly changed the text (erased ' mostly')
p2, l31: Maybe useful to include a sentence or two on observations of BrONO2 in the stratosphere
(e.g., Höpfner et al.)
Thanks, we accordingly added a sentence (Further constraints on stratospheric Bry (range 20 - 25
ppt) were obtained by satellite-borne measurements of BrONO2 in the mid-IR spectral range at
nighttime (Höpfner et al., 2009).
p5, l1: a few lines below, the phrase "a large number of species, including O3, NO2 and BrO" is used,
which may be appropriate here as well. I suggest to give this list of possible species only once (and
consistent) and refer to it.
We accordingly changed the text (….beside for some other species, see above)….)
p5, l1: maybe better move long list of references into section 2.1
We keep it.
p7, l29: "The received limb radiances ...": need a few more words that this refers to the mini-DOAS
measurements. However, I suggest merging section 2.6 with 2.1, as this is an essential part of the
mini-DOAS data analysis.
We accordingly changed the text (The measured limb radiances of the mini-DOAS instrument….) but
we kept section 2.6, since it describes the tool (together with the modelled curtains) used in the
interpretation of our measurements (section 2.1), rather than being a part of the DOAS method
description.
p8, l4: Strange sentence: "Demonstrates that Earth sphericity, ...are relevant". I suggest to rather say,
"...are relevant and taken here into account."
Thanks, we accordingly changed the text to ….The simulation indicates that correctly accounting for
the Earth's sphericity, the atmospheric refraction, cloud cover, ground albedo etc. is relevant for the
interpretation of UV/vis/near-IR limb measurements performed within the middle atmosphere
(Deutschmann et al., 2011).
p8, l23: "which together contain 1 ppt of bromine atoms" just repeats the first part of
the sentence, or I don't understand what is meant here.
We accordingly changed the text, i.e., erased ( … , which together contain 1~ppt of bromine atoms.)
p8, l25: include explicitly that no sea salt aerosol source is assumed, in contrast to some other
recent studies (e.g., Saiz-Lopez et al.)
Thanks a lot. We accordingly changed the text (see our detailed responses to reviewer 1)
p8, l28: "growth rate": why is the growth rate relevant? Because the CH4 content
varies with age-of-air?
Yes, since present TOMCAT/SLIMCAT run is integrated from 1979 onwards, see P8, L27 of the revised
manuscript.
p9, l1: How? By changing the BrONO2 photolysis, by changing the rate of BrO + NO2,
or both?
By scaling k(BrO+NO2).
p9, l18: include "as well as" after the reference to Jensen et al.

We accordingly changed the text.

p10, l1: closing bracked has to be after "tropical"

We accordingly changed the text.

p10, l6+l7: "optical" -> "optimal"

We accordingly changed the text.

p10, l32: I couldn't easily find information on the integration time. Please organise the description in a way that all relevant information for the DOAS measurements can be easily found at one place of the manuscript. Currently this is distributed over Secs. 2.1, 2.6 and 3

On page 9, line 3 of the original manuscript is was already written (For all model levels and for the time resolution (~30 s) of the mini-DOAS measurements….). We accordingly changed the text (P9, l31) to "longer signal integration times (than the standard integration time which is 30 s), and are thus averaged "

p11, l1: Is really a higher spatial resolution required, or a finer temporal (or SZA) resolution?

A high spatial resolution is required, since the instrument probed air masses 1000 s ahead the aircraft (which for a cruise speed of 200 m/s corresponds to 200 km, see Stutz et al. 2016, Figure 10, which corresponds to a $\Delta SZA = 0.2°$ during setting sun) and the time resolution of TOMCAT/SLIMCAT is already 30 s (or in terms of distance about 6 km).

Fig 7: Maybe it would make sense to indicate in the caption of Fig. 7 that DOAS data quality for SF4 is reduced?

We added the following to the legend of Figure 6 in the revised manuscript: Note that DOAS analysis of BrO for SF4-2013 is somewhat uncertain because the Fraunhofer reference spectra (taken via a diffuser) are affected by temporally changing residual structures likely due to ice deposits or some other residues on the zenith diffuser (see text).

Fig. 10: I assume for Fig. 10, model data are used as they are, i.e. not altitude adjusted?

All measured data are compared to the modelled data assuming the same altitude interpolation, i.e. all curtains are altitude shifted by the same amount until measured and modelled $O_3$ agree. Please note that the altitude shift is typically smaller than the vertical resolution of the model, which is about 1 km in the TTL.

p12, l4: The agreement between measured and modeled NO2 is indeed strong evidence for the validity of the approach, but be careful with the reasoning: It is not possible to validate both measurements and model at the same time from this comparison.

We agree, but since in-situ measured and modelled $CH_4$ is also found to largely agree (once all curtains are vertically shifted until measured and modelled $CH_4$ agree) a dynamical reason for an assumed fortuitous agreement can be ruled out. Further since $NO_2$ is a photochemical active species and air masses of quite different $NO_2$ mixing ratios were probed, an unrecoginzed major issue with the scaling method and/or adopted NOx photochemistry is highly unlikley.

p12, l7: better say "surface air mixing ratios"

We accordingly changed the text.

p12, l9: "data is " -> "data are"

We accordingly changed the text.

p12, l13: Not sure if you can draw this conclusion from this comparison. Does this not simply show that there is some spatial variability in CH2Br2 while the model assumes a constant mixing ratio at the surface? See next sentence.

You are right and we accordingly changed the text to… This is most likely due to an underestimation of the surface concentration (1 ppt), variable mixing ratios at the surface not correctly considered in the model, and/or errors in the atmospheric lifetime due to reactions of CH2Br2 with OH radicals in the model (e.g. Mellouki et al. (1992), Ko et al. (2013), WMO (2014)).

p12, l16: The sentence should finish before "... to be implemented in the model". Whether or not this should be implemented in a model is a totally different issue.

We accordingly changed the text.

p12, l27: I assume Wang et al and Volkamer et al use the same measurements, so better say "the TORERO measurements reported by Wang et al. and Volkamer et al."

We accordingly changed the text.

p13, l3: I suggest to discuss similarities and differences to the results of Wang et al and Volkamer et al., but limit the speculations about possible discrepancies.

We accordingly changed the text.

My impression is that too much weight is given on explaining possible differences to the TORERO results, while other studies are not mentioned.

We are not sure about this regarding (1) our concerns towards TORERO BrO profiles reported in Wang et al., (2016), (2) our findings on BrO in the TTL reported in the present study (3) previous BrO measurements in the TTL (Dorf et al. 2008), (4) not yet published BrO data (e.g., collected during different HALO missions during the past 4 years), and (5) the comments of Dix&Volkamer to the present manuscript.

p13, l3: spell out the name of the aircraft

We accordingly changed the text.

p13, l6: rephrase sentence, avoid the double use of "but"

We accordingly changed the text.

p13, l6/7: This sentence does not contain any solid information and could be removed:

It is trivial that any two measurements that are not performed at the same place at the same time could differ just by chance.

We accordingly changed the text (erased the sentence).

p13, l8-12: I find this statement problematic: What do you want to imply?

We accordingly changed the text to …. they inferred the BrO profiles using the optimal estimation technique with constraints based on measured O2-O2, in-situ measured aerosol parameters and remotely measured Mie extinction Volkamer et al. (2015). However, we strongly feel that a 1-D treatment of the RT used in the interpretation of UV/vis/NIR limb measurements is not justified, if OE is used for the mathematical inversion (see also our response to the comments of Dix&Volkamer).

We also changed to text (P13, L29 -34). In conclusion, even though the reported TORERO flights 12 and 17 were performed under clear-skies (Volkamer et al.,2015), the extent to which the 2-D (and under cloudy sky 3-D) dimensionality of the underlying radiative transfer problem, in particular relevant when using optimal estimation for profile inversion, impacts the results.  In particular, unaccounted scattering due to aerosols and (probably) optically thin upper tropospheric clouds, lower level clouds, or changing overhead stratospheric BrO could have contributed to the reported elevated BrO in the UT, and around the bottom of the TTL.

p13, l33: remove "Again"

We accordingly changed the text.

p14, l3: remove "and others" - already contained in "e.g."

We accordingly changed the text.

p14, l4: What does this mean: "with these features in mind"?

We accordingly changed the text (we erased "with these features in mind"?).

p14, l5: Probably misleading formulation: if you really know there is a bias of 2ppt you could correct for it.

Essentially you are right, but without a physically consistent explanation for the processes, we feel a correction is inappropiate. Consequently we leave this formaulation as is.

p14, l6: "bottom to" -> "bottom of"

We accordingly changed the text.

p14, l10: Not sure what you mean here. Increasing CH2Br2 in the model would be easy, and does not require a detailed back trajectory study.

Yes, for the constant low bias in assumed CH2Br2. Adjusting it in the model would need another, computer intensive, model run from 1979 onwards. However, it is fair to assume that Bry levels would linearly follow the assumed CH2Br2, so this effort is not really necessary.

This would likely not "remove flight to flight scatter". However, including a  more detailed source parameterization (i.e., time and space dependent) likely would.

p14, l14: "well be" -> "will be"

We accordingly changed the text.

p14, l15: "gap in" or "gap between" ?

We checked the correct usage. Both versions are possible.

p16, l26: "climate is most sensitive": maybe better say more carefully "where ozone changes have the largest impact on radiative forcing"

We accordingly changed the text.

p16, l33: "oxidizing capacity due to expected increase in VSLS emissions" is probably not what you mean. There are actually three possible processes at work: (1) changes in atmospheric transport, (2) changes in OH, affecting VSLS lifetimes and (3) changes in VSLS emissions due to aquaculture.

We accordingly changed the text.

p17, l4: "some" -> "important" (?)

We accordingly changed the text.

p17, l12: what kind of "adjustments" are performed here? Please give more details and justify!

We accordingly changed the text to … The measured and modeled TTL concentrations of CH2Br2, CHBr3 are found to compare reasonably well to the surface concentrations and atmospheric life times of both species adopted in the model.

p18, l3: Hossaini et al., 2016 is now published

We accordingly changed the text.

p18, l9: remove extra "to"

We accordingly changed the text.

In the acknowledgment, we added the following sentence: The authors are grateful to the comments given by two anonymous reviewers, and the comments of Barbara Dix and Rainer Volkamer (CU, Boulder, USA).

Additional references:

1. Knecht, M.: Simulation of radiative field modification due to tropical clouds, Master thesis, Institut für Umweltphysik, Universität of Heidelberg, Heidelberg, Germany, 2015.

2. Hueneke, T., The scaling method applied to HALO measurements: Inferring absolute trace gas concentrations from airborne limb spectroscopy under all sky conditions, PhD thesis, University of Heidelberg, Heidelberg, Germany, 2016.

3. Raecke R., Atmospheric Spectroscopy of Trace Gases and Water Vapor in the Tropical Tropopause Layer from the NASA Global Hawk, Master thesis, Institut für Umweltphysik, Universität of Heidelberg, Heidelberg, Germany, 2012.

4. Werner, B., Spectroscopic UV/vis limb measurements from aboard the NASA Global Hawk: Implications for the photochemistry and budget of bromine in the tropical tropopause layer, PhD thesis, University of Heidelberg, Heidelberg, Germany, 2016.

---

## Author Comment (AC3) · 20 Dec 2016

Response (in red) to the comments of Barbara Dix, and Rainer Volkamer

We appreciate the response and the clarification on the results reported by Wang et al., and Volkamer et al., (2015). It was not our intent to question these results, but rather to put them in context to our findings, both from the methodology standpoint as well as with respect to the final results. Ultimately we believe that the disagreement between the observations motivates further study of bromine chemistry in the UTLS as well as more effort in ensuring the accuracy of high-altitude limb-scanning DOAS observations.

We have reformulated the respective paragraphs in the manuscript to avoid any a misunderstanding of our intentions:

*"It is possible that the TORERO observations Wang et al. (2015) and Volkamer et al. (2015) off the western coasts of South and Central America, i.e. further south than the ATTREX region but during the same season, encountered an unusual meteorological situation that would have caused downward transport of bromine rich air from the lower stratosphere to the UT and the bottom of the TTL (up to about 14 km), or that sea salt released bromine played a role (e.g., Schmidt et al. (2016)). However, our study has identified possible problems when using optimal estimation technique with constraints based for example on measured O2-O2 for high altitude aircraft limb observations. The RT below the aircraft and in particular in the lower troposphere plays a crucial role for the observations, due to the much higher O2-O2 concentrations. Also since individual limb measurements already cover an area of typical 200 x 20 km in front of the aircraft (see Figure 5 in Stutz et al. (2016)), and even more crucial when applying optimal estimation for profile inversion a series of measurements taken during the ascent and descent of the GH are jointly inverted. Hence the radiative field and its time dependence needs to be known over a larger food-print (i.e., the RT is 2-D, or even 3-D plus its time dependence over the period of a single profile measurement)."*

In the following we will provide some brief thoughts on the comments by Volkamer and Dix (in red).

We generally agree that "skylight analyzed for the O2-O2 absorption in limb direction may carry additional, or even predominantly information on the radiative transfer of lower atmospheric layers", but want to point out that BrO profiles published in Volkamer et al. (2015) and Wang et al (2015) were neither affected by underneath cloud cover, nor by cirrus above. Sections 2.10 and 3.1 in Volkamer et al. (2015) discuss explicitly the effect of aerosol and clouds, and make fully transparent that the presented RF12 and RF17 case studies are not affected by clouds. Furthermore, we show below HSRL data from these two flights (Fig. 1) that make transparent that no aerosols or thin cirrus layers were present above the aircraft.

It is our experience that cloud free conditions for the geometry of a limb-DOAS system, i.e. up to 200 km ahead and 20 km on the side, is quite rare, especially in the tropics and sub-tropics. In addition, the interpretation of the limb-observations requires 2-D (or even 3-D) radiative transfer calculation and information on the spatial 3D distribution of atmospheric scatters (e.g., Oikarinen, 2002; Figures 5 and 10 in Stutz et al., 2016; Raecke, 2013 see the Figure 1 provided below). In addition, radiative transfer condition can change during ascent or decent manoeuvres of an aircraft, which add another degree of complexity. Because, we did not have this information and clouds were nearly always present during ATTREX, we had to rely on a scaling technique with a trace gas that has a similar vertical distribution as BrO, i.e. ozone, to overcome the challenges of this radiative transfer challenge.
We agree with the comment that under cloud free conditions the combined radiative transfer and optimal estimation approach to retrieve vertical trace gas profiles should give reliable results.

Moreover, the authors are referred to Fig. 2 and Section 2.1 in Dix et al. (2016a), where it is shown that in cloud-free conditions measurements of O2-O2 are suitable to constrain RTM up to 15 km. The

statement by the authors is too broad, and certainly does not apply to the Wang et al. and Volkamer et al. case studies. This should be corrected. Figure 1. Comparison of O4 ratios at 360 nm and 477 nm with HSRL particulate backscatter cross section data for TORERO RF04, RF12 and RF17 (Dix et al. (2016a); Volkamer et al. (2015); Wang et al. (2015)). Altitude resolved HSRL backscatter data is plotted and color coded along the flight track. Larger signals denote the presence of aerosol/clouds. HSRL is either measuring above or below the aircraft. The shading directly around the flight track seen in part of RF12 and RF17 is a near field effect that leads to erroneous large back scatter signals by HSRL. Green boxes in RF12 and RF17 mark data periods that were used for BrO, IO and NO2 optimal estimation profile retrievals as published in Volkamer et al. (2015). Regular HSRL upward scans show that for these time periods no aerosol or cloud layers were present above the aircraft. For more information see Dix et al. (2016a and b). p.13, line 22ff:

See our comments above. We should also add that the ATTREX mission included a downward looking LIDAR which provided information on clouds and aerosol below the aircraft. However, in most cases this information proved to be insufficient to constrain the radiative transfer with the required accuracy.

This is incorrect, and a misleading reflection of the literature. First, Volkamer and Wang et al. (2015) used a stratospheric model (RAQMS) to study the influence of changing BrO concentrations above, and show that potential changes in the stratospheric BrO VCD, or apparent changes in the measured limb dSCDs due to a changing tropopause altitude do not affect the results. Second, the authors are referred to section 2.10 in Volkamer et al. (2015), and Fig. S4 in the SI text of Wang et al. (2015) for the excellent agreement with the aircraft microwave temperature profiler measurements and the location of the thermal tropopause in the model. Third, the supplement of Volkamer et al. (2015) shows that the stratospheric profile above the aircraft is accurately corrected. Finally, Dix et al. (2016a) used RAQMS BrO profiles for the correction of stratospheric BrO contributions to the limb dSCD measurement, and confirms excellent agreement with the optimal estimation case study profiles from Wang et al. (2015) and Volkamer et al. (2015) using a parameterization method within low error bars.
p.13, line 27ff:

We acknowledge that Wang et al. (2015) and Volkamer et al. (2015) considered the overhead BrO column and that there results rely on model calculations. However, as our study points out, even a sophisticated and well-tested stratospheric CTMs, such as SLIMCAT, have problems accurately simulating the details of the vertical BrO profile at flight altitudes in the UTLS. To our knowledge, RAQMS is not a CTM in which the stratosphere is represented very well, which would worsen this problem.
In particular, certain dynamical processes are often not properly resolved by CTM's (see Figure 2 below). These may include mixing of air masses across the UTLS around the subtropical and polar jet, transient vertically and horizontal propagating gravity waves, Kelvin waves in the tropics, planetary wave in the sub-tropical surface zone and / or those acting at the edge of the polar vortex (Figure 2). We also show in Stutz et al. (Figure 11) that only a fraction $\alpha$ ( = 0.15 - 0.6) of the measured BrO absorption in the TTL/LS is due to line-of-sight absorption, but the majority of the absorption is due to the overhead BrO (and eventually due to light being back-reflected from the troposphere below). Therefore, potential spatial and temporal changes of both contributions to the total absorption have to be carefully considered in the data analysis of the limb observations.

This conclusion is incorrect in all aspects listed. See our above responses. TORERO flight RF12 and RF17 are neither affected by aerosol/cloud extinction above, nor lower level clouds below, nor changing stratospheric BrO.
Please see our responses on the relevant RT (above) and the necessity to properly resolve the (mostly dynamics related) spatial structures of the stratospheric composition in the scale relevant for our method.

We respectfully disagree, and show below that the results presented in Fig. 3b of Werner et al. (2016) and Wang et al., (2015) are in fact quite compatible. Werner et al. show that optimal estimation (OE) profile retrievals in Figs. 3a and b yield within error bars the same results for the altitude range between 14.5 and 18.5 km, regardless of a priori profile choice. This shows that the OE inversion is well constrained by measurements for these altitudes. However, below 14.5 km, the measurements by Werner et al. are not well constrained, and essentially follow the a priori in both cases shown. The "unexpected kink around 12 to 13 km" is therefore not unexpected at all, but to the contrary, it is the expected result of the OE solution that transitions from 'constrained by measurements' (above 14 km) to reproducing the a priori profile at lower altitudes (below 12 km). This behavior is likely reflected in the averaging kernel that are not shown, and should be included in the manuscript. Also, is the OE based on limb spectra only or are downward scans included? This information is missing in the paper.

We run more than 100 test inversions to study the sensitivity of the OE, for example to different a priori profiles, internal and external constraints et cetera. Because of the large sensitivity of the inferred profile on the a priori information for the atmospheric part not directly probed during the limb observation (and due other constraints which are not well-defined), we avoided this approach in our analysis and rather used the newly developed $O_3$ scaling technique (see the green profile in Figure 11 in the revised manuscript).
The kink is because our measurements are not compatible with the findings of Wang et al. BrO profile. As the comment correctly indicates our measurements are barely sensitive to BrO below ~13 km and therefore the retrieved profile is "pulled" towards the a priori BrO profile (Wang et al.). The altitude range 13 – 18 km the average kernels (AK) are around 1 indicating that the inferred BrO is predominately determined by the observations (see Figure 3). We did not include the AK in the paper as we ultimately we do not use the OE approach.
In the end this exercise confirms that our observations and those by Wang et al. above 13km are different and that this difference is likely not due to an OE problem.

Furthermore, section 4.4 in Stutz et al. (2016) states that GH measurements during SF3 are compatible with up to 1.5 pptv of BrO directly below flight altitude. This is quite compatible with the TORERO campaign average BrO vertical profile, which shows a significant decrease of BrO above 14 km, with a mean of 1.86 ± 0.16 pptv at 13.5 km, and 1.38 ± 0.16 pptv at 14.5 km (Dix et al., 2016b).

This upper limit is based on the residual noise of the DOAS retrieval, i.e. [BrO] < 1.5 ppt, and is thus not a proof of the presence of this much BrO. Then interpretation of BrO levels on these scales will depend crucially on the fine-scale dynamics and vertical profiles. It is thus nearly impossible to determine if our observations are compatible with those by Dix et al.

The TORERO average profile is compared with model predictions in Fig. 5 in Schmidt et al., (2016), and is shown with a better resolution in Fig. 10 of Dix et al. (2016b) (included as Fig.2 here). Notably, the case studies in Volkamer et al. (2015) and Wang et al. (2015) are 100% consistent within low error bars (5%) with the parameterization retrieval (Dix et al., 2016a), if the same data subsets are compared. These case studies had probed primarily air masses influenced by convection over oceans. The lower mean BrO for the complete TORERO data set is mainly reflecting different air mass histories, consistent with the variability in Bry noted in Wang et al. (2015), and our hypothesis that sea-salt derived Bry is a source for BrO in the upper free troposphere downwind of marine convection.

We agree that an unrecognized source of bromine is required in the tropical UT/TTL (> 12 km) to explain the TORERO results. Whether this source is sea salt will need to be further investigated as supporting reports are somewhat contradictory. For example Froyd et al., (2009) found in the air analysed from aboard the NASA WB-57 southwest of Central America during the Pre-AVE and CR-AVE campaigns in February 2004 and 2008 that the fraction of sea salt containing aerosols strongly decreased from < 5% in the 4 - 12 km region to virtually zero above 12 km (Figure 4 in Froyd et al.,

2009). Schmidt et al. (2006) modelled a total Br*y* of ~3 ppt in the middle troposphere of the tropics (Figure 2). Accordingly, about 1 ppt might due to bromine released by sea salt given the 30% statement cited above. Therefore, it is hard to see how sea salt may give rise to 2 ppt of [BrO] at 13.5 km in the tropics during daytime. Nevertheless, it is difficult to rule out this source for a particular observation and thus it seems prudent to further investigate the possibility of a sea salt source.

In the end we think the overreaching question is why the two studies disagree and which of the two observations is more representative of the TTL. This is a question that will only be answered through further observations.

In the acknowledgments, we added the following sentence: The authors are grateful for the comments given by two anonymous reviewers, and the comments of Barbara Dix and Rainer Volkamer (CU, Boulder, USA).

Refs:
1. Hueneke T., The scaling method applied to HALO measurements: Inferring absolute trace gas concentrations from airborne limb spectroscopy under all sky conditions, PhD thesis, Institut für Umweltphysik, Universität of Heidelberg, Heidelberg, Germany, 2016.
2. Froyd, K. D., D. M. Murphy, T. J. Sanford, D. S. Thomson, J. C. Wilson, L. Pfister, and L. Lait (2009), Aerosol composition of the tropical upper troposphere, *Atmos. Chem. Phys.*, *9*(13), 4363–4385, doi:10.5194/acp-9-4363-2009.
3. Murphy, D. M., Thomson, D. S., and Mahoney, M. J.: In Situ Measurements of Organics, Meteoritic Material, Mercury, and Other Elements in Aerosols at 5 to 19 Kilometers, Science, 282, 1664, doi:10.1126/science.282.5394.1664, 1998.
4. Murphy, D. M. and Thomson, D. S.: Halogen ions and NO+ in the mass spectra of aerosols in the upper troposphere and lower stratosphere, Geophys. Res. Lett., 27, 3217–3220, doi:10.1029/1999GL011267, 2000.
5. Murphy, D. M., Cziczo, D. J., Hudson, P. K., and Thomson, D. S.: Carbonaceous material in aerosol particles in the lower stratosphere and tropopause region, J. Geophys. Res., 112, 4203, doi: 10.1029/2006JD007297, 2007. Oikarinen, L.: Effect of surface albedo variations on UV-visible limb-scattering measurements of the atmosphere, J. Geophys. Res., 107, 15 1–15, doi:10.1029/2001JD001492, 2002.-
6. Raecke 2013; Atmospheric Spectroscopy of Trace Gases and Water Vapor in the Tropical Tropopause Layer from the NASA Global Hawk, Master Thesis, University of Heidelberg, Heidelberg, 2013. Upon request we can provide a digital copy of the thesis.
7. Schmidt, J. A., Jacob, D. J., Horowitz, H. M., Hu, L., Sherwen, T., Evans, M. J., Liang, Q., Suleiman, R. M., Oram, D. E., Le Breton, M., Percival, C. J., Wang, S., Dix, B., and Volkamer, R.: Modeling the observed tropospheric BrO background: Importance of multi-phase chemistry and implications for ozone, OH, and mercury, Journal of Geophysical Research: Atmospheres, 121, 11,819–11,835, doi:10.1002/2015JD024229, http://dx.doi.org/10.1002/2015JD024229, 2015JD024229, 2016.

[Figure]

Figure 1: Horizontal sensitivity of DOAS measurements from the Global Hawk: Shown are horizontal box sensitivities in the altitude grid layer of flight altitude for a DOAS measurement with 0° telescope elevation angle at 18:24 UT, 27:1 ° N / 133:5 ° W, SZA = 57:2 °, SRAA = 102:2 °, detector altitude 17.1 km (top), and for a DOAS measurement at point P, 02:03 UT, 5:3 ° S / 150:0 ° W, SZA = 57:2 °, SRAA = 77:9 °, detector altitude 18.2 km (bottom). The yellow arrow line denotes the incident direction of photons from the sun. The intersection with the black line (flight path) describes the model detector position. Every tick on the black line describes a 0 ° DOAS measurement (from Raecke, 2013).

[Figure]

Figure 2: CLaMS predicted curtains of $O_3$, BrO and $NO_2$ and OClO for the Polstracc HALO flight from Kiruna on January 31, 2016 as function of flight time. The red lines indicate the flight trajectory of the aircraft. Please note that (a) the spatial and temporal structure of $O_3$, $NO_2$, BrO and OClO modelled by

CLaMS in the stratosphere (c.f., mixing around the the polar jet at around 11:45 UTC and less around 9:45 UTC), which show the challenge in assuming a constant overhead slant column for DOAS based concentration retrievals, and (b) that CLaMS is not not very skillfull in modelling BrO in the troposphere, due to missing bromine sinks derived from borminated hydrocarbon degradation. The figure was kindly provided by J.U. Groß, Forschungszentrum Jülich, Jülich Germany.

[Figure]

Figure 3: Averaging Kernels of the OE inversion of Fig. 3b (used in the comparison with the Wang et al., 2015 BrO profile.)

---

## Editor Decision (ED1)

P1, L3: Change "see" to "see a companion paper";

P1, L5: Remove "(14.5 -18.5 km)", which is repeated in L14-15;

P2, L1-2: Using the symbol θ's or θs to represent potential temperature can be misleading as one may consider it as another variable different from θ. I would suggest changing "for θ's" to "for potential temperature (θ)" and "θs =" to "θ =" (in Line 1), and changing "for θ's" to "for θ" (in Line 2).